



# About the effects of polarising optics on lidar signals and the Δ90-calibration

**Volker Freudenthaler**[1]
[1]Fakultät für Physik, Meteorologisches Institut, Ludwig–Maximilians–Universität, 80333
München, Theresienstrasse 37, Germany
Correspondence to: V. Freudenthaler (volker.freudenthaler@lmu.de)
**Abstract**
This paper provides a model for assessing the effects of polarising optics on the signals of
typical lidar systems, which is based on the description of the individual optical elements of
the lidar and of the state of polarisation of the light by means of the Müller-Stokes formalism.
General analytical equations are derived for the dependence of the lidar signals on
polarisation parameters, for the linear depolarisation ratio, and for the signals of different
polarisation calibration set-ups. The equations can also be used for the calculation of
systematic errors caused by non-ideal optical elements, their rotational misalignment, and by
non-ideal laser polarisation. We present the description of the lidar signals including the
polarisation calibration in a closed form, which can be applied for a large variety of lidar
systems.
**1   Introduction**
The purpose of atmospheric depolarisation measurements with lidar, first described by
Schotland et al. (1971), is mainly to discern between more or less depolarising scatterers. The
discrimination of ice and water clouds was the main focus in the beginning. Sassen (1991)
and Sassen (2005) give an overview about the early work related to that. Aerosol and their
interaction with clouds became more important in the last decade because of their
insufficiently understood direct and indirect roles in the feedback mechanisms of climate
change (Boucher et al., 2013). Multi-wavelength lidar measurements including the
depolarisation ratio can be used to discern aerosol types (Sugimoto et al., 2002; Sugimoto and
Lee, 2006; Ansmann et al., 2011; Burton et al., 2014; Groß et al., 2014) and to retrieve micro-
physical aerosol properties by means of inversion algorithms (Müller et al., 1999; Ansmann





and Müller, 2005; Gasteiger et al., 2011; Veselovskii et al., 2013; Böckmann and Osterloh,
2014; Müller et al., 2014) . Pérez-Ramírez et al. (2013) show the impact of systematic errors
of the lidar data on the retrieval of micro-physical particle properties. The additional
measurement of the linear (or circular) depolarisation ratio improves the retrievals (Böckmann
and Osterloh, 2014; Gasteiger and Freudenthaler, 2014). But the depolarisation ratios are
often derived from lidar measurements assuming more or less ideal lidar set-ups neglecting
the effects of small system misalignments and of non-ideal optical elements on the
polarisation, which can lead to considerable errors in the retrieved depolarisation ratio
(Reichardt et al., 2003; Alvarez et al., 2006; Freudenthaler et al., 2009; Mattis et al., 2009).
According to Chipman (2009a) Chap. 15.27, one of the primary difficulties in performing
accurate polarisation measurements is the systematic error due to non-ideal polarisation
elements. Most inclined optical surfaces and optical coatings on beam-splitters are polarising,
wherefore all lidars must be considered as "incomplete light-measuring polarimeters"
(Chipman, 2009a), even if they are not intended to measure the depolarisation ratio.
As model calculations of aerosol scattering properties advance (Nousiainen et al., 2011;
Kahnert et al., 2014), the modellers need accurate measurements with small and reliable error
bars in order to verify and improve their models. In order to estimate the uncertainties and to
improve the measurements, we have to find the error sources. The usual way to do this is to
compare the measurements with a model and to investigate the deviations. The only reliable
atmospheric model for comparison is the model of the molecular linear depolarisation ratio $\delta_m$
(Behrendt and Nakamura, 2002; Freudenthaler et al., 2015). But the measured values $\delta_m^*$ of
the very small real $\delta_m$ (on the order of 0.004) are usually a number of times higher, which
makes it difficult to use for calibration with a simple model as $\delta_m^* = A\delta + B$ ((Sassen and
Benson, 2001; Reichardt et al., 2003); see also S.9). At present, polarisation calibration
techniques of lidars are often not accurate enough to sufficiently determine the two
parameters $A$ and $B$, and actually, as we will show in the following, the model itself is
insufficient. But how accurate do we have to be? How accurate can we be? What are the
critical parts and adjustments? How can set-ups be improved with minimal costs and
complexity, and how can existing lidar systems be checked? To answer these questions, we
need a better model for the lidar set-up, which is complete and flexible enough to be applied
to a variety of lidar systems and can describe various calibration techniques.



Astronomical polarisation measurement set-ups are very similar to lidar set-ups. Elaborate
theoretical and experimental investigations of the influence of polarising optics and
corresponding corrections for astronomical telescopes and detection optics using the theory of
polarimetry and ellipsometry (see Azzam (2009); Chipman (2009a)) can be found quite
frequently in the literature (Skumanich et al., 1997; Socas-Navarro et al., 2011; Breckinridge
et al., 2015). Although the usefulness of a lidar with polarisation diversity had been realised
early (Pal and Carswell, 1973), the need for a complete description with the Müller-Stokes
formalism has, to our knowledge, been first expressed by Anderson (1989), but focused only
on the atmospheric scattering process. Instrumental aspects including some error calculations
have been included by Beyerle (1994), Cairo et al. (1999), Biele et al. (2000), Behrendt and
Nakamura (2002), Reichardt et al. (2003), Alvarez et al. (2006), Del Guasta et al. (2006),
Hayman and Thayer (2009), Mattis et al. (2009), (Freudenthaler et al., 2009), Hayman (2011),
Hayman and Thayer (2012), David et al. (2013), Geier and Arienti (2014), Di et al. (2015),
and Volkov et al. (2015). The errors mainly considered are the diattenuation of the receiver
optics (see Sect. 2.2), the cross talk of the polarising beam-splitter, non-ideal characteristics of
the calibration, and rotational misalignment of polarising components.
In this work we describe lidar set-ups from the laser to the detector by means of the Stokes-
Müller formalism (Chipman, 2009b) including the transmitter and receiver optics. The Stokes
vector describes the flux and the state of polarisation of the light, and the Müller matrices
describe how optical elements change the Stokes vector. We develop equations for the two
signals of a polarisation sensitive lidar and for the signals of the polarisation calibration,
which are necessary to retrieve the linear depolarisation ratio and the total lidar signal, using
different calibration techniques and lidar set-ups. In order to enable the evaluation of the final
errors and to analyse their dependencies on certain optical parameters or misalignments of
individual optical elements, we will derive first the full equations and then try to find more
simple analytical formulations neglecting minor error sources to get an overview of the main
critical parameters.
For this we neglect the polarisation effects of lenses and of telescope mirrors with small
incidence angles of the light beam (Seldomridge et al., 2006) (Clark and Breckinridge, 2011),
but 45° folding mirrors as in Newtonian-type telescopes must be considered (Breckinridge et
al., 2015; Di et al., 2015), and stress-birefringence in windows and lenses or unfavourable
coatings may cause severe polarisation effects. Errors caused by a light beam which is



divergent or inclined towards the optical axis are not discussed here; this means the light
beams are assumed to be either perfectly parallel before and after polarisation optics, or that
an optical element is insensitive to the incident angle regarding polarisation.
Basic information about the polarisation topics can be found in Goldstein (2003), Clarke
(2009), and in the chapters by Azzam (2009); Bennett (2009a,b); Chipman (2009b,a) of the 3$^{rd}$
edition of the Handbook of Optics (Bass, 2009). The authors of these chapters follow the
Muller-Nebraska convention Muller (1969) for the definition of signs and directions regarding
e.g. the coordinate system (see S.1), as we do in this work.

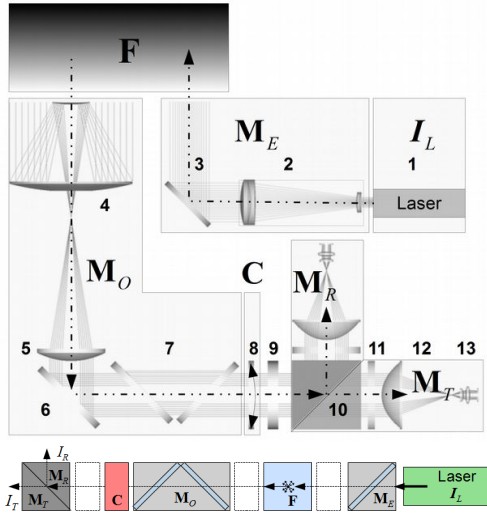

Figure 1  Top: Exemplary depolarisation lidar set-up with laser 1, beam expander 2, steering
mirror 3, receiving telescope 4, collimator 5, folding mirror 6, dichroic beam-splitters 7, a
rotating element for polarisation calibration 8, interference filter 9, and polarising beam-
splitter cube 10 (PBS, polarising beam-splitter). The neutral density filters and cleaning
polarisers 11, detector optics 12, and the detectors 13. The system can be subdivided in
functional blocks which can be described with the Stokes-Müller formalism: $I_L$ is the Stokes
vector of the laser source, $M_E$ is the Müller matrix of the the laser emitter optics, $F$ of the
atmospheric backscattering volume including depolarisation, $M_O$ includes receiver optics as
beam-splitters, $C$ is the calibrator, and $M_{T,R}$ is the polarising beam-splitter including the
detector optics for the transmitted (T) and reflected (R) optical branches. Bottom: simplified
schematic of the setup.



Most of the lidar set-ups for depolarisation measurement reported in the literature are
explicable with the schematic in Fig. (1), in which the individual parts of a lidar system are
grouped in modules, which are in general describable by Müller matrices of combinations of
diattenuators, retarders, and rotators (see Sect.2.2). The set-up in Fig. 1 can be described with
Eq. (1).
$I_{T,R} = \eta_{T,R} \mathbf{M}_{T,R} \mathbf{C} \mathbf{M}_O \mathbf{F} \mathbf{M}_E I_L$ (1)
Symbols for Müller matrices are bold ($\mathbf{M}$), vectors are bold-italic ($\boldsymbol{I}$), and variables italic ($I$).
The laser beam with Stokes vector $I_L$ is expanded and directed towards the atmosphere with
backscatter matrix $\mathbf{F}$ by the emitter module with Müller matrix $\mathbf{M}_E$. The backscattered photons
are received by the telescope with a subsequent collimation lens and dichroic beam-splitters in
the receiver optics module $\mathbf{M}_O$. A polarisation calibrator with Müller matrix $\mathbf{C}$ is placed here
before the polarising beam-splitter cube (10) with Müller matrices $\mathbf{M}_T$ for the transmitted and
$\mathbf{M}_R$ for the reflected path, their opto-electronic gains $\eta_{T,R}$, and the final Stokes vectors $I_{T,R}$ at
the detectors. The opto-electronic gains $\eta_{T,R}$ include the attenuation of all non-polarising
optical elements as neutral density and bandpass filters and, the quantum efficiency of the
detectors, and the amplification of the electronic system. The scattering volume $\mathbf{F}$ can be at
any distance from the lidar (lidar-range), because we assume that the extinction in the range
between the lidar and the scattering volume $\mathbf{F}$ is polarisation independent and that signal
contributions due to forward or multiple scattering in this range can be neglected. Therefore
we neglect all lidar-range dependencies in the following equations.  We also do not consider
range dependent effects as the overlap function and the range dependent transmission of
interference filters and dichroic beam-splitters, which are sensitive to the also range
dependent incident angle on the optics.
Various lidar systems employ different calibration techniques with calibrating devices with
Müller matrix $\mathbf{C}$ at different places in the optical setup, with the respective equations:
before the polarising beam-splitter           $I_S = \eta_S \mathbf{M}_S \mathbf{C} \mathbf{M}_O \mathbf{F} \mathbf{M}_E I_L$ (2)
before the receiver optics           $I_S = \eta_S \mathbf{M}_S \mathbf{M}_O \mathbf{C} \mathbf{F} \mathbf{M}_E I_L$ (3)
behind the laser emitter optics           $I_S = \eta_S \mathbf{M}_S \mathbf{M}_O \mathbf{F} \mathbf{C} \mathbf{M}_E I_L$ (4)
before the laser emitter optics           $I_S = \eta_S \mathbf{M}_S \mathbf{M}_O \mathbf{F} \mathbf{M}_E \mathbf{C} I_L$ (5)





In the following we report just a few examples from the literature with sufficient description
of their calibration technique. Pal and Carswell (1973) used three telescopes with Glan-
Thomson prisms in the receiver optics (Eq. (2)) at 0°, 45°, and 90° orientation with respect to
the laser polarisation to determine the first three Stokes parameters of the scattered light, and
calibrated them by mechanically switching all polarisers to 0° orientation. Houston and
Carswell (1978) extended this set-up by a fourth telescope with a $\lambda/4$ plate to measure all four
Stokes parameters, with the same calibration technique as before. The relative polarisation
sensitivity of the CALIOP lidar on CALIPSO (Winker et al., 2009) is calibrated with a
pseudo-depolariser before the polarising beam-splitter (Hunt et al., 2009), which is described
by Eq. (2). Del Guasta et al. (2006) calibrate the gain ratio $\eta_R/\eta_T$ .of their polarimetric lidar
with an unpolarised light source before the polarising beam-splitter (Eq. (2)) and determine
the receiving optics Müller matrix $\mathbf{M}_O$ with a linearly polarised light source and rotating the
receiving optics, which corresponds to Eq. (3) with a mechanical rotation matrix $\mathbf{C}$. Similar
rotation calibration before the polarising beam-splitter is applied with RALI (Nemuc et al.,
2013) and the Raymetrics LR331D400 (Bravo-Aranda et al., 2013)  with a mechanical
rotation Δ90-calibration (see Sect. 5), and with a $\lambda/2$ plate rotation in the MULIS
(Freudenthaler et al., 2009) and the Cloud Physics Lidar (McGill et al., 2002; Liu et al.,
2004). A sheet polariser at 45° is used before the polarising beam-splitter in the AD-Net lidars
(Shimizu et al., 2004). Mechanical rotation before the receiving optics (Eq. (3)) is employed
for the DLR HSRL (Esselborn et al., 2008), for POLIS (Freudenthaler et al., 2009), and by
Nisantzi et al. (2014). For the McMurdo lidar (Snels et al., 2009) and the PollyXT
(Engelmann et al., 2015) a linear polariser is used before the receiving optics. An unpolarised
light source before the receiver telescope is used by Mattis et al. (2009).  Spinhirne et al.
(1982) use a $\lambda/2$ plate for polarisation rotation in the output beam (Eq. (4)). The HSRL-1
(Hair et al., 2008) and HSRL-2 (Burton et al., 2015) as well as David et al. (2012) use a $\lambda/2$
plate as rotation calibrator before some parts of the emission optics (Eq. (5)). Roy et al. (2011)
and Cao et al. (2010) use a $\lambda/2$ plate before the emitter optics (Eq. (5)), but they switch the
plane of emitted polarisation continually between horizontal and vertical and calculate the
linear depolarisation ratio from the geometric mean of both measurements, which makes a
separate calibration unnecessary. However, the equations of this work can still be used for the
error analysis. Polarisation switching between laser pulses and with only one detection
channel is done by Platt (1977)  with mechanical rotation of the receiver optics, by  Eloranta
and Piironen (1994) with a $\lambda/2$ plate after the emitter optics (Eq. (4)), by Seldomridge et al.





(2006) with a nematic liquid crystal before the polarising beam-splitter (Eq. (2)), and by
(Flynn et al., 2007) with a $\lambda/2$ plate before the emitter optics (Eq. (5)). Although the explicit
equations in this work consider only one variable polarising element (i.e. the calibrator), the
equations for more complex lidar setups as with a polarising beam-splitter and a $\lambda/4$ plate in
the common emitter/receiver path ((Eloranta, 2005; David et al., 2013) or with different
variable polarisation elements in the emitter/receiver path (Kaul et al., 2004; Hayman et al.,
2012; Volkov et al., 2015) can be constructed with the equations provided in this work. Snels
et al. (2009) present an overview of some potential error sources and other existing
polarisation calibration techniques including calibration with assumed known depolarisation
from molecules ("clear sky") or clouds with spherical particles.
The equations presented in this work can be used for the design of lidar systems, especially
for the determination of the requirements for certain components in order to achieve the
desired measurement accuracy, for the analysis of the performance of existing lidar systems
by means of different calibration set-ups, and for the final error calculation with respect to the
polarisation characteristics.
One of the main uncertainties is the orientation of the plane of polarisation of the laser beam
(angle $\alpha$) with respect to the orientation of the polarising beam-splitter (briefly laser rotation),
because first, the plane of polarisation of the laser might not only be determined by the
orientation of the Pockels cell in the laser cavity, but also by the orientation of the crystals for
second and third harmonics generation and by the harmonic separation beam-splitters.
Second, the laser and emitter optics are often mounted on a separate optical breadboard,
which might be rotated with respect to the receiver breadboard. Furthermore, laser
manufacturers usually provide neither an indication of the accuracy of the orientation nor an
accurate mechanical reference for it, the orientation cannot be measured easily, and finally, the
orientation can change with time and environmental conditions. We take into account that in
lidar labs it is usually not possible to perform elaborate and accurate measurements as in an
optical lab equipped for ellipsometric measurements. Therefore we want to use simple tools
and as few as possible measurements - at best with the tools which we already use for the
atmospheric depolarisation measurements.
Some optical parts can be made almost ideal and some misalignments can be made very small
so that they become negligible. For these cases often much simpler equations can be derived,
which show the residual influence of the other non-ideal parts, and which can be used directly



in lidar retrieval algorithms. It becomes also clear in which cases corrections are not possible,
when additional measurements with simple set-ups can help to retrieve the properties of the
disturbing parts, and where one has to be careful in the design of a lidar system to avoid non-
correctable errors. We want to find the set-ups and calibrators, with which the calibration can
be measured with the least errors, and we want equations to assess the final uncertainties in
the retrieved lidar products. Set-ups with 90° separated limit stops can be made very accurate
(< 0.1°) by means of working machines. Motorised holders with sufficient resolution and
accuracy are commercially available. An example for an almost ideal part is the linear
polariser. Polarising sheet filters are available with high extinction, well specified by
manufacturers. They are relatively insensitive to the incident angle, work over a sufficiently
large wavelength range, and are thin, which means that they can be placed even in already
existing lidar systems with little space for additional optics. Additionally, they are available in
large size at an affordable price - in contrast to crystal polarisers and wave plates, and thus
they can also be placed before the telescope. Wave plates and circular polarisers made of
plastic sheets are usually not as well specified concerning their phase shift, acceptance angle
and wavelength range. For other places, which require only small diameters, true zero-order
$\lambda/2$ plates can be used.
Since the atmosphere is not stable and the laser power might change between two consecutive
measurements, the absolute signals change. But if we use the ratios of the cross and parallel
signals, which only change with the atmospheric polarisation parameter $a$, we can easily find
atmospheric situations which introduce negligible errors in the calculations. Therefore we
only use signal ratios for the calibrations.
Most of the problems can probably be solved with a much smaller theoretical framework. But
then often questions arise, how the one or other misalignment, rotation, additional retardance
or diattenuation would influence the final results. The impotence of less extended
formulations to answer these questions will always leave an uncomfortable uncertainty. This
work is an attempt to provide the tools to answer some of these questions, with the
disadvantage of being rather extended.
Section 2 provides a simplified example as an introduction and preparation for Sect. 3, where
we introduce the concepts and parameters which are necessary to formulate the equations in
such a general way that they can be applied to a large variety of lidar systems. In order to
generalise and to simplify the expressions, several binary parameters are introduced in the


equations, which enable us to describe orthogonal orientations of individual elements with
just one expression and which reduce the number of equations considerably. In Sect. 4 we
develop the general equations for the lidar signals of normal atmospheric measurements
(*standard measurements* in the following) and for the linear depolarisation ratio. In Sect. 5 we
introduce the general concept of the 45° and Δ90 calibrations, which is then applied in Sect. 6
to 10 for different calibrators and in the subsections for different positions of the calibrators in
the emitter-receiver optics. We inlcude the following types of calibrators: unpolarised light
(Sect. 6), which has to be inserted by an additional light source or diffuser and has therefore
some disadvantages; the mechanical and λ/2 plate rotator (Sect. 7); the linear polariser (Sect.
8), which can be easily included in existing systems; the λ/4 plate (Sect. 9), which can also be
used to determine the amount of circular polarisation (S.14); and the circular polariser (Sect.
10). General purpose equations used in several sections are shifted to the appendices, and
common equations or concepts, which can also be found in standard text books, are collected
in the supplement in order to show their form with the variables used in this work.
**2    The basic Müller-Stokes representation of lidar signals with polarisation**
In this chapter we use a simple example of Fig.(1), described with Eq. (2), to introduce some
basic concepts. It contains a calibrator **C** before the polarising beam-splitter and neglects the
polarising effects of the receiver optics $\mathbf{M}_O$, i.e.
$$\boldsymbol{I}_{T,R} = \eta_{T,R}\mathbf{M}_{T,R}\mathbf{C}\boldsymbol{F}\boldsymbol{I}_L \qquad (6)$$
The total power $I_L$ and the state of polarisation of horizontal-linear polarised laser light are
represented by the Stokes vector
$$\boldsymbol{I}_L = I_L \begin{pmatrix} 1 \\ 1 \\ 0 \\ 0 \end{pmatrix} \qquad (7)$$
The magnitude $I_L$ of the Stokes vector is the total light beam intensity. It is directly
measurable with a light detector for the flux of photons. Because a lidar includes optics as
telescope and lenses, which change the diameter or focus the light beam, here the colloquial
intensity means the radiant flux or radiant energy per unit time. However, the finally
measured quantities are the electronic signals $I_T$ and $I_R$ of the detectors in the transmitted and
reflected paths. We use flux, intensity and signal alternatively, depending on the context.




## 2.1 Depolarising atmospheric aerosol
Müller matrices describe the linear interaction between polarised light and an optical system
(optical elements or medium). For any input, represented as a Stokes vector, the Müller matrix
produces a unique output, in the form of another Stokes vector. For the backscattering of a
volume of randomly oriented, non-spherical particles with rotation and reflection symmetry
the Müller matrix $\mathbf{F}$ can be written as (van de Hulst, 1981; Mishchenko and Hovenier, 1995;
Mishchenko et al., 2002)

$$\mathbf{F} = \begin{pmatrix} F_{11} & 0 & 0 & 0 \\ 0 & F_{22} & 0 & 0 \\ 0 & 0 & -F_{22} & 0 \\ 0 & 0 & 0 & F_{44} \end{pmatrix} = F_{11} \begin{pmatrix} 1 & 0 & 0 & 0 \\ 0 & a & 0 & 0 \\ 0 & 0 & -a & 0 \\ 0 & 0 & 0 & 1-2a \end{pmatrix} \tag{8}$$

with the polarisation parameter $a$ (Chipman, 2009b; Eq. (93))

$$a = \frac{F_{22}}{F_{11}} \tag{9}$$

and

$$F_{44} = F_{11} - 2F_{22} = F_{11}(1-2a) \tag{10}$$

Note, that in some literature (Flynn et al., 2007; Gimmestad, 2008; Roy et al., 2011; Gasteiger
and Freudenthaler, 2014) the *de-polarisation parameter* $d = (1 - a)$ is used, and in Borovoi et
al. (2014) $d$ is called *polarisation parameter*. In Volkov et al. (2015) $e = a$ (for randomly
oriented particles) is called *sphericity index*. However, in this work we use the polarisation
parameter $a$ for the reason of brevity, which is the fraction of the backscattered light that
maintains the emitted linear polarisation.
The matrix $\mathbf{F}$ in Eq. (8) describes a pure depolariser $\mathbf{M}_\Delta$ (Lu and Chipman, 1996), but
including a mirror reflection $\mathbf{M}_M$ for the backscattering direction, with the backscatter
coefficient $F_{11}$.

$$\mathbf{F} = \mathbf{M}_M \mathbf{M}_\Delta = F_{11} \begin{pmatrix} 1 & 0 & 0 & 0 \\ 0 & 1 & 0 & 0 \\ 0 & 0 & -1 & 0 \\ 0 & 0 & 0 & -1 \end{pmatrix} \begin{pmatrix} 1 & 0 & 0 & 0 \\ 0 & a & 0 & 0 \\ 0 & 0 & a & 0 \\ 0 & 0 & 0 & 2a-1 \end{pmatrix} \tag{11}$$



$F_{11}$ and $a$ are the only range dependent parameters in all the following equations. The volume
linear depolarisation ratio $\delta$ of the scattering volume, which contains particles and air
molecules, can be written as (Mishchenko and Hovenier, 1995)

$$\delta = \frac{F_{11} - F_{22}}{F_{11} + F_{22}} = \frac{1-a}{1+a} \Rightarrow a = \frac{1-\delta}{1+\delta} \tag{12}$$

The Stokes vector $\boldsymbol{I}_{in}$ of horizontal-linear polarised light $I_L$ reflected by the atmosphere $\mathbf{F}$ and
incident in the receiving optics is

$$\boldsymbol{I}_{in} = \mathbf{F}\boldsymbol{I}_L = F_{11} \begin{pmatrix} 1 & 0 & 0 & 0 \\ 0 & a & 0 & 0 \\ 0 & 0 & -a & 0 \\ 0 & 0 & 0 & 1-2a \end{pmatrix} I_L \begin{pmatrix} 1 \\ 1 \\ 0 \\ 0 \end{pmatrix} = F_{11} I_L \begin{pmatrix} 1 \\ a \\ 0 \\ 0 \end{pmatrix} \tag{13}$$

## 2.2   Optical parts: diattenuator with retardation
All other optical elements in the lidar receiver can be described as a combination of
diattenuators and retarders (Lu and Chipman, 1996) (retarding diattenuators; Eq. (14)). Often
a polarising beam-splitter cube is used for splitting in transmitted and a reflected components
polarised parallel and perpendicular with respect to the laser polarisation. But also polarising
or even non-polarising beam-splitter plates with subsequent polarisation filters (analysers) can
be used. All of them and combinations of them can be described with the Müller matrix of a
polarising beam-splitter (PBS) (Pezzaniti and Chipman, 1994), considering the remarks in
S.4. The matrix of the transmitting part is

$$\mathbf{M}_T = \frac{1}{2} \begin{pmatrix} T_T^p + T_T^s & T_T^p - T_T^s & 0 & 0 \\ T_T^p - T_T^s & T_T^p + T_T^s & 0 & 0 \\ 0 & 0 & 2\sqrt{T_T^p T_T^s}\cos\Delta_T & 2\sqrt{T_T^p T_T^s}\sin\Delta_T \\ 0 & 0 & -2\sqrt{T_T^p T_T^s}\sin\Delta_T & 2\sqrt{T_T^p T_T^s}\cos\Delta_T \end{pmatrix} =$$
$$= T_T \begin{pmatrix} 1 & D_T & 0 & 0 \\ D_T & 1 & 0 & 0 \\ 0 & 0 & Z_T c_T & Z_T s_T \\ 0 & 0 & -Z_T s_T & Z_T c_T \end{pmatrix} \tag{14}$$

with the intensity transmission coefficients (transmittance) for light polarised parallel ($T^p$) and
perpendicular ($T^s$) to the plane of incidence of the PBS, the diattenuation parameter $D_T$, and
the average transmittance $T_T$, i.e. for unpolarised light. $\Delta_T$ is the difference of the phase shifts





of the parallel and perpendicular polarised electrical fields (retardance) according to the
Muller-Nebraska convention (Muller, 1969).

$$T_T = \frac{T_T^p + T_T^s}{2}, \quad D_T = \frac{T_T^p - T_T^s}{T_T^p + T_T^s}, \quad Z_T = \frac{2\sqrt{T_T^p T_T^s}}{T_T^p + T_T^s} = \sqrt{1 - D_T^2},$$
$$c_T = \cos\varDelta_T, \quad s_T = \sin\varDelta_T, \quad \varDelta_T = \varphi_T^p - \varphi_T^s \tag{15}$$

Please note, that this definition differs in two ways from the definition in Chipman (2009b):
the retardance is defined differently there ($\varDelta_X = \varphi_X^S - \varphi_X^P$), and we denote with $D$ the
horizontal diattenuation parameter $d_h$ (Chipman, 2009b) and not the diattenuation magnitude
$D_{mag} = |D|$ (see S.4). The Müller matrix for the reflecting part of the PBS Eq. (16) includes a
mirror reflection (S.6) with the corresponding intensity reflection coefficients (reflectance) for
light polarised parallel ($R_p = T_R{}^p$) and perpendicular ($R_s = T_R{}^s$) to the plane of incidence (S.1)
of the polarising beam-splitter.

$$\mathbf{M}_R = T_R \begin{pmatrix} 1 & D_R & 0 & 0 \\ D_R & 1 & 0 & 0 \\ 0 & 0 & -Z_R c_R & -Z_R s_R \\ 0 & 0 & Z_R s_R & -Z_R c_R \end{pmatrix} = T_R \begin{pmatrix} 1 & 0 & 0 & 0 \\ 0 & 1 & 0 & 0 \\ 0 & 0 & -1 & 0 \\ 0 & 0 & 0 & -1 \end{pmatrix} \begin{pmatrix} 1 & D_R & 0 & 0 \\ D_R & 1 & 0 & 0 \\ 0 & 0 & Z_R c_R & Z_R s_R \\ 0 & 0 & -Z_R s_R & Z_R c_R \end{pmatrix} \tag{16}$$

$$T_R = \frac{T_R^p + T_R^s}{2}, \quad D_R = \frac{T_R^p - T_R^s}{T_R^p + T_R^s}, \quad Z_R = \frac{2\sqrt{T_R^p T_R^s}}{T_R^p + T_R^s} = \sqrt{1 - D_R^2},$$
$$c_R = \cos\varDelta_R, \quad s_R = \sin\varDelta_R, \quad \varDelta_R = \varphi_R^p - \varphi_R^s \tag{17}$$

In order to simplify the derivation of the equations, we describe both the reflecting and
transmitting matrices with the matrix $\mathbf{M}_S$, and replace the subscript $_S$ (for splitter) by $_T$
(transmitting) or $_R$ (reflecting) where appropriate, which means
$D_S \in \{D_R, D_T\}, \quad \mathbf{M}_S \in \{\mathbf{M}_R, \mathbf{M}_T\}, \quad I_S \in \{I_R, I_T\}$        (18)
It has to be emphasised, that for this reason we can't use the diattenuation magnitude $D_{mag}$,
which is always positive and almost exclusively used in other publications, but have to use the
diattenuation parameter $D$, which changes the sign when $T_R^s$ becomes larger than $T_R{}^p$ (see
S.3). Please keep also in mind that usually $D_R < 0$, that $\mathbf{M}_R$ includes an additional mirror
reflection, and that fluxes measured after the PBS are not influenced by the addition of an
ideal mirror reflection in the optical path.
**2.3 Calibration, linear depolarisation ratio, and total signal**
Eq. (6) shows the Stokes vectors of the transmitted ($I_T$) and reflected ($I_R$) channels, alias $I_S$,
after the polarising beam-splitter $\mathbf{M}_S$ (PBS) without calibrator, i.e. $\mathbf{C} = \mathbf{1}$ = identity matrix. Eq.
(6) represents the standard lidar measurement at the axial rotation of 0°, neglecting for now
additional optics in $\mathbf{M}_O$.

$$I_S\left(0°\right) = \eta_S \mathbf{M}_S \mathbf{F} I_L = \eta_S \mathbf{M}_S I_{in} =$$

$$= \eta_S T_S \begin{pmatrix} 1 & D_S & 0 & 0 \\ D_S & 1 & 0 & 0 \\ 0 & 0 & Z_S c_S & Z_S s_S \\ 0 & 0 & -Z_S s_S & Z_S c_S \end{pmatrix} F_{11} I_L \begin{pmatrix} 1 \\ a \\ 0 \\ 0 \end{pmatrix} = \eta_S T_S F_{11} I_L \begin{pmatrix} 1 + D_S a \\ D_S + a \\ 0 \\ 0 \end{pmatrix} \tag{19}$$

The measured signals $I_S$ are

$$I_S\left(0°\right) = \eta_S T_S F_{11} I_L \left(1 + D_S a\right) \tag{20}$$

which correspond to the transmitted and reflected intensities, include the individual channels
gains $\eta_S$, i.e. $\eta_T$ and $\eta_R$, which are the product of the electronic amplification of the detectors,
the amplifiers, and of the optical attenuation due to polarisation insensitive attenuation of all
optics including neutral density and interference filters. The latter is in general different in the
two channels. We can solve the equation of the ratio of the measured reflected to the
transmitted signals

$$\frac{I_R}{I_T}\left(0°\right) = \frac{\eta_R T_R \left(1 + D_R a\right)}{\eta_T T_T \left(1 + D_T a\right)} = \frac{\eta_R \left(T_R^p + T_R^s \delta\right)}{\eta_T \left(T_T^p + T_T^s \delta\right)} \tag{21}$$

for the linear depolarisation ratio $\delta$ if we know the calibration factor

$$\eta \equiv \frac{\eta_R T_R}{\eta_T T_T} \tag{22}$$

(with reflectance $T_R$ and transmittance $T_T$ for unpolarised light) and the transmission
parameters of the polarising beam-splitter $T_T^p$, $T_T^s$, $T_R^p$, and $T_R^s$ for the correction of its cross
talk. We could get the calibration factor $\eta$ already with the measurements in Eq. (21) if the
light incident on the analyser was unpolarised, i.e. $a = 0$. Else, $\eta$ can be determined by means
of calibration measurements, e.g. by rotating the the PBS including the detectors by +45° or
−45° about the optical axis (Eq. (23)).





$$I_S(\pm 45°) = \eta_S \mathbf{M}_S \mathbf{R}(\pm 45°)\mathbf{F}I_{in} =$$

$$= \eta_S \mathbf{M}_S \begin{pmatrix} 1 & 0 & 0 & 0 \\ 0 & 0 & \mp 1 & 0 \\ 0 & \pm 1 & 0 & 0 \\ 0 & 0 & 0 & 1 \end{pmatrix} F_{11}I_L \begin{pmatrix} 1 \\ a \\ 0 \\ 0 \end{pmatrix} = \eta_S T_S \begin{pmatrix} 1 & D_S & 0 & 0 \\ D_S & 1 & 0 & 0 \\ 0 & 0 & Z_S c_S & Z_S s_S \\ 0 & 0 & -Z_S s_S & Z_S c_S \end{pmatrix} F_{11}I_L \begin{pmatrix} 1 \\ 0 \\ \pm a \\ 0 \end{pmatrix} =$$

$$= \eta_S T_S F_{11}I_L \begin{pmatrix} 1 \\ D_S \\ \pm a Z_S c_S \\ \mp a Z_S s_S \end{pmatrix} \tag{23}$$

With the rotations $\mathbf{R}(\pm 45°)$ it is intended to produce at the entrance of the PBS equal light
intensities in the transmitted and reflected paths, independent of the atmospheric
depolarisation. The error from an inaccurate $\pm 45°$ alignment can be reduced by the $\Delta 90$-
calibration explained in Sect. 5. From Eq. (23) we get the signal intensities
$I_S(\pm 45°) = \eta_S T_S F_{11}I_L$      (24)
and the calibration factor $\eta$ from the signal ratio
$\dfrac{I_R}{I_T}(\pm 45°) = \dfrac{\eta_R T_R}{\eta_T T_T} = \eta$      (25)
With known $\eta$ we can express the measured signal ratio $\delta^*$ in Eq. (21) as
$\delta^* \equiv \dfrac{1}{\eta}\dfrac{I_R}{I_T}(0°) = \dfrac{I_T}{I_R}(\pm 45°)\dfrac{I_R}{I_T}(0°) = \dfrac{T_T}{T_R}\dfrac{T_R^p + T_R^s \delta}{T_T^p + T_T^s \delta}$      (26)
which is almost equal to the linear depolarisation ratio $\delta$, but still includes the diattenuation
and cross talk of the imperfect polarising beam-splitter. From $\delta^*$ we retrieve the linear
depolarisation ratio $\delta$
$\delta = \dfrac{\delta^* T_R T_T^p - T_T T_R^p}{T_T T_R^s - \delta^* T_R T_T^s}$      (27)
With the assumption for good PBSs
$T_T^s \ll 1 \Rightarrow \left\{ T_R^s \approx 1, \ T_T \approx 0.5 T_T^p, \ T_R \approx 0.5\left(1 + T_R^p\right) \right\}$      (28)
we get an approximation
$\delta \approx \delta^* - T_R^p\left(1 - \delta^*\right)$      (29)





Next we will determine the total lidar backscatter signal from the two signals $I_T$ and $I_R$
measured at 0°. This is the range dependent signal, which we use for the inversion of the
backscatter coefficient $F_{11}$ with the lidar inversion methods. From Eq. (20) we can get $F_{11}$
either from the transmitted or from the reflected signal

$$F_{11} = \frac{I_S(0°)}{\eta_S T_S I_L (1 + D_S a)} \qquad (30)$$

The polarisation parameter $a$ can be extracted from the signal ratio in Eq. (21)

$$a = \frac{\eta I_T - I_R}{I_R D_T - \eta I_T D_R}, \qquad (31)$$

and substituted in Eq. (30) to yield

$$I_L F_{11} = \frac{\eta_T T_T D_T I_R - \eta_R T_R D_R I_T}{\eta_T T_T \eta_R T_R (D_T - D_R)} = \frac{1}{D_T - D_R} \left( \frac{D_T I_R}{\eta_R T_R} - \frac{D_R I_T}{\eta_T T_T} \right). \qquad (32)$$

Equation (32) shows that we cannot determine an absolute $F_{11}$ without an absolute calibration
of the individual channel gains $\eta_R$ and $\eta_T$ and knowledge of the laser intensity $I_L$. However, for
the lidar signal inversions, which use a reference value at a certain range or similar, we only
need a relative, range dependent $F_{11}$. Hence we can choose any of the range independent
parameters in Eq. (32), in which only $I_T$ and $I_R$ are range dependent, which we cancel and get

$$F_{11} \propto D_T I_R - \eta D_R I_T = \frac{T_T^p - T_T^s}{T_T^p + T_T^s} I_R - \eta \frac{T_R^p - T_R^s}{T_R^p + T_R^s} I_T. \qquad (33)$$

In case the polarising beam-splitter is ideal, i.e. $T_T^p = T_R^s = 1$ and $T_T^s = T_R^p = 0$, and hence $D_R =$
−1 and $D_T = +1$, Eq. (33) becomes as expected

$$F_{11} \propto I_R + \eta I_T, \qquad (34)$$

Please bear in mind that in general $T_R^s > T_R^p$, and therefore $(T_R^p - T_R^s) < 0$ and $D_R < 0$
according to our definition in Eq. (17).
Summarising: we have to find the calibration factor $\eta$ and correct the cross talk. $\delta$ is retrieved
from two signals at 0° represented by $\delta^*$, Eq. (26), plus two signals for the calibration factor at
±45°, Eq. (25), and the knowledge of the PBS parameters $T_T^p$, $T_T^s$, $T_R^p$, and $T_R^s$ for the
correction of the cross talk.





## 3 Complete Müller-Stokes lidar setup with rotation of optical elements

In the previous section, a basic lidar setup is described with the Müller-Stokes formalism as
an introduction, which includes only a horizontal-linear polarised laser, the matrices for the
atmospheric aerosol backscattering and depolarisation, and the polarising beam-splitter. In
order to expand this setup to a realistic but still manageable model for a large variety of lidar
systems and calibration techniques, we introduce in this section some concepts and variables,
which will enable us to describe the variety of setups with as few as possible equations.
The Stokes-Müller formalism (Chipman, 2009b) represents four linear equations (Eq. (35)),
which relate the four output with the four input Stokes parameters.

$$
\boldsymbol{I}_{out} = \begin{pmatrix} I_{out} \\ Q_{out} \\ U_{out} \\ V_{out} \end{pmatrix} = \mathbf{M}\boldsymbol{I}_{in} = \begin{pmatrix} M_{11} & M_{12} & M_{13} & M_{14} \\ M_{21} & M_{22} & M_{23} & M_{24} \\ M_{31} & M_{32} & M_{33} & M_{34} \\ M_{41} & M_{42} & M_{43} & M_{44} \end{pmatrix} \begin{pmatrix} I_{in} \\ Q_{in} \\ U_{in} \\ V_{in} \end{pmatrix} =
$$

$$
= M_{11}I_{in} \begin{pmatrix} 1 & m_{12} & m_{13} & m_{14} \\ m_{21} & m_{22} & m_{23} & m_{24} \\ m_{31} & m_{32} & m_{33} & m_{34} \\ m_{41} & m_{42} & m_{43} & m_{44} \end{pmatrix} \begin{pmatrix} i_{in} \\ q_{in} \\ u_{in} \\ v_{in} \end{pmatrix}
$$

(35)

The small letter matrix ($m_{ij}$) and vector components at the right of Eq. (35) are normalised by
their first element, i.e. $M_{11}$ and $I_{in}$ ; hence $m_{11} = i_{in} = 1$. However, in the following we usually
keep the variable $i_{in}$ in order to allow for later expansions of the equations. While the first
Stokes vector parameter $I_{out}$ can be directly detected with a photon detector, the other output
Stokes parameters can each be determined with two measurements of output intensities using
additional polarisation elements (Chipman, 2009a) (see Eq. S.2.2). We derive the backscatter
coefficient $F_{11}$ and the linear polarisation parameter $a$ of the Müller matrix $\mathbf{F}$ of the
atmosphere (see Sect. 2.1) from the first two equations of $I_{out}$ and $Q_{out}$ in Eq. (35), which in
turn are determined from the two measurements of $I_R$ and $I_T$ using the two orthogonal linear
analysers of the polarising beam-splitter. For the determination of each additional unknown
parameter we need additional measurements. For the relative calibration factor $\eta$ of the two
polarisation signals $I_R$ and $I_T$ we use an additional calibrator element with Müller matrix $\mathbf{C}$.
The lidar setup shown in Fig (1) is described by Eq. (6), i.e. $\boldsymbol{I}_S = \eta_S \mathbf{M}_S \mathbf{C} \mathbf{M}_O \mathbf{F} \mathbf{M}_E \boldsymbol{I}_L$, where
the matrices $\mathbf{M}_{T,R}$ (alias $\mathbf{M}_S$) represent the two paths of the polarising beam-splitter, i.e.
subscripts $T$ for transmission and $R$ for reflection. Since the laser in our model can be





arbitrarily polarised and because "parallel" and "perpendicular" are defined relative to the
incident plane of a beam-splitter (superscripts $p$ and $s$, respectively; see  S.1) and don't
necessarily describe the polarisation behind it with respect to the laser polarisation, we can't
use these terms here for the two branches behind the polarising beam-splitter. $\mathbf{C}(\Psi)$ describes
the calibrator matrix, which can be a mechanical rotation of the detection optics by $\Psi$ or an
optical device as a polarising sheet filter rotated by angle $\Psi$, for example. The purpose of the
calibrator device is to produce equal intensities for both polarisation channels, independent of
the laser light polarisation and independent of backscattering characteristics of the
atmosphere. This is e.g. achieved with an ideal polarising sheet filter oriented at 45° with
respect to the incident plane of the PBS. The calibration factor $\eta$ of the relative sensitivity of
both polarisation channels can be retrieved from the ratio of the measured intensities. The
calibration factor includes electronic gains and the polarisation transmission of optical
elements behind the calibrator. In our model the calibrator can be at three different positions
in the optical chain, which are indicated by the red blocks in Fig. (2). The calibrator positions
and the respective equations are these:
behind the laser emitter optics $\mathbf{M}_E$ $\qquad I_S = \eta_S \mathbf{M}_S \mathbf{M}_O \mathbf{F} \mathbf{C} \mathbf{M}_E I_L = \eta_S \mathbf{M}_S \mathbf{M}_O \mathbf{F} \mathbf{C} I_{in}$ (36)
before the telescope / receiver optics $\mathbf{M}_O$ $\quad I_S = \eta_S \mathbf{M}_S \mathbf{M}_O \mathbf{C} \mathbf{F} \mathbf{M}_E I_L = \eta_S \mathbf{M}_S \mathbf{M}_O \mathbf{C} I_{in}$ (37)
before the polarising beam-splitter $\mathbf{M}_S$ $\qquad I_S = \eta_S \mathbf{M}_S \mathbf{C} \mathbf{M}_O \mathbf{F} \mathbf{M}_E I_L = \eta_S \mathbf{M}_S \mathbf{C} I_{in}$ (38)
In case the telescope and/or the collimating lens don't change the state of polarisation of the
incoming light, the placement of the calibrator after those elements is equivalent to the
position before the telescope.
We develop the equations for all three positions of the calibrator, and additionally for the
calibration with an unpolarised light source before the receiving optics (Sect. 6). In the
equations we use as calibrator elements the Müller matrix $\mathbf{C}$ as a place holder for any sort of
calibrator, which are $\mathbf{M}_{rot}$ for mechanical rotation or by means of a λ/2 plate, $\mathbf{M}_P$ for a linear
polariser, $\mathbf{M}_{QW}$ for a λ/4 plate, and $\mathbf{M}_{CP}$ for a circular polariser.

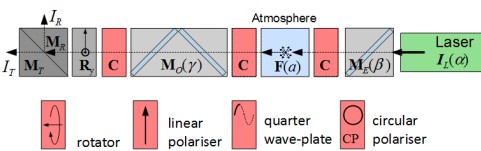



Figure 2: Schematic of a 2-channel, polarisation sensitive lidar setup (compare Fig. 1) with
Müller matrix block elements and different calibrator (red block) positions (top), and three
options for the calibrator $\mathbf{C}$ (bottom). $\boldsymbol{I}_L$: laser Stokes vector, $\mathbf{M}_E$: emitter optics; $\mathbf{F}$:
atmospheric backscatter matrix with polarisation parameter a; $\mathbf{M}_O$: receiver optics; $\mathbf{R}_y$:
rotation matrix for the 0° (y=+1) and 90° (y=-1)detection setup (see text); $\mathbf{M}_{T,R}$: transmitted
and reflected part of the polarising beam-splitter; $\boldsymbol{I}_{T,R}$: transmitted and reflected detection
signals. Angles α, β, and γ are rotations around the optical axis.
## 3.1 The analyser <bra| and input |ket> vectors
The general structure of all the considered lidar setups can be described with three groups of
optical elements: elements before the calibrator, the calibrator, and elements behind the
calibrator. To simplify the equations, we combine the matrices after the calibrator to an
analyser matrix $\mathbf{A}_S$, and the matrices before the calibrator together with the Stokes vector of
the laser beam $\boldsymbol{I}_L$ to an input Stokes vector $\boldsymbol{I}_{in}$. Since $\mathbf{A}_S$ and $\boldsymbol{I}_{in}$ are the same for all calibrator
types, they have to be derived only once and can then be used for the different setups. "After"
and "before" denote the order with respect to the light direction, i.e. from right to left in the
Müller-Stokes equations.
Since photo detectors are, in general, insensitive to the polarisation, we measure the intensity
$I_S$ at the detector, which is the first parameter of the output Stokes vector. $I_S$ is determined by
the top row of a matrix $\mathbf{A}_S$ and an input vector $\boldsymbol{I}_{in}$.

$$\begin{pmatrix} I_S \\ - \\ - \\ - \end{pmatrix} = \eta_S \mathbf{A}_S \boldsymbol{I}_{in} = \eta_S \begin{pmatrix} A_{11} & A_{12} & A_{13} & A_{14} \\ - & - & - & - \\ - & - & - & - \\ - & - & - & - \end{pmatrix} \begin{pmatrix} I_{in} \\ Q_{in} \\ U_{in} \\ V_{in} \end{pmatrix} = \eta_S \begin{pmatrix} A_{11}I_{in} + A_{12}Q_{in} + A_{13}U_{in} + A_{14}V_{in} \\ - \\ - \\ - \end{pmatrix} \quad (39)$$

Using the <bra|ket> matrix-vector notation (see App. B and App. D), we define for this work
the row vector $\langle\mathbf{A}_S|$ as the top row of a matrix $\mathbf{A}_S$,

$$\langle \mathbf{A}_S | = \langle A_{11} \quad A_{12} \quad A_{13} \quad A_{14} | \quad (40)$$

and use analogously the column vector $|\boldsymbol{I}_{in}\rangle$. With this notation the equation for the intensity
$I_S$ can be written as

$$I_S = \eta_S \langle \mathbf{A}_S | \boldsymbol{I}_{in} \rangle = \eta_S I_{in} \langle A_{11} \quad A_{12} \quad A_{13} \quad A_{14} | I_{in} \quad Q_{in} \quad U_{in} \quad V_{in} \rangle =$$
$$= \eta_S I_{in} \left( A_{11}I_{in} + A_{12}Q_{in} + A_{13}U_{in} + A_{14}V_{in} \right) \quad (41)$$




For example, the equation for signal $I_S$ of a calibration measurement with the calibrator before
the PBS (see Eq. (38)) can be expressed as
$$I_S\left(\text{y,x},\varepsilon\right)=\eta_S\left\langle\mathbf{M}_S\mathbf{R}_\text{y}\left|\mathbf{C}\left(\text{x}45°+\varepsilon\right)\right|\mathbf{M}_O\mathbf{F}\mathbf{M}_E\boldsymbol{I}_L\right\rangle=\eta_S\left\langle\mathbf{A}_S\left|\mathbf{C}\right|\boldsymbol{I}_{in}\right\rangle \qquad (42)$$
and the respective standard atmospheric measurement signals without the calibrator can be
expressed with the same vectors $<\mathbf{A}_S|$ and $|\boldsymbol{I}_{in}>$ as
$$I_S\left(\text{y}\right)=\eta_S\left\langle\mathbf{M}_S\mathbf{R}_\text{y}\left|\mathbf{M}_O\mathbf{F}\mathbf{M}_E\boldsymbol{I}_L\right\rangle=\eta_S\left\langle\mathbf{A}_S\left|\boldsymbol{I}_{in}\right\rangle\right. \qquad (43)$$
In Eqs. (42) and (43) we already used the binary operators y, x, and the variable $\varepsilon$ for different
rotation angles, and the rotation matrix $\mathbf{R}_y$, which will be explained in detail in Sect. 3.3.
**3.2  Laser polarisation and atmospheric depolarisation**
The light leaving commercial Nd:YAG lasers is usually linearly polarised. Manufacturers
often specify a polarisation "purity" > 95% or similar, which is not very accurate. Actually,
the laser light is often much better polarised, but the measurement of the polarisation of
individual lasers in a series is expensive and it can change during the operation and with
ageing of the laser. Probably for that reason the manufacturers seem to specify a lower limit
which they can assure under all circumstances. A secure method to ensure a high degree of
linear polarisation is to use a polariser as the last element at the laser output. Often the
orientation of the laser polarisation relative to the orientation of the polarising beam-splitter in
the receiving optics is not well known, first, because the state of polarisation of short laser
pulses with high power is difficult to measure accurately, and second, the state of polarisation
of the laser can change during the operation of the laser over periods with changing
environmental conditions. Hence we consider a possible rotation $\alpha$ of the plane of horizontal-
linear polarisation of the laser (laser rotation). Furthermore, beam expanders and especially
steering mirrors after the laser can degrade the degree of linear polarisation considerably
producing elliptical polarised light. Hence we start with an emitter Stokes vector with
arbitrary state of polarisation leaving the laser, which includes all effects of cleaning, shaping
and steering optics
$$\boldsymbol{I}_E=\mathbf{M}_E\boldsymbol{I}_L=T_E I_L\left|i_E\quad q_E\quad u_E\quad v_E\right\rangle \qquad (44)$$
We will develop all equations first for a general emitter beam polarisation as in Eq. (44), and
then as an explicit example for a linearly polarised laser with intensity $I_L$ and laser rotation $\alpha$



(see App. D) to elaborate the errors due to misalignments of the calibration and measurement
optics.
$$\boldsymbol{I}_L(\alpha) = I_L \left| 1 \quad c_{2\alpha} \quad s_{2\alpha} \quad 0 \right\rangle \tag{45}$$
Depolarisation of the laser (with linear polarisation parameter $a_L$), caused by volume or
surface scattering in or on optical elements, is hardly probable, and the scattered radiation
reaching the lidar telescope would be negligible. However, it is briefly treated in S.3. The
Stokes vector $\boldsymbol{I}_F$, which is reflected by the atmosphere with scattering matrix $\mathbf{F}(a)$ with linear
polarisation parameter $a$ from a generally polarised emitter $\boldsymbol{I}_E$, is (see S.3)
$$\frac{\boldsymbol{I}_F(a)}{F_{11}T_E I_L} = \frac{\mathbf{F}(a)\left| \mathbf{M}_E \boldsymbol{I}_L \right\rangle}{F_{11}T_E I_L} = \left| i_E \quad a q_E \quad -a u_E \quad (1-2a)v_E \right\rangle \tag{46}$$
## 3.3  Receiver optics and calibrator
In order to investigate the effect of misalignments of the optical elements on the final
measurement and the calibration results, i.e. the total signal and the linear depolarisation ratio,
we apply to each optical element in Eqs. (36) to (38) an additional rotation error about the
optical axis (see Fig. (2)). The reference coordinate system is in general defined by the
incident plane of the polarising beam-splitter (Fig. (3)), wherefore no rotation error is
considered in $\mathbf{M}_S$. Nevertheless, the polarising beam-splitter can be mechanically rotated by
90° in some existing lidar systems without changing the rest of the setup. We include this
additional fixed rotation by introducing the rotation matrix $\mathbf{R}_y$ with the polarising beam-
splitter orientation parameter y (Fig. (3)). For y = +1 the parallel laser polarisation is detected
in the transmitted channel and for y = −1 in the reflected channel. This seems a bit confusing,
but it is necessary to get control of all the actually existing lidar set-ups. The rotation matrix
$\mathbf{R}_y$ is shown in Eq. (47).
$$\mathbf{R}_y = \mathbf{R}(y) = \begin{pmatrix} 1 & 0 & 0 & 0 \\ 0 & y & 0 & 0 \\ 0 & 0 & y & 0 \\ 0 & 0 & 0 & 1 \end{pmatrix} \Rightarrow \begin{matrix} \mathbf{R}(y=-1) = \mathbf{R}(90°) \\ \mathbf{R}(y=+1) = \mathbf{R}(0°) \end{matrix} \tag{47}$$



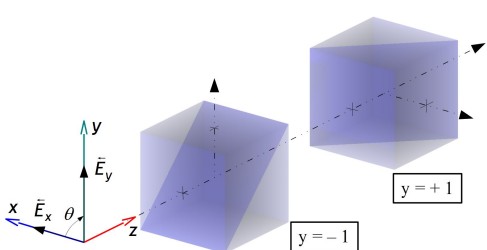

Figure 3: Definition of the global coordinate reference system and the binary operator y with respect to the
incident plane of the polarising beam-splitter. If the polarising beam-splitter orientation parameter y = +1, the
vibration of the horizontal-linear polarisation with vector $E_x$ is parallel to the plane of incidence, while for y = −
1 it is perpendicular.
The whole lidar system shown in Fig.(2) is then described by Eq. (48) with rotation angles α,
β, γ, and $\Psi$ around the optical axis.
$$I_S(y, \Psi, \gamma, a, \beta, \alpha) = \eta_S \mathbf{M}_S \mathbf{R}(y) \mathbf{C}(\Psi) \mathbf{M}_O(\gamma) \mathbf{F}(a) \mathbf{M}_E(\beta) I_L(\alpha)$$  (48)

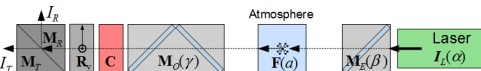

It would be possible to include the $\mathbf{R}_y$ rotation by changing the laser angle $\alpha$ in Eq. (48), but
we choose to do it before the polarising beam-splitter for two reasons: first we want to use the
angle $\alpha$ only for rotation errors, and second, in some lidar systems a rotation of the receiving
optics is used for the calibration, and with these setups a change between the two $\mathbf{R}_y$ versions
of a lidar is easily accomplished and can be used for certain test measurements without
changing the rest of the equations. On the other hand, an arbitrary rotation of the laser
polarisation is usually not possible. A rotation $\gamma$ of a retarding diattenuator $\mathbf{M}_O$ can complicate
the equations considerably, as it converts linearly polarised light into elliptically polarised,
which cannot be analysed by a simple polarising beam-splitter. Therefore, diattenuating and
retarding optics before the polarising beam-splitter should be carefully oriented with their
eigen axes parallel to the ones of the polarising beam-splitter to avoid the resulting
uncertainties. Such an element can e.g. be a dichroic beam-splitter, which does not reflect
exactly to 0° or 90°. For what we call Δ90-calibration, we use two calibrator orientations
$\mathbf{C}(\Psi)$ with
$$\begin{aligned} \Psi^+ &= +45° + \varepsilon \\ \Psi^- &= -45° + \varepsilon \end{aligned}$$  (49)





so that
$$\Psi^{+} - \Psi^{-} = 90° \qquad (50)$$
We choose these special angles, because in the geometric mean of two calibrations at
orientations exactly 90° apart the error terms sometimes compensate very well. Note, that the
$\Delta 90$ error angle $\varepsilon$ describes the rotational misalignment of the whole $\Delta 90$-calibrator setup
with respect to the polarising beam-splitter, not the error in the 90° difference. So, $\pm 45°$
means either $+45°$ or $-45°$, and $\Delta 90$ means the combination of measurements at $+45° + \varepsilon$ and
$-45° - \varepsilon$. To obtain general equations, we combine these angles using the binary operator x for
calibrations
$$x = \pm 1: \quad \Psi(x, \varepsilon) = x45° + \varepsilon \qquad (51)$$
We use this definition in a setup with a rotation calibrator $\mathbf{M}_{rot}$ (Sect. 7)
$$\mathbf{C}(\Psi, h) = \mathbf{M}_{rot}(x45° + \varepsilon, h) = \mathbf{M}_{rot}(x, \varepsilon, h) \qquad (52)$$
with the binary operator h to discern between a mechanical (h = +1) and a $\lambda/2$ plate rotation
(S.10.15), and can express the four equations for the reflected and transmitted signals $I_R$ and $I_T$
of the two calibration measurements at $\Psi = \pm 45° + \varepsilon$ with the one formula Eq. (53)
$$I_S(y, x, \varepsilon, h, \gamma, a, \beta, \alpha) = \eta_S \left\langle \mathbf{M}_S \mathbf{R}_y \middle| \mathbf{M}_{rot}(x, \varepsilon, h) \middle| \mathbf{M}_O(\gamma) \mathbf{F}(a) \mathbf{M}_E(\beta) \boldsymbol{I}_L(\alpha) \right\rangle \qquad (53)$$
and the four equations for the standard measurements at $\Psi = 0°$ (y = +1) and $\Psi = 90°$ (y = −1)
using the same analyser and input Stokes vectors with just another formula Eq. (54)
$$I_S(y, \varepsilon, h, \gamma, a, \beta, \alpha) = \eta_S \left\langle \mathbf{M}_S \mathbf{R}_y \middle| \mathbf{R}(\varepsilon) \mathbf{M}_h \middle| \mathbf{M}_O(\gamma) \mathbf{F}(a) \mathbf{M}_E(\beta) \boldsymbol{I}_L(\alpha) \right\rangle \qquad (54)$$
Using the rotation calibrator we have to consider the same alignment error $\varepsilon$ for the standard
measurements at 0° and 90° as for the calibration at the $\pm 45°$, because this calibrator is not
removed from the lidar setup after the calibration measurements. Hence we have to differ, if
necessary, between $\varepsilon$ for the standard measurements and $\varepsilon_{P,QW,CP}$, with $P$ for the polariser, $QW$
for the $\lambda/4$ plate, and $CP$ for the circular polariser. Please note that $\varepsilon = 0$ for all other
calibrators.
**4   Retrieval of the total signal and of the linear depolarisation ratio**
The final goal of this work is to investigate how the polarisation calibration factor, the linear
depolarisation ratio, and the total lidar signal can be retrieved from the measurements $I_T$ and



$I_R$, how much the various rotational misalignments and the crosstalk of the calibrator influence
them, and how the deviations can possibly be corrected. The   standard atmospheric
measurement signals $I_S$ in Eq. (54) include a rotational error $\varepsilon$ before the polarising beam-
splitter.

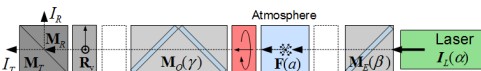

We get Eq. (55) for the analyser part with Eqs. (D.5), (S.5.1.6), and (S.10.15.2), and with the
most general input $\boldsymbol{I}_E$ from Eq. (E.31) with atmospheric polarisation parameter $a$ we get the
signal $I_S$ from Eq. (S.7.1.2)

$$\langle \mathbf{A}_S(y)|\mathbf{R}(\varepsilon,h) = \langle \mathbf{M}_S\mathbf{R}_y|\mathbf{R}(\varepsilon)\mathbf{M}_h = T_S\langle 1 \quad yc_{2\varepsilon}D_S \quad -yhs_{2\varepsilon}D_S \quad 0| \tag{55}$$

$$\frac{I_S}{\eta_S T_S T_{rot} T_O F_{11} T_E I_L} = \frac{\langle \mathbf{A}_S(y)|\mathbf{R}(\varepsilon)\mathbf{M}_h|\boldsymbol{I}_{in}(\gamma,a)\rangle}{T_S T_{rot} T_O F_{11} T_E I_L} = \frac{\langle \mathbf{M}_S\mathbf{R}_y|\mathbf{R}(\varepsilon)\mathbf{M}_h|\mathbf{M}_O(\gamma)\mathbf{F}(a)\boldsymbol{I}_E\rangle}{T_S T_{rot} T_O F_{11} T_E I_L} =$$
$$= \left(1 + yD_S D_O c_{2\gamma+h2\varepsilon}\right)i_E - yD_S Z_O s_O s_{2\gamma+h2\varepsilon}v_E +$$
$$+ a\left\{D_O\left(c_{2\gamma}q_E - s_{2\gamma}u_E\right) + yD_S\left[\left(c_{2\varepsilon}q_E + s_{h2\varepsilon}u_E\right) - s_{2\gamma+h2\varepsilon}\left(W_O\left(s_{2\gamma}q_E + c_{2\gamma}u_E\right) - 2Z_O s_O v_E\right)\right]\right\}$$

$\hfill$ (56)

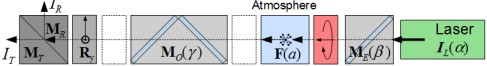

In case the rotational error is before the receiving optics, the equation becomes more complex.
With Eq. (D.7) for the analyser part and (E.26) for the input vector we get from Eq. (S.7.2.1)

$$\frac{I_S}{\eta_S T_S T_O T_{rot} F_{11} T_E I_L} = \frac{\langle \mathbf{A}_S(y,\gamma)||\boldsymbol{I}_{in,\varepsilon}(\varepsilon,h,a)\rangle}{T_S T_O T_{rot} F_{11} T_E I_L} = \frac{\langle \mathbf{M}_S\mathbf{R}_y\mathbf{M}_O(\gamma)||\mathbf{R}(\varepsilon)\mathbf{M}_h\mathbf{F}(a)\boldsymbol{I}_E\rangle}{T_S T_O T_{rot} F_{11} T_E I_L} =$$
$$= \left(1 + yD_O D_S c_{2\gamma}\right)i_E - yD_S Z_O s_O s_{2\gamma}hv_E +$$
$$+ a\left\{\begin{array}{l} D_O\left[c_{2\gamma-2\varepsilon}q_E - s_{2\gamma-2\varepsilon}hu_E\right] - yD_S W_O s_{2\gamma}\left[s_{2\gamma-2\varepsilon}q_E + c_{2\gamma-2\varepsilon}hu_E\right] + \\ + yD_S\left[q_E c_{2\varepsilon} + hu_E s_{2\varepsilon} + 2s_{2\gamma}Z_O s_O hv_E\right] \end{array}\right\} \tag{57}$$





The case of rotational error behind the emitter optics can be retrieved from Eq. (57) by simply
replacing $\varepsilon$ with $-\varepsilon$ according to S.7.3. Special cases of $\boldsymbol{I}_E$ for Eqs. (56) and (57) can be found
in Sect. E.2.
**4.1   General formulations for the total signal and the linear depolarisation ratio**
From Eqs. (56) and (57) we see that all standard signals $I_S$ can be expressed by introducing
two variables $G_S$ and $H_S$ for the terms without and with atmospheric polarisation, respectively,
$$I_S = \eta_S T_S T_O T_{rot} F_{11} T_E I_L \left( G_S + a H_S \right) \tag{58}$$
Using Eq. (56) as an example, the two variables are
$$G_S\left(\mathrm{y},\varepsilon,\mathrm{h},\gamma\right) = \left(1 + \mathrm{y}D_S D_O \mathrm{c}_{2\gamma+\mathrm{h}2\varepsilon}\right)i_E - \mathrm{y}D_S Z_O \mathrm{s}_O \mathrm{s}_{2\gamma+\mathrm{h}2\varepsilon}v_E$$
$$H_S\left(\mathrm{y},\varepsilon,\mathrm{h},\gamma,\beta,\alpha\right) =$$
$$= D_O\left(\mathrm{c}_{2\gamma}q_E - \mathrm{s}_{2\gamma}u_E\right) + \mathrm{y}D_S\left[\left(\mathrm{c}_{2\varepsilon}q_E + \mathrm{s}_{\mathrm{h}2\varepsilon}u_E\right) - \mathrm{s}_{2\gamma+\mathrm{h}2\varepsilon}\left(W_O\left(\mathrm{s}_{2\gamma}q_E + \mathrm{c}_{2\gamma}u_E\right) - 2Z_O \mathrm{s}_O v_E\right)\right] \tag{59}$$

With Eq. (58) the measured signal ratio becomes
$$\delta^* = \frac{1}{\eta}\frac{I_R}{I_T} = \frac{G_R + a H_R}{G_T + a H_T} \tag{60}$$
with the calibration factor $\eta = \dfrac{\eta_R T_R}{\eta_T T_T}$, which has to be determined with one of the methods in
the following chapters. $G_S$ and $H_S$ describe the polarisation cross-talk terms of the lidar setup
depending on the diattenuation parameters $D$ and the retardation (described by $\mathrm{s}_O$ and $\mathrm{c}_O$) of
the individual optical elements, depending on the relative rotation of the elements and on the
polarisation parameter of the atmosphere $a$. From Eq. (60) we retrieve the general equations
for the polarisation parameter $a$ in Eq. (61) and for the linear depolarisation ratio $\delta$ in Eq. (62)
(compare Eq. (12)).
$$a = \frac{\delta^* G_T - G_R}{H_R - \delta^* H_T} \tag{61}$$
$$\delta = \frac{1-a}{1+a} = \frac{\delta^*\left(G_T + H_T\right) - \left(G_R + H_R\right)}{\left(G_R - H_R\right) - \delta^*\left(G_T - H_T\right)} \tag{62}$$
Remind that $\delta^*$ and hence $a$ and $\delta$ are range dependent. For the retrieval of the total lidar
signal, which is equivalent to $F_{11}$, we substitute Eq. (61) in Eq. (58) in the transmitted or the




reflected version of $I_S \in \{I_T, I_R\}$ and replace $\delta^*$ by Eq. (60). Using the transmitted signal $I_T$
from Eq. (58) we get Eq. (63), and after some restructuring (see Eqs.(S.8.1) and (S.8.2) ) we
get the attenuated backscatter coefficient Eq. (64).
$$\eta_T T_T T_O F_{11} T_E I_L = \frac{I_T}{G_T + aH_T} \tag{63}$$
$$F_{11} = \frac{1}{T_O T_E I_L} \frac{H_R \dfrac{I_T}{\eta_T T_T} - H_T \dfrac{I_R}{\eta_R T_R}}{H_R G_T - H_T G_R} \tag{64}$$
For the inversion of the lidar signal we only need the relative attenuated backscatter
coefficient, for which we can get a much simpler formula by removing all factors in Eq. (64)
which are not range dependent (compare Eq. 32 ff), which yields Eq.(65)
$$F_{11} \propto \eta H_R I_T - H_T I_R \tag{65}$$
The individual calibration methods can add errors and uncertainties due to additional optics
with unknown diattenuation and retardation and due to rotation errors. The possible
uncertainties of the calibration factor $\eta$ can be assessed from the analytical expressions of the
gain ratio $\eta^*$ (see Sect.(5)).
For systems without a polarising beam-splitter, i.e. pure backscatter lidars with one channel
for each wavelength, the total signal is $I_T$ from the transmitted signal, but with $D_S = D_T = 0$,
and without calibrator (=> h = 1) and without calibrator rotation error angle ε. Hence, we get
from both Eqs. (56) and (57) the transmitted signal with Eq. (66)
$$D_T = 0, T_T = 1, \varepsilon = 0, y = 1 \Rightarrow$$
$$I_T = \eta_T T_T T_O F_{11} T_E I_L \left[ i_E + a D_O \left( c_{2\gamma} q_E - s_{2\gamma} u_E \right) \right] \tag{66}$$

which shows that there is a distortion of the total signal due to the receiver optics
diattenuation and depending on the atmospheric depolarisation, even if the laser beam behind
the emitter optics is perfectly horizontal-linearly polarised and without receiver optics
rotation. i.e. Eq. (66) with
$$\gamma = 0, T_E = 1, i_E = q_E = 1, u_E = 0 \Rightarrow$$
$$I_T = \eta_T T_O F_{11} I_L \left[ 1 + a D_O \right] \tag{67}$$



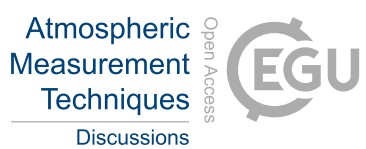

## 4.2 Simplifications for standard measurements

For the case of Eq. (56), i.e. rotational error $\varepsilon$ before the polarising beam-splitter, and with a general emitter Stokes vector $I_E = T_E I_L |i_E \ \ q_E \ \ u_E \ \ v_E>$ (see Sect. App. E.2) we get from Eq. (59) the variables in Eq. (58), i.e. $I_S = \eta_S T_S T_O T_{rot} F_{11} T_E I_L (G_S + aH_S)$ :

| | $G_S$ | $H_S$ | |
|---|---|---|---|
| General | $(1 + yD_S D_O c_{2\gamma+h2\varepsilon})i_E - yD_S Z_O s_O s_{2\gamma+h2\varepsilon} v_E$ | $D_O(c_{2\gamma}q_E - s_{2\gamma}u_E) + yD_S\left[(c_{2\varepsilon}q_E + s_{h2\varepsilon}u_E) - s_{2\gamma+h2\varepsilon}\left\{W_O(s_{2\gamma}q_E + c_{2\gamma}u_E) - 2Z_O s_O v_E\right\}\right]$ | (68) |
| $\varepsilon = 0$ | $(1 + yD_S D_O c_{2\gamma})i_E - yD_S Z_O s_O s_{2\gamma} v_E$ | $D_O(c_{2\gamma}q_E - s_{2\gamma}u_E) + yD_S\left[q_E - s_{2\gamma}\left\{W_O(s_{2\gamma}q_E + c_{2\gamma}u_E) - 2Z_O s_O v_E\right\}\right]$ | (69) |
| $\gamma = 0$ | $(1 + yD_S D_O c_{h2\varepsilon})i_E - yD_S Z_O s_O s_{h2\varepsilon} v_E$ | $D_O q_E + yD_S\left[c_{2\varepsilon}q_E + s_{h2\varepsilon}Z_O(c_O u_E + 2s_O v_E)\right]$ | (70) |
| $\gamma = \varepsilon = 0$ | $(1 + yD_S D_O)i_E$ | $(D_O + yD_S)q_E$ | (71) |
| $D_O = W_O = s_O = 0$ | $1$ | $yD_S(c_{2\varepsilon}q_E + s_{h2\varepsilon}u_E)$ | (72) |

The same as above, but with a rotated, linearly polarised emitter Stokes vector $I_E = |i_E \ \ q_E \ \ u_E \ \ v_E\rangle = T_E I_L = T_E I_L |1 \ \ c_{2\alpha} \ \ s_{2\alpha} \ \ 0\rangle \Rightarrow$

| | $G_S$ | $H_S$ | |
|---|---|---|---|
| General | $1 + yD_S D_O c_{2\gamma+h2\varepsilon}$ | $D_O c_{2\alpha+2\gamma} + yD_S(c_{2\alpha-2\varepsilon} - s_{2\gamma+h2\varepsilon}s_{2\alpha+2\gamma}W_O)$ | (73) |
| $\alpha = \varepsilon = 0$ | $1 + yD_S D_O c_{2\gamma}$ | $D_O c_{2\gamma} + yD_S(1 - s_{2\gamma}^2 W_O)$ | (74) |
| $\gamma = 0$ | $1 + yD_S D_O c_{h2\varepsilon}$ | $D_O c_{2\alpha} + yD_S(c_{2\alpha-2\varepsilon} - s_{2\alpha}s_{h2\varepsilon}W_O)$ | (75) |
| $\alpha = \gamma = 0$ | $1 + yD_S D_O c_{h2\varepsilon}$ | $D_O + yD_S c_{2\varepsilon}$ | (76) |
| $\gamma = \varepsilon = 0$ | $1 + yD_S D_O$ | $(D_O + yD_S)c_{2\alpha}$ | (77) |
| $\alpha = \gamma = \varepsilon = 0$ | $1 + yD_S D_O$ | $D_O + yD_S$ | (78) |
| $D_O = W_O = 0$ | $1$ | $+yD_S c_{2\alpha-2\varepsilon}$ | (79) |





**5    The 45° and Δ90 calibration, the gain ratios and calibration factor**
The measured, apparent calibration factor $\eta^*$ of the polarisation channels, which we call in the
following gain ratio in contrast to the calibration factor $\eta$, can be determined from the two
calibration signals $I_S$, i.e. $I_T$ and $I_R$, with a calibrator at $+45°$ or $-45°$, which we call 45°-
calibration (Eq. (80)). The calibration factor $\eta$ is not directly measurable. Hence we need
equations to retrieve $\eta$ from the measured $\eta^*$.

$$\left.\begin{array}{l}\eta^*\left(+45°\right)=\dfrac{I_R\left(+45°\right)}{I_T\left(+45°\right)}\\[2mm]\eta^*\left(-45°\right)=\dfrac{I_R\left(-45°\right)}{I_T\left(-45°\right)}\end{array}\right\}\rightarrow\eta^*=\dfrac{I_R}{I_T}\left(\text{x}45°\right)\tag{80}$$

$\eta^*$ includes alignment errors and cross talks. The theoretical dependence of these errors and
cross-talks on the known parameters of our lidar model (Fig. 1) can be determined using the
analytical expressions of Eqs. (81) and (82).
$I_S\left(\text{y,x}45°+\varepsilon\right)=\eta_S\left\langle\mathbf{A}_S\left(\text{y}\right)\middle|\mathbf{C}\left(\text{x}45°+\varepsilon\right)\middle|\boldsymbol{I}_{in}\right\rangle$ \hfill (81)
$\eta^*=\dfrac{I_R\left(\text{y,x}45°+\varepsilon\right)}{I_T\left(\text{y,x}45°+\varepsilon\right)}=\dfrac{\eta_R\left\langle\mathbf{A}_R\left(\text{y}\right)\middle|\mathbf{C}\left(\text{x}45°+\varepsilon\right)\middle|\boldsymbol{I}_{in}\right\rangle}{\eta_T\left\langle\mathbf{A}_T\left(\text{y}\right)\middle|\mathbf{C}\left(\text{x}45°+\varepsilon\right)\middle|\boldsymbol{I}_{in}\right\rangle}$ \hfill (82)
The theoretical correction $K$ of the gain ratio to get the calibrator factor can be retrieved from
the analytical expression Eq. (83), which is then used to correct the measurement Eq. (84).
$K=\dfrac{\eta^*}{\eta}=\eta^*\dfrac{\eta_T T_T}{\eta_R T_R}=\dfrac{T_T}{T_R}\dfrac{\left\langle\mathbf{A}_R\left(\text{y}\right)\middle|\mathbf{C}\left(\text{x}45°+\varepsilon\right)\middle|\boldsymbol{I}_{in}\right\rangle}{\left\langle\mathbf{A}_T\left(\text{y}\right)\middle|\mathbf{C}\left(\text{x}45°+\varepsilon\right)\middle|\boldsymbol{I}_{in}\right\rangle}$ \hfill (83)
$\eta=\dfrac{1}{K}\eta^*=\dfrac{1}{K}\dfrac{I_R}{I_T}\left(\text{x}45°\right)$ \hfill (84)
Furthermore, additional equations for the estimation of the uncertainty of $\eta$ can be derived
from Eq. (83). Since the errors due to $\varepsilon$ cancel very well at orientations of the calibrator
exactly Δ90 apart (i.e. $x = \pm 1$), as we will see in the following sections, a better estimation of
the gain ratio can be retrieved from the geometric mean of the two gain ratios at $\pm45°$, which
we call Δ90-calibration.



$$\eta^*_{\Delta 90} \equiv \sqrt{\eta^*\left(+45° + \varepsilon\right)\eta^*\left(-45° + \varepsilon\right)} = \sqrt{\frac{I_R\left(+45° + \varepsilon\right)}{I_T\left(+45° + \varepsilon\right)} \cdot \frac{I_R\left(-45° + \varepsilon\right)}{I_T\left(-45° + \varepsilon\right)}} \tag{85}$$
While the two calibration signals $I_T$ and $I_R$ are taken at the same time, the two measurements
for the $\Delta 90$-calibration at $x45° + \varepsilon$ are done subsequently, and the atmosphere can change in
between. If the gain ratio $\eta^*$ in Eq. (82) depends on the atmospheric polarisation parameter $a$,
the $\Delta 90$ gain ratio $\eta^*_{\Delta 90}$ in Eq. (85) depends also on the temporal change of $a$. In order to
avoid this dependency, we either have to choose an appropriate setup and adjust it so that $\eta^*$
doesn't depend on $a$, or we have to choose a calibration range in which $a$ doesn't change with
time. In the following we assume the latter, i.e. that the atmospheric polarisation parameter $a$
does not change in the calibration range between the two calibration measurements at $x45° + \varepsilon$.
This does not mean that the backscatter coefficient, an extrinsic parameter, must not change,
but only that the aerosol composition with its intrinsic parameter $a$ stays the same and that the
contribution of the air molecules to $a$ is negligible. Nevertheless, in Sect. 11 we describe a
method to determine and consequently correct for $\varepsilon$, which is one of the major factors in the
$a$-dependency of $\eta^*$. By the way, the method of 90° different polariser angles to reduce errors
in polarimetric measurements seems to be common in ellipsometry (Nee, 2006).
In the following sections we derive $\mathbf{A}_S$ and $\boldsymbol{I}_{in}$ for several positions of the calibrator $\mathbf{C}$, and
with that we will analyse special cases of the measurements $I_S$ and the retrieved calibration
factor $\eta$. The most general equation Eq. (86) for our lidar model, with e.g. a calibrator before
the PBS, contains eight optical parameters of the four optical elements and the atmosphere,
and four variables, i.e. the rotation angles of the optical elements and of the laser polarisation.
Note, because detectors only detect the flux of light, the retardation of the polarising beam-
splitter $\Delta_S$ is irrelevant. For each setup we firstly derive the general formulations (Eq. (86)).
Then, in order to reduce the complexity of the equations and to carve out the most important
and useful relations, we neglect certain parameters and variables in the detailed equations of
special cases. We often omit the explicit description of the laser emitter optics $\mathbf{M}_E$ (Eq. (87)),
which means that we assume the light emitted to the atmosphere as arbitrarily polarised (see
App. E.2) $\boldsymbol{I}_E = \mathbf{M}_E \boldsymbol{I}_L = T_E I_L \left| i_E \quad q_E \quad u_E \quad v_E \right\rangle$. If necessary $\boldsymbol{I}_E$ can be expanded in the final
equations by the appropriate ones in App. E. But we also consider the more simple case of a
rotated linearly polarised laser $\boldsymbol{I}_E = \boldsymbol{I}_L = I_L \left| 1 \quad c_{2\alpha} \quad s_{2\alpha} \quad 0 \right\rangle$. Furthermore, it is quite easy to
remove the cross talk of the polarising beam-splitter $\mathbf{M}_S$ by means of additional polarisation
filters behind it, which removes many terms in the equation (Eq. (88)). We call such an
analyser "cleaned". The rotation $\gamma$ of the receiving optics $\mathbf{M}_O$ is very disturbing, which can be
avoided in the very beginning of the lidar design (Eq. (89)). And at last, this paper provides
the tools to determine how good a calibrator must be to be considered as ideal. With such a
calibrator the equations become less complex (Eq.(90)).
$$\boldsymbol{I}_S = \eta_S \mathbf{M}_S(D_S)\mathbf{R}_y \mathbf{C}(D_C, \Delta_C, \varepsilon)\mathbf{M}_O(D_O, \Delta_O, \gamma)\mathbf{F}(a)\mathbf{M}_E(D_E, \Delta_E, \beta)\boldsymbol{I}_L(\alpha) \qquad (86)$$
$$\boldsymbol{I}_S = \eta_S \mathbf{M}_S(D_S)\mathbf{R}_y \mathbf{C}(D_C, \Delta_C, \varepsilon)\mathbf{M}_O(D_O, \Delta_O, \gamma)\mathbf{F}(a) \qquad \boldsymbol{I}_E \qquad (87)$$
$$\boldsymbol{I}_S = \eta_S \; \mathbf{M}_{S\,clean} \; \mathbf{R}_y \mathbf{C}(D_C, \Delta_C, \varepsilon)\mathbf{M}_O(D_O, \Delta_O, \gamma)\mathbf{F}(a) \qquad \boldsymbol{I}_E \qquad (88)$$
$$\boldsymbol{I}_S = \eta_S \mathbf{M}_S(D_S)\mathbf{R}_y \mathbf{C}(D_C, \Delta_C, \varepsilon)\mathbf{M}_O(D_O, \Delta_O, 0)\mathbf{F}(a) \qquad \boldsymbol{I}_E \qquad (89)$$
$$\boldsymbol{I}_S = \eta_S \mathbf{M}_S(D_S)\mathbf{R}_y \qquad \mathbf{C}_{ideal} \qquad \mathbf{M}_O(D_O, \Delta_O, \gamma)\mathbf{F}(a) \qquad \boldsymbol{I}_E \qquad (90)$$
**6   Calibration with unpolarised input before the receiving optics**

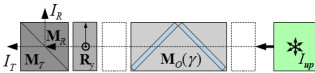

In principle, an additional light source with a known state of polarisation, which is placed
before the telescope, can be used for the calibration. For other states of polarisation of the
calibration light source the equations in Sect. (7.2) can be used together with the appropriate
description of the input Stokes vector. But the beam from an additional light source has some
disadvantages, because it fills the apertures of the individual optical elements differently than
the backscattered light from the lidar laser, and also the distribution of the incident angles on
elements with limited acceptance angles, as dichroic beams splitters and interference filters, is
different. Furthermore, the wavelength band of the light source is usually different from that
of the lidar laser, which introduces wavelength dependent transmission, diattenuation, and
retardation effects. This can lead to errors in the calibration factor, which can additionally be
range dependent. Such errors are very difficult to assess. We therefore prefer to use the
atmospheric backscatter of the lidar laser for the calibration, which provides the same spatial
and angular characteristics and the same wavelengths for the calibration as for the
measurements. Nevertheless, the output Stokes vector $\boldsymbol{I}_S$ of an unpolarised light source before
the receiving optics is given by Eq. (91).



$$I_S = \eta_S \mathbf{M}_S \mathbf{R}_y \mathbf{M}_O I_{up} \Rightarrow \mathbf{A}_S = \mathbf{M}_S \mathbf{R}_y \mathbf{M}_O \quad \text{and} \quad I_{in} = I_{up}$$ (91)
With the analyser vector from Eq. (D.7) and the unpolarised input Stokes vector $I_{in}$ before the
lidar optics from Eq. (92) we get the calibration signals in Eq. (93).
$$I_{in} = I_{up} = I_{up} \left| 1 \quad 0 \quad 0 \quad 0 \right\rangle$$ (92)
$$I_S = \eta_S \left\langle \mathbf{M}_S \mathbf{R}_y \mathbf{M}_O (\gamma) \right| I_{up} = \eta_S T_S T_O I_{up} \left( 1 + y D_S c_{2\gamma} D_O \right)$$ (93)
The gain ratio can be retrieved directly with Eq. (93)
$$\eta^* = \frac{I_R}{I_T} = \frac{\eta_R T_R}{\eta_T T_T} \frac{1 + y D_R D_O c_{2\gamma}}{1 + y D_T D_O c_{2\gamma}} = \eta \frac{1 + y D_R D_O c_{2\gamma}}{1 + y D_T D_O c_{2\gamma}}$$ (94)
Error sources are the unknown receiver optics rotation $\gamma$ and the diattenuation $D_O$. With a
cleaned analyser $\mathbf{M}_S$ (see S.10.10) and $\gamma = 0$ we get from Eq. (94)
$$\frac{\eta^*}{\eta} = \frac{1 - y D_O}{1 + y D_O}$$ (95)
With $D_O = \dfrac{T_O^p - T_O^s}{T_O^p + T_O^s}$ we get the gain ratios for the two setups y = ±1 from
$$\frac{\eta^*(y = +1)}{\eta} = \frac{T_O^s}{T_O^p}, \quad \frac{\eta^*(y = -1)}{\eta} = \frac{T_O^p}{T_O^s}$$ (96)
As there are no calibrator induced rotational errors $\varepsilon$, all equations for the standard
measurements of Sect. 4 are with $\varepsilon = 0°$.

## 7 Calibration with a rotator - mechanical or by λ/2 plate (HWP)

With an ideal HWP rotator the input Stokes vector is rotated with respect to the coordinate
system, while with the mechanical rotator the polarising beam-splitter and, if so, the receiving
optics are rotated in the opposite direction to achieve the same effect. Mathematically the
latter means a rotation of the coordinate system (see S.5). Furthermore, the rotation with a
HWP includes a retardance of 180° and hence a mirroring of the input Stokes vector (see
S.10.13, Eq. (S.10.13.2)). We combine the two methods in the rotator matrix $\mathbf{M}_{rot}$ (S.10.15) by
introducing the rotator operator h (Eq. (S.10.15.1)), which is h = +1 for the mechanical rotator
and h = −1 for the HWP rotator.





## 7.1   Calibration with a rotator before the polarising beam-splitter

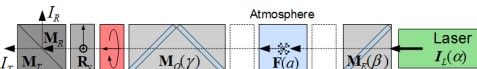

The general formula for the output Stokes vector $I_S$ with a rotation calibrator $\mathbf{M}_{rot}$ before the
polarising beam-splitter is Eq. (97).

$$I_S = \eta_S \mathbf{M}_S \mathbf{R}_y \mathbf{M}_{rot}(x45° + \varepsilon, h) \mathbf{M}_O(\gamma) \mathbf{F}(a) \mathbf{M}_E(\beta) I_L(\alpha) =$$
$$= \eta_S \mathbf{A}_S(y) \mathbf{M}_{rot}(x45° + \varepsilon, h) I_{in}(\gamma, a, \beta, \alpha)$$
(97)

With the analyser part $\mathbf{A}_S$ from Eq. (D.5), $\mathbf{M}_{rot}$ from Eq. (S.10.15.1), and the input Stokes
vector $I_{in}$ from App. E.4 we get Eq. (98) for the calibration signals, and with the expanded
input Stokes vector Eq. (E.31) we get from Eq. (98) the general calibration signals Eq. (99).

$$\frac{I_S}{\eta_S T_S T_{rot} I_{in}} = \frac{\langle \mathbf{M}_S \mathbf{R}_y | \mathbf{M}_{rot}(x45° + \varepsilon, h) | I_{in}\rangle}{T_S T_{rot} I_{in}} =$$
$$= \left\langle \begin{matrix} 1 \\ yD_S \\ 0 \\ 0 \end{matrix} \left| \begin{matrix} 1 & 0 & 0 & 0 \\ 0 & -xs_{2\varepsilon} & -xhc_{2\varepsilon} & 0 \\ 0 & xc_{2\varepsilon} & -xhs_{2\varepsilon} & 0 \\ 0 & 0 & 0 & h \end{matrix} \right| \begin{matrix} i_{in} \\ q_{in} \\ u_{in} \\ v_{in} \end{matrix}\right\rangle = \left\langle \begin{matrix} 1 \\ -xys_{2\varepsilon}D_S \\ -xyhc_{2\varepsilon}D_S \\ 0 \end{matrix} \left\| \begin{matrix} i_{in} \\ q_{in} \\ u_{in} \\ v_{in} \end{matrix} \right\rangle =$$
$$= i_{in} - xyD_S(s_{2\varepsilon}q_{in} + hc_{2\varepsilon}u_{in})$$
(98)

$$\frac{I_S}{\eta_S T_S T_{rot} T_O F_{11} T_E I_L} = \frac{\langle \mathbf{M}_S \mathbf{R}_y | \mathbf{M}_{rot}(x45° + \varepsilon, h) | \mathbf{F}(a) I_E\rangle}{T_S T_{rot} T_O F_{11} T_E I_L} =$$
$$= i_E + aD_O(c_{2\gamma}q_E - s_{2\gamma}u_E) - xyD_S \left\{ \begin{matrix} s_{2\varepsilon+h2\gamma}D_O i_E + a(s_{2\varepsilon}q_E - hc_{2\varepsilon}u_E) + \\ +hc_{2\varepsilon+h2\gamma}\left[W_O a(s_{2\gamma}q_E + c_{2\gamma}u_E) + Z_O s_O(1-2a)v_E\right] \end{matrix} \right\}$$
(99)

Since $i_{in}$ in Eq. (98) is independent of $\varepsilon$, x, and y, we can define the function $E$ in Eq. (100)
and get for the calibration signals Eq. (101) and for the gain ratios $\eta^*$ (Sect. 5) Eq. (102).

$$E(\varepsilon, h, \gamma, a, \beta, \alpha) \equiv \frac{s_{2\varepsilon}q_{in} + hc_{2\varepsilon}u_{in}}{i_{in}} =$$
$$= \frac{s_{2\varepsilon+h2\gamma}D_O i_E + a(s_{2\varepsilon}q_E - hc_{2\varepsilon}u_E) + hc_{2\varepsilon+h2\gamma}\left[W_O a(s_{2\gamma}q_E + c_{2\gamma}u_E) + Z_O s_O(1-2a)v_E\right]}{i_E + aD_O(c_{2\gamma}q_E - s_{2\gamma}u_E)}$$
\#(100)

$$I_S = \eta_S T_S T_{rot} I_{in}[1 - xyD_S E]i_{in}$$
(101)





$$\eta^* = \frac{I_R}{I_T} = \frac{\eta_R T_R}{\eta_T T_T} \frac{1 - \mathrm{xy} D_R E}{1 - \mathrm{xy} D_T E} = \eta \frac{1 - \mathrm{xy} D_R E}{1 - \mathrm{xy} D_T E} \qquad (102)$$
Eq. (103) shows the gain ratio from the Δ90-calibration, assuming that the polarisation
parameter $a$ doesn't change in the calibration range between the two calibration
measurements, i.e. $E_+ = E_-$ (see Sect. 5).
$$\frac{\eta^*_{\Delta 90}}{\eta} = \sqrt{\frac{1 - \mathrm{y} D_R E_+}{1 - \mathrm{y} D_T E_+} \frac{1 + \mathrm{y} D_R E_-}{1 + \mathrm{y} D_T E_-}} = \sqrt{\frac{1 - D_R^{\,2} E^2}{1 - D_T^{\,2} E^2}} \qquad (103)$$
• Special cases: We immediately see that it is advantageous to use a cleaned analyser (see
S.10.10), because with $D_T = 1$, $D_R = -1$ Eq. (102) becomes Eq. (104) and all possible errors in
the Δ90-calibration from Eq. (103) are removed in Eq. (105), besides the problem of temporal
change of $a$.
$$\begin{aligned} & D_T = +1, D_R = -1 \Rightarrow \\ & \frac{\eta^*}{\eta} = \frac{1 + \mathrm{xy} E}{1 - \mathrm{xy} E} \end{aligned} \qquad (104)$$
$$\frac{\eta^*_{\Delta 90}}{\eta} = \sqrt{\frac{1 - E^2}{1 - E^2}} = 1 \Rightarrow \quad \eta = \eta^*_{\Delta 90} \qquad (105)$$
• From Eq. (100) we get Eq. (106) without emitter and receiver optics rotation, without laser
rotation, but with calibrator rotation $\varepsilon$ and with a horizontal-linearly polarised laser $\boldsymbol{I_L}$ (Eq.
(E.5)).
$$\begin{aligned} & \gamma = \beta = \alpha = 0 \wedge \boldsymbol{I}_E = \boldsymbol{I}_L = I_L \left| 1 \quad 1 \quad 0 \quad 0 \right\rangle \Rightarrow \\ & E(\varepsilon, \mathrm{h}, 0, a, 0, 0) = s_{2\varepsilon} \frac{D_O + a}{1 + a D_O} \end{aligned} \qquad (106)$$
• If additionally without calibrator rotation error $\varepsilon$, Eq. (106) becomes Eq. (107) and thus $\eta^*$
and $\eta^*_{\Delta 90}$ are independent of the atmospheric polarisation parameter $a$ and any atmospheric
changes (see Eqs. (102) and (103)).
$$\begin{aligned} & \varepsilon = 0 \Rightarrow \\ & E(0, \mathrm{h}, 0, a, 0, 0) = 0 \end{aligned} \qquad (107)$$
• A more general case without receiver optics rotation $\gamma$ and without calibrator rotation $\varepsilon$, but
with unknown laser and emitter optics rotation, Eq (100) becomes Eq. (108).





with $\gamma = \varepsilon = 0 \Rightarrow E(0,\mathrm{h},0,a,\beta,\alpha) = \dfrac{u_{in}}{i_{in}} = \dfrac{\mathrm{h}Z_O\big[\mathrm{s}_O(1-2a)v_E - \mathrm{c}_O a u_E\big] + (\mathrm{h}-1)a u_E}{i_E + aD_O q_E}$    (108)
Eq. (108) stays quite complex if we use $\boldsymbol{I}_E$ with rotated emitter optics (Eq. (E.12)), and even if
we assume a linearly polarised laser (Eq. (E.9)).
• With a horizontal-linearly polarised laser (Eq. (E.13)) aligned with the rotated emitter optics
($\alpha = \beta$) we get from Eq. (100)

with $\alpha = \beta \wedge \boldsymbol{I}_E = T_E I_L (1+D_E) \| 1 \quad \mathrm{c}_{2\alpha} \quad \mathrm{s}_{2\alpha} \quad 0 \rangle \Rightarrow$

$$E(\varepsilon,\mathrm{h},\gamma,a,\alpha,\alpha) = \frac{\mathrm{s}_{2\varepsilon+\mathrm{h}2\gamma}D_O + a\Big[\big(\mathrm{s}_{2\varepsilon}\mathrm{c}_{2\alpha} - \mathrm{hc}_{2\varepsilon}\mathrm{s}_{2\alpha}\big) + \mathrm{hc}_{2\varepsilon+\mathrm{h}2\gamma}W_O\big(\mathrm{s}_{2\gamma}\mathrm{c}_{2\alpha} + \mathrm{c}_{2\gamma}\mathrm{s}_{2\alpha}\big)\Big]}{1 + aD_O\big(\mathrm{c}_{2\gamma}\mathrm{c}_{2\alpha} - \mathrm{s}_{2\gamma}\mathrm{s}_{2\alpha}\big)} = $$
$$= \frac{\mathrm{s}_{2\varepsilon+\mathrm{h}2\gamma}D_O + a\big(\mathrm{hc}_{2\varepsilon+\mathrm{h}2\gamma}\mathrm{s}_{2\gamma+2\alpha}W_O + \mathrm{s}_{2\varepsilon-\mathrm{h}2\alpha}\big)}{1 + aD_O\mathrm{c}_{2\gamma+2\alpha}}$$
   (109)

Note: $D_E = 0$ means without emitter optics, and $W_O = (1 - Z_O\mathrm{c}_O)$.
• Eq. (109) with laser, emitter and receiver optics aligned with each other becomes

with $\alpha = \beta = -\gamma \wedge \boldsymbol{I}_E = T_E I_L (1+D_E) \| 1 \quad \mathrm{c}_{2\alpha} \quad \mathrm{s}_{2\alpha} \quad 0 \rangle \Rightarrow$

$$E(\varepsilon,\mathrm{h},\gamma,a,-\gamma,-\gamma) = \mathrm{s}_{2\varepsilon+\mathrm{h}2\gamma}\frac{D_O + \mathrm{h}a}{1 + aD_O}$$
   (110)

• Eq. (109) with receiver optics and calibrator aligned =>

with $\alpha = \beta, \varepsilon = -\mathrm{h}\gamma \wedge \boldsymbol{I}_E = T_E I_L (1+D_E) \| 1 \quad \mathrm{c}_{2\alpha} \quad \mathrm{s}_{2\alpha} \quad 0 \rangle \Rightarrow$

$$E(-\gamma,\mathrm{h},\gamma,a,\alpha,\alpha) = \frac{\mathrm{hs}_{2\gamma+2\alpha}a(1 - Z_O\mathrm{c}_O - \mathrm{h})}{1 + aD_O\mathrm{c}_{2\gamma+2\alpha}}$$
   (111)

In summary: the $\Delta 90$-calibration with a cleaned analyser results in a calibration factor $\eta$
independent of $\boldsymbol{I}_{in}$, i.e independent of any optics before the calibrator and independent of the
rotation error $\varepsilon$ of the calibrator. Calibrations without a cleaned analyser include error terms
which increase rapidly with increasing $\varepsilon$ and $\alpha$ for the individual $\pm 45°$ calibrations (Bravo-
Aranda et al., 2016), because $D_T$ and $D_R$ in the numerator and denominator have opposite
signs in Eq. (102). The geometric mean of the two $\pm 45°$ calibrations in Eq. (103) removes the
opposite signs and the increasing error with increasing $\varepsilon$ and $\alpha$ is reduced by orders of
magnitude compared to the individual $\pm 45°$ calibrations (Freudenthaler et al., 2009).



## 7.2 Calibration with a rotator before the receiving optics

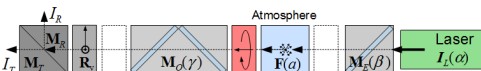

The general formula for the output Stokes vector $I_S$ with rotation calibrator before the
receiving optics $\mathbf{M}_O$ and the polarising beam-splitter $\mathbf{M}_S$ is given in Eq. (112).

$$
\begin{aligned}
I_S &= \eta_S \mathbf{M}_S \mathbf{R}_y \mathbf{M}_O(\gamma) \mathbf{M}_{rot}(x45°+\varepsilon, h) \mathbf{F}(a) \mathbf{M}_E(\beta) I_L(\alpha) = \\
&= \eta_S \mathbf{A}_S(y,\gamma) \mathbf{M}_{rot}(x45°+\varepsilon, h) I_{in}(a,\beta,\alpha)
\end{aligned}
\tag{112}
$$

With $\mathbf{A}_S$ from Eq. (D.7), $\mathbf{M}_{rot}$ from Eq. (S.10.15.1), and $I_{in}$ from App. E.3, i.e. Eq. (E.19), we
get Eq. (113) for the calibration signals using the trigonometric relations in S.12.

$$
\begin{aligned}
\frac{I_S}{\eta_S T_S T_O T_{rot} F_{11} T_E I_L} &= \frac{\langle \mathbf{M}_S \mathbf{R}_y \mathbf{M}_O(\gamma) | \mathbf{M}_{rot}(x45°+\varepsilon, h) | \mathbf{F}(a) I_E \rangle}{T_S T_O T_{rot} F_{11} T_E I_L} = \\
&= \left\langle \begin{matrix} 1+yc_{2\gamma}D_O D_S \\ c_{2\gamma}D_O+yD_S(1-s_{2\gamma}^2 W_O) \\ s_{2\gamma}(D_O+yc_{2\gamma}D_S W_O) \\ -ys_{2\gamma}D_S Z_O s_O \end{matrix} \left| \begin{matrix} 1 & 0 & 0 & 0 \\ 0 & -xs_{2\varepsilon} & -xhc_{2\varepsilon} & 0 \\ 0 & xc_{2\varepsilon} & -xhs_{2\varepsilon} & 0 \\ 0 & 0 & 0 & h \end{matrix} \right| \begin{matrix} i_E \\ aq_E \\ -au_E \\ (1-2a)v_E \end{matrix} \right\rangle = \\
&= (1+yc_{2\gamma}D_O D_S)i_E - yhs_{2\gamma}D_S Z_O s_O(1-2a)v_E + \\
&\quad -xa\{D_O(q_E s_{2\varepsilon-2\gamma}-hu_E c_{2\varepsilon-2\gamma})-yD_S[s_{2\gamma}W_O(q_E c_{2\varepsilon-2\gamma}+hu_E s_{2\varepsilon-2\gamma})-(q_E s_{2\varepsilon}-hu_E c_{2\varepsilon})]\}
\end{aligned}
\tag{113}
$$

Special cases: Without receiver optics rotation, i.e. $\gamma=0$, Eq. (113) becomes Eq. (114), which
is less complex and independent of retardation terms $Z_O s_O$ and $W_O$, and the gain ratios $\eta^*$
(Sect. 5) can be written as Eqs. (115) and (116).

$$
\gamma=0 \Rightarrow
$$
$$
I_S/(\eta_S T_S T_O T_{rot} F_{11} T_E I_L) = (1+yD_O D_S)i_E - xa(D_O+yD_S)(q_E s_{2\varepsilon}-hu_E c_{2\varepsilon})
\tag{114}
$$

$$
\frac{\eta^*}{\eta} = \frac{(1+yD_O D_R)i_E - xa(D_O+yD_R)(q_E s_{2\varepsilon}-hu_E c_{2\varepsilon})}{(1+yD_O D_T)i_E - xa(D_O+yD_T)(q_E s_{2\varepsilon}-hu_E c_{2\varepsilon})}
\tag{115}
$$

$$
\frac{\eta^*_{\Delta 90}}{\eta} = \sqrt{\frac{(1+yD_O D_R)^2 i_E^2 - (D_O+yD_R)^2(q_E s_{2\varepsilon}-hu_E c_{2\varepsilon})^2}{(1+yD_O D_T)^2 i_E^2 - (D_O+yD_T)^2(q_E s_{2\varepsilon}-hu_E c_{2\varepsilon})^2}}
\tag{116}
$$

• With a cleaned analyser (see S.10.10) Eqs. (115) and (116) become Eqs. (117) and (118).





$$\gamma = 0°, D_T = +1, D_R = -1 \Rightarrow$$

$$\frac{\eta^*}{\eta} = \frac{1 - yD_O}{1 + yD_O} \frac{i_E + xya\left(q_E s_{2\varepsilon} - hu_E c_{2\varepsilon}\right)}{i_E - xya\left(q_E s_{2\varepsilon} - hu_E c_{2\varepsilon}\right)}$$ (117)
$$\frac{\eta^*_{\Delta 90}}{\eta} = \frac{1 - yD_O}{1 + yD_O}$$ (118)
The gain ratio $\eta^*_{\Delta 90}$ in Eq. (118) is independent of the input Stokes vector, i.e. the laser
polarisation, independent of the calibrator type (mechanical or $\lambda/2$ plate rotation) and of the
calibrator rotation $\varepsilon$. Using the two calibration setups Eqs. (118) and (105) it is possible to
retrieve the receiver optics diattenuation parameter $D_O$ (Belegante et al., 2016) . Furthermore,
with this setup and the measured gain ratio $\eta^*_{\Delta 90}$ from Eq. (118) we get the polarisation
parameter $a$ (Eq.(119)) and the backscatter coefficient $F_{11}$ (Eq.(120)) with Eq. (78) directly
from the measurement signals $I_R$ and $I_T$ according to Eqs. (61) and (65) without the explicit
knowledge of $D_O$ or any other correction.
$$a = y\frac{\eta^*_{\Delta 90}I_T - I_R}{\eta^*_{\Delta 90}I_T + I_R}$$ (119)
$$F_{11} \propto \eta^*_{\Delta 90}I_T + I_R$$ (120)
**7.3   Calibration with a rotator behind the emitter optics**

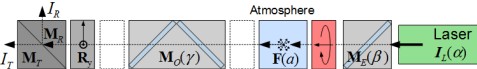

The general formula for the output Stokes vector $\boldsymbol{I}_S$ with rotation calibrator $\mathbf{M}_{rot}$ (Eq.
(S.10.15.2)) behind the emitter optic $\mathbf{M}_E$ and all derivations therefrom can be derived from the
previous Sect. 7.2 using Eq. (121) and considering the mirror effect of $\mathbf{F}$ and the associated
sign changes in the rotation angle (S.6.3) when mathematically moving the calibrator $\mathbf{M}_{rot}$
from behind the emitter optics $\mathbf{M}_E$ to before the receiving optics $\mathbf{M}_O$. Regarding the rotation
and mirror relations see S.5 and S.6.

$$\begin{aligned}
\boldsymbol{I}_S &= \eta_S \mathbf{M}_S \mathbf{R}_y \mathbf{M}_O(\gamma)\mathbf{F}(a)\mathbf{M}_{rot}(x45° + \varepsilon, h)\mathbf{M}_E(\beta)\boldsymbol{I}_L(\alpha) = \\
&= \eta_S \mathbf{M}_S \mathbf{R}_y \mathbf{M}_O(\gamma)\mathbf{F}(a)\mathbf{R}(x45°)\mathbf{R}(\varepsilon)\mathbf{M}_h\mathbf{M}_E(\beta)\boldsymbol{I}_L(\alpha) = \\
&= \eta_S \mathbf{M}_S \mathbf{R}_y \mathbf{M}_O(\gamma)\mathbf{R}(-x45°)\mathbf{R}(-\varepsilon)\mathbf{M}_h\mathbf{F}(a)\mathbf{M}_E(\beta)\boldsymbol{I}_L(\alpha) \\
&= \eta_S \mathbf{M}_S \mathbf{R}_y \mathbf{M}_O(\gamma)\mathbf{M}_{rot}(-x45° - \varepsilon, h)\mathbf{F}(a)\mathbf{M}_E(\beta)\boldsymbol{I}_L(\alpha)
\end{aligned}$$ (121)





**8 Calibration with a linear polariser (P)**
A linear polariser is a retarding linear diattenuator (Sect. S.10.3). The output of an ideal linear
polariser is linearly polarised light independent of the state of polarisation of the input, which
seems to be ideal for our purpose. Polarising sheet filters are thin and have large acceptance
angles. Hence they can be easily included in existing lidar systems, even in diverging or
converging light paths as close to the telescope focus. However, to achieve an acceptable
uncertainty of the calibration factor, a rather good extinction ratio of the linear polariser of
order $10^{-4}$ and better is necessary. Crystal polarisers exhibit such high extinction ratios, but the
available diameters are limited, they are bulky and have smaller acceptance angles. Wire grid
and liquid crystal polarisers usually don't show high enough extinction ratios. A linear
polariser is described in the same way as a polarising beam-splitter, which is a retarding
diattenuator (S.4 and S.10.3 ff), with high diattenuation ($\mathbf{D}_P \approx 1$). Since the standard
atmospheric measurements have to be performed without the linearly polarising calibrator,
there is no rotational misalignment $\varepsilon$ for the standard measurement signals of Sect. 4. As the
equations become too complex with a real linear polariser with diattenuation and retardation,
we use a real linear polariser only in Sect. 8.1 to show as an example how the uncertainty of
the extinction ratio influences the accuracy of the calibration factor, and else we use an ideal
linear polariser. The general formula with a real linear polariser can be found in App. C.2.
**8.1 Calibration with a linear polariser before the polarising beam-splitter**

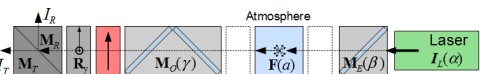

$$\begin{aligned}\boldsymbol{I}_S &= \eta_S \mathbf{M}_S \mathbf{R}(\mathbf{y}) \mathbf{M}_P(\mathbf{x}45° + \varepsilon) \mathbf{M}_O(\gamma) \mathbf{F}(a) \mathbf{M}_E(\beta) \boldsymbol{I}_L(\alpha) = \\ &= \eta_S \mathbf{A}_S(\mathbf{y}) \mathbf{M}_P(\mathbf{x}45° + \varepsilon) \boldsymbol{I}_{in}(\gamma, a, \beta, \alpha)\end{aligned}$$
(122)

With Eq. (D.5) for the analyser part $\mathbf{A}_S$, Eq. (S.10.6.1) for the rotated linear polariser, and $\boldsymbol{I}_{in}$
from App. E.4 we get the general calibration signals Eq. (123).





$$\frac{I_S}{\eta_S T_S T_P I_{in}} = \frac{\langle \mathbf{M}_S \mathbf{R}_y | \mathbf{M}_P \left( \text{x}45° + \varepsilon \right) | \mathbf{I}_{in} \rangle}{T_S T_P I_{in}} =$$
$$= \left\langle \begin{matrix} 1 - \text{xys}_{2\varepsilon} D_P D_S \\ -\text{xs}_{2\varepsilon} D_P + \text{y} D_S \left( 1 - \text{c}_{2\varepsilon}^2 W_P \right) \\ \text{xc}_{2\varepsilon} D_P - \text{ys}_{2\varepsilon} \text{c}_{2\varepsilon} W_P D_S \\ -\text{xyc}_{2\varepsilon} Z_P \text{s}_P D_S \end{matrix} \middle| \begin{matrix} i_{in} \\ q_{in} \\ u_{in} \\ v_{in} \end{matrix} \right\rangle =$$
$$= i_{in} + \text{y} D_S \left[ q_{in} - \text{c}_{2\varepsilon} W_P \left( \text{c}_{2\varepsilon} q_{in} + \text{s}_{2\varepsilon} u_{in} \right) \right] -$$
$$- \text{x} \left[ D_P \left( \text{s}_{2\varepsilon} q_{in} - \text{c}_{2\varepsilon} u_{in} \right) + \text{y} D_S \left( \text{s}_{2\varepsilon} D_P i_{in} + \text{c}_{2\varepsilon} Z_P \text{s}_P v_{in} \right) \right]$$
(123)

• Special cases: Without calibrator rotation error $\varepsilon$ Eq.(123) becomes Eq. (124).

$$\varepsilon = 0 \Rightarrow$$

$$\frac{I_S}{\eta_S T_S T_P I_{in}} = i_{in} + \text{y} D_S \left[ 1 - W_P \right] q_{in} + \text{x} \left[ u_{in} D_P - \text{y} D_S Z_P \text{s}_P v_{in} \right] =$$
$$= i_{in} + \text{x} D_P u_{in} + \text{y} D_S Z_P \left( \text{c}_P q_{in} - \text{s}_P v_{in} \right)$$
(124)

• We get with a cleaned analyser and horizontal-linearly polarised input $\mathbf{I}_{in}$, with Eq. (124) the
gain ratios (Sect. 5) in Eq. (125).

$$\varepsilon = 0, D_T = +1, D_R = -1, \mathbf{I}_{in} = | 1 \quad 1 \quad 0 \quad 0 \rangle \Rightarrow$$

$$\frac{\eta^*}{\eta} = \frac{1 - \text{y} Z_P}{1 + \text{y} Z_P}$$
(125)

• Using Eq. (S.10.10.8) for the extinction ratio $\rho$ of the real linear polariser, we get the
approximation Eq. (126) for the gain ratios depending on $\rho$, with which we can estimate the
error of the gain ratio if we use a real polariser with extinction ratio $\rho$ for the measurements
but assume an ideal polariser as calibrator in the correction equations. Eq. (126) with $\rho = 10^{-5}$
and $\rho = 10^{-4}$, e.g., gives relative errors of the gain ratios of about 1.3% and 8%, respectively.

with $\rho = k_2 / k_1$ and $k_2 \ll k_1 \Rightarrow$

$$\frac{\eta^*}{\eta} \approx \frac{1 - 2\text{y}\sqrt{\rho}}{1 + 2\text{y}\sqrt{\rho}} \approx 1 - 4\text{y}\sqrt{\rho}$$
(126)

• With an ideal linear polariser Eq.(123) becomes Eq. (127), and the gain ratios Eq. (128) are
independent of $\mathbf{I}_{in}$, i.e. independent of the laser polarisation, of the atmospheric depolarisation,
and of any optics before the calibrator. The error due to the calibrator rotation $\varepsilon$ is largely
reduced with the $\Delta$90-calibration in Eq. (129) compared to the ±45°-calibration in Eq. (128).



$$D_P = 1 \Rightarrow W_P = 1, Z_P = 0 \Rightarrow$$

$$\frac{I_S}{\eta_S T_S T_P I_{in}} = \left(1 - \mathrm{xys}_{2\varepsilon} D_S\right)\langle 1 \quad -\mathrm{xs}_{2\varepsilon} \quad \mathrm{xc}_{2\varepsilon} \quad 0 | i_{in} \quad q_{in} \quad u_{in} \quad v_{in}\rangle =$$
$$= \left(1 - \mathrm{xys}_{2\varepsilon} D_S\right)\left[ i_{in} - \mathrm{x}\left(\mathrm{s}_{2\varepsilon} q_{in} - \mathrm{c}_{2\varepsilon} u_{in}\right)\right]$$
(127)

$$\frac{\eta^*}{\eta} = \frac{1 - \mathrm{xys}_{2\varepsilon} D_R}{1 - \mathrm{xys}_{2\varepsilon} D_T}$$
(128)

$$\frac{\eta^*_{\Delta 90}}{\eta} = \sqrt{\frac{1 - \mathrm{s}_{2\varepsilon}^{\,2} D_R^{\,2}}{1 - \mathrm{s}_{2\varepsilon}^{\,2} D_T^{\,2}}}$$
(129)

• If additionally a cleaned analyser is used (see S.10.10), Eqs. (128) and (129) become Eqs.
(130) and (131). Eq. (130) is of the form of Eq. (193) and can be used to determine ε (see
Sect. 11). Eq. (131) shows that the Δ90-calibration with a cleaned analyser is free of ε error.

with  $D_P = 1, W_P = 1, Z_P = 0, D_T = +1, D_R = -1 \Rightarrow$

$$\frac{\eta^*}{\eta} = \frac{1 + \mathrm{xys}_{2\varepsilon}}{1 - \mathrm{xys}_{2\varepsilon}}$$
(130)

$$\eta^*_{\Delta 90} = \eta = \frac{\eta_R T_R^s}{\eta_T T_T^p}$$
(131)

**8.2   Calibration with an ideal linear polariser before the receiving optics**

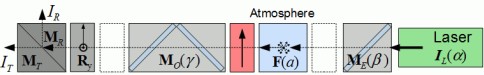

$$I_S = \eta_S \mathbf{M}_S \mathbf{R}(\mathrm{y}) \mathbf{M}_O(\gamma) \mathbf{M}_P(\mathrm{x}45° + \varepsilon) \mathbf{F}(a) \mathbf{M}_E(\beta) I_L(\alpha) =$$
$$= \eta_S \mathbf{A}_S(\mathrm{y}, \gamma) \mathbf{M}_P(\mathrm{x}45° + \varepsilon) I_{in}(a, \beta, \alpha)$$
(132)

With Eq. (D.7) for the analyser part $\mathbf{A}_S$, Eq. (S.10.8.6) for the ideal linear polariser $\mathbf{M}_P$, and
any of the input Stokes vectors $I_{in}$ of App. E.3 we get Eq. (133) for the calibration signals $I_S$.
Since the last term of Eq. (133) is independent of the analyser diattenuation parameters $D_S$,
this term cancels in the ratio of the gain ratios (Sect. 5) in Eq. (134), which are therefore
independent of the input Stokes vector.





*with* $D_P = 1 \Rightarrow$

$$\frac{I_S}{\eta_S T_S T_O T_P F_{11} T_E I_L} = \frac{\langle \mathbf{M}_S \mathbf{R}_y \mathbf{M}_O(\gamma) | \mathbf{M}_P(x45° + \varepsilon) | \mathbf{F}(a) \mathbf{I}_E \rangle}{T_S T_O T_P F_{11} T_E I_L} =$$

$$= \left\langle \begin{matrix} 1 + yc_{2\gamma} D_S D_O \\ c_{2\gamma} D_O + yD_S\left(1 - s_{2\gamma}^2 W_O\right) \\ s_{2\gamma}\left(D_O + yc_{2\gamma} D_S W_O\right) \\ -ys_{2\gamma} D_S Z_O s_O \end{matrix} \middle\| \begin{matrix} 1 \\ -xs_{2\varepsilon} \\ xc_{2\varepsilon} \\ 0 \end{matrix} \right\rangle \left\langle \begin{matrix} 1 \\ -xs_{2\varepsilon} \\ xc_{2\varepsilon} \\ 0 \end{matrix} \middle\| \begin{matrix} i_E \\ aq_E \\ -au_E \\ (1-2a)v_E \end{matrix} \right\rangle = \qquad (133)$$

$$= \left[\left(1 + yc_{2\gamma} D_S D_O\right) - x\left\{s_{2\varepsilon-2\gamma} D_O + yD_S\left[s_{2\varepsilon} - s_{2\gamma} c_{2\varepsilon-2\gamma} W_O\right]\right\}\right]\left[i_E - xa\left(s_{2\varepsilon} q_E + c_{2\varepsilon} u_E\right)\right]$$

$2 \quad \dfrac{\eta^*}{\eta} = \dfrac{\left(1 + yc_{2\gamma} D_O D_R\right) - x\left[s_{2\varepsilon-2\gamma} D_O + yD_R\left(s_{2\varepsilon} - s_{2\gamma} c_{2\varepsilon-2\gamma} W_O\right)\right]}{\left(1 + yc_{2\gamma} D_O D_T\right) - x\left[s_{2\varepsilon-2\gamma} D_O + yD_T\left(s_{2\varepsilon} - s_{2\gamma} c_{2\varepsilon-2\gamma} W_O\right)\right]} \qquad (134)$

• Special cases: Eq.(134) gets neither • with a cleaned analyser alone (Eq. (135)) nor • without
receiver optics rotation $\gamma$ alone (Eq. (136)) very simple, but • with both conditions Eq. (137) is
of the form of Eq. (193) and can be used to estimate the calibrator rotation $\varepsilon$ (see Sect. 11).
The corresponding $\Delta$90-calibration in Eq. (138) can be used together with the calibration
measurements which directly yield $\eta$ (see Eqs. (131) or (105), for example) to determine the
diattenuation parameter $D_O$ of the receiving optics.

with $D_P = 1, D_T = +1, D_R = -1 \Rightarrow$

$9 \quad \dfrac{\eta^*}{\eta} = \dfrac{\left(1 - yc_{2\gamma} D_O\right) - x\left[s_{2\varepsilon-2\gamma} D_O - y\left(s_{2\varepsilon} - s_{2\gamma} c_{2\varepsilon-2\gamma} W_O\right)\right]}{\left(1 + yc_{2\gamma} D_O\right) - x\left[s_{2\varepsilon-2\gamma} D_O + y\left(s_{2\varepsilon} - s_{2\gamma} c_{2\varepsilon-2\gamma} W_O\right)\right]} \qquad (135)$

with $D_P = 1, \gamma = 0 \Rightarrow$

$10 \quad \dfrac{\eta^*}{\eta} = \dfrac{\left(1 + yD_O D_R\right) - xs_{2\varepsilon}\left[D_O + yD_R\right]}{\left(1 + yD_O D_T\right) - xs_{2\varepsilon}\left[D_O + yD_T\right]} \qquad (136)$

with $D_P = 1, D_T = +1, D_R = -1, \gamma = 0 \Rightarrow$

$11 \quad \dfrac{\eta^*}{\eta} = \dfrac{\left(1 - yD_O\right) - xs_{2\varepsilon}\left(D_O - y\right)}{\left(1 + yD_O\right) - xs_{2\varepsilon}\left(D_O + y\right)} = \dfrac{1 - yD_O}{1 + yD_O}\dfrac{1 + xys_{2\varepsilon}}{1 - xys_{2\varepsilon}} \qquad (137)$

$12 \quad \dfrac{\eta^*_{\Delta90}}{\eta} = \dfrac{1 - yD_O}{1 + yD_O} \qquad (138)$



## 1  8.3  Calibration with an ideal linear polariser behind the emitter optics

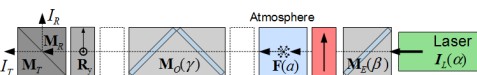

$$I_S = \eta_S \mathbf{M}_S \mathbf{R}_y \mathbf{M}_O(\gamma) \mathbf{F}(a) \mathbf{M}_P(x45° + \varepsilon) \mathbf{M}_E(\beta) \boldsymbol{I}_L(\alpha) =$$
$$= \eta_S \mathbf{A}_S(y, \gamma, a) \mathbf{M}_P(x45° + \varepsilon) \boldsymbol{I}_{in}(\beta, \alpha)$$
(139)

With Eq. (D.13) for the analyser part $\mathbf{A}_S$, Eq. (S.10.8.6) for the ideal linear polariser $\mathbf{M}_P$, and
any of the emitter Stokes vectors $\boldsymbol{I}_E$ of App. E.2 we get the calibration signals $I_S$ in Eq. (140).
Since the the last term of Eq. (140) is independent of analyser diattenuation parameters $D_S$, it
cancels in the ratio of the gain ratios (Sect. 5), and the gain ratios in Eq. (141) are independent
of the input Stokes vector.

with $D_P = 1 \Rightarrow$

$$\frac{I_S}{\eta_S T_S T_O F_{11} T_P I_{in}} = \frac{\langle \mathbf{M}_S \mathbf{R}_y \mathbf{M}_O(\gamma) | \mathbf{F}(a) \mathbf{M}_P(x45° + \varepsilon) | \boldsymbol{I}_E \rangle}{T_S T_O F_{11} T_P I_E} =$$
$$= \left\langle \begin{matrix} 1 + yc_{2\gamma} D_S D_O \\ a\left[ c_{2\gamma} D_O + yD_S\left(1 - s_{2\gamma}^2 W_O\right)\right] \\ -a\, s_{2\gamma}\left(D_O + yc_{2\gamma} D_S W_O\right) \\ -(1-2a)\, ys_{2\gamma} D_S Z_O s_O \end{matrix} \middle| \begin{matrix} 1 \\ -xs_{2\varepsilon} \\ xc_{2\varepsilon} \\ 0 \end{matrix} \right\rangle \left\langle \begin{matrix} 1 \\ -xs_{2\varepsilon} \\ xc_{2\varepsilon} \\ 0 \end{matrix} \middle| \begin{matrix} i_E \\ q_E \\ u_E \\ v_E \end{matrix} \right\rangle =$$
$$= \left[\left(1 + yc_{2\gamma} D_S D_O\right) - ax\left\{ s_{2\varepsilon+2\gamma} D_O + yD_S\left[ s_{2\varepsilon} + s_{2\gamma} c_{2\varepsilon+2\gamma} W_O\right]\right\}\right]\left[ i_E - x\left( s_{2\varepsilon} q_E - c_{2\varepsilon} u_E\right)\right]$$
(140)

$$\frac{\eta^*}{\eta} = \frac{\left(1 + yc_{2\gamma} D_O D_R\right) - xa\left[ s_{2\varepsilon+2\gamma} D_O + yD_R\left( s_{2\varepsilon} + s_{2\gamma} c_{2\varepsilon+2\gamma} W_O\right)\right]}{\left(1 + yc_{2\gamma} D_O D_T\right) - xa\left[ s_{2\varepsilon+2\gamma} D_O + yD_T\left( s_{2\varepsilon} + s_{2\gamma} c_{2\varepsilon+2\gamma} W_O\right)\right]}$$
(141)

• Special cases: Eq.(141) with a cleaned analyser becomes Eq. (142), without receiver optics
rotation Eq. (143), and with both conditions Eq. (144). Eq. (144) is of the form of Eq. (199)
and can be used to determine ε (see Sect. 11). As before in Eq.(138) the corresponding Δ90-
calibration becomes Eq. (145),

with $D_P = 1, D_T = +1, D_R = -1 \Rightarrow$

$$\frac{\eta^*}{\eta} = \frac{\left(1 - yc_{2\gamma} D_O\right) - xa\left[ s_{2\varepsilon+2\gamma} D_O - y\left( s_{2\varepsilon} + s_{2\gamma} c_{2\varepsilon+2\gamma} W_O\right)\right]}{\left(1 + yc_{2\gamma} D_O\right) - xa\left[ s_{2\varepsilon+2\gamma} D_O + y\left( s_{2\varepsilon} + s_{2\gamma} c_{2\varepsilon+2\gamma} W_O\right)\right]}$$
(142)

with $D_P = 1, \gamma = 0 \Rightarrow$

$$\frac{\eta^*}{\eta} = \frac{\left(1 + yD_O D_R\right) - xas_{2\varepsilon}\left[ D_O + yD_R\right]}{\left(1 + yD_O D_T\right) - xas_{2\varepsilon}\left[ D_O + yD_T\right]}$$
(143)





with $D_P = 1, D_T = +1, D_R = -1, \gamma = 0 \Rightarrow$

$$\frac{\eta^*}{\eta} = \frac{(1 - yD_O) - xas_{2\varepsilon}(D_O - y)}{(1 + yD_O) - xas_{2\varepsilon}(D_O + y)} = \frac{(1 - yD_O)(1 + xyas_{2\varepsilon})}{(1 + yD_O)(1 - xyas_{2\varepsilon})}$$
(144)

$$\frac{\eta^*_{\Delta 90}}{\eta} = \frac{1 - yD_O}{1 + yD_O}$$
(145)

## 9  Calibration with a λ/4 plate (QWP)
A λ/4-plate (QWP) is a retarding linear diattenuator (S.4) with 90° phase shift between the
polarisation parallel and perpendicular to the fast axis and without diattenuation (S.10.16 ff).
Further details can be found in Bennett (2009a), Bennett (2009b), and Chipman (2009b).
Oriented at ±45° relative to incident linear polarisation, its output is circularly polarised. Since
the equations with a real QWP with retardation error ω (S.10.16) are too complex, we
consider ω only in Sect. 9.1 to show with an example how this uncertainty influences the
accuracy of the calibration factor. The general formula with a real QWP can be found in App.
C.3.
### 9.1  Calibration with a λ/4 plate before the the polarising beam-splitter

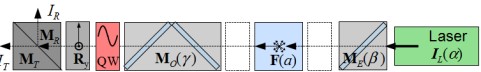

$$I_S = \eta_S \mathbf{M}_S \mathbf{R}_y \mathbf{M}_{QW}(x45° + \varepsilon, \omega) \mathbf{M}_O(\gamma) \mathbf{F}(a) I_E =$$
$$\eta_S \mathbf{A}_S(y) \mathbf{M}_{QW}(x45° + \varepsilon, \omega) I_{in}(\gamma, a, \beta, \alpha)$$
(146)

With Eq. (D.5) for the analyser part $\mathbf{A}_S$, Eq. (S.10.16.3) for the λ/4 plate $\mathbf{M}_{QW}$ with phase shift
error ω, and with the input Stokes vector $I_{in}$ from App. E.4 we get the calibration signals $I_S$ in
Eq. (147).

$$\Delta_{QW} = 90° + \omega \Rightarrow$$
$$\frac{I_S}{\eta_S T_S T_{QW} I_{in}} = \frac{\langle \mathbf{M}_S \mathbf{R}_y | \mathbf{M}_{QW}(x45° + \varepsilon, \omega) | I_{in} \rangle}{\eta_S T_S T_{QW} I_{in}} =$$

$$= \left\langle \begin{matrix} 1 \\ yD_S \\ 0 \\ 0 \end{matrix} \middle| \begin{pmatrix} 1 & 0 & 0 & 0 \\ 0 & s_{2\varepsilon}^2 - c_{2\varepsilon}^2 s_\omega & -s_{2\varepsilon}c_{2\varepsilon}(1 + s_\omega) & -xc_{2\varepsilon}c_\omega \\ 0 & -s_{2\varepsilon}c_{2\varepsilon}(1 + s_\omega) & c_{2\varepsilon}^2 - s_{2\varepsilon}^2 s_\omega & -xs_{2\varepsilon}c_\omega \\ 0 & xc_{2\varepsilon}c_\omega & xs_{2\varepsilon}c_\omega & -s_\omega \end{pmatrix} \middle| \begin{matrix} i_{in} \\ q_{in} \\ u_{in} \\ v_{in} \end{matrix} \right\rangle =$$

$$= i_{in} + yD_S \left[ s_{2\varepsilon}(s_{2\varepsilon}q_{in} - c_{2\varepsilon}u_{in}) - c_{2\varepsilon}s_\omega(c_{2\varepsilon}q_{in} + s_{2\varepsilon}u_{in}) - xc_{2\varepsilon}c_\omega v_{in} \right]$$
(147)



• Special cases: For the investigation of the effect of the phase shift error ω we neglect the
rotation error ε in Eq. (147) and get the calibration signals Eq. (148) and the gain ratios Eq.

3  (149).

$$\varepsilon = 0 \Rightarrow$$
$$I_S = \eta_S T_S T_{QW} I_{in} \left[ i_{in} - yD_S \left( \mathrm{s}_\omega q_{in} + \mathrm{xc}_\omega v_{in} \right) \right]$$
(148)

$$\frac{\eta^*}{\eta} = \frac{i_{in} - yD_R \mathrm{s}_\omega q_{in} - xyD_R \mathrm{c}_\omega v_{in}}{i_{in} - yD_T \mathrm{s}_\omega q_{in} - xyD_T \mathrm{c}_\omega v_{in}}$$
(149)

• With a cleaned analyser, the gain ratios from Eq. (149) become Eq. (150) and for the Δ90-
calibration Eq. (151), from which we can estimate the influence of a phase shift error ω.

$$\text{with} \quad \varepsilon = 0, D_T = +1, D_R = -1 \Rightarrow$$
$$\frac{\eta^*}{\eta} = \frac{i_{in} + y \mathrm{s}_\omega q_{in} + xy \mathrm{c}_\omega v_{in}}{i_{in} - y \mathrm{s}_\omega q_{in} - xy \mathrm{c}_\omega v_{in}}$$
(150)

$$\frac{\eta^*_{\Delta 90}}{\eta} = \sqrt{\frac{\left( i_{in} + y \mathrm{s}_\omega q_{in} \right)^2 - \mathrm{c}_\omega^2 v_{in}^2}{\left( i_{in} - y \mathrm{s}_\omega q_{in} \right)^2 - \mathrm{c}_\omega^2 v_{in}^2}}$$
(151)

• Without phase shift error ω in Eq. (147) but with calibrator rotation error ε we get the
calibration signals Eq. (152) and the gain ratios Eq. (153).

$$\omega = 0 \Rightarrow$$
$$I_S = \eta_S T_S T_{QW} I_{in} \left\{ i_{in} + yD_S \left[ \mathrm{s}_{2\varepsilon} \left( \mathrm{s}_{2\varepsilon} q_{in} - \mathrm{c}_{2\varepsilon} u_{in} \right) - \mathrm{xc}_{2\varepsilon} v_{in} \right] \right\}$$
(152)

$$\frac{\eta^*}{\eta} = \frac{i_{in} + yD_R \mathrm{s}_{2\varepsilon} \left( \mathrm{s}_{2\varepsilon} q_{in} - \mathrm{c}_{2\varepsilon} u_{in} \right) - xyD_R \mathrm{c}_{2\varepsilon} v_{in}}{i_{in} + yD_T \mathrm{s}_{2\varepsilon} \left( \mathrm{s}_{2\varepsilon} q_{in} - \mathrm{c}_{2\varepsilon} u_{in} \right) - xyD_T \mathrm{c}_{2\varepsilon} v_{in}}$$
(153)

$$\omega = 0, D_T = +1, D_R = -1 \Rightarrow$$
$$\frac{\eta^*_{\Delta 90}}{\eta} = \sqrt{\frac{\left( i_{in} - y \mathrm{s}_{2\varepsilon} \left( \mathrm{s}_{2\varepsilon} q_{in} - \mathrm{c}_{2\varepsilon} u_{in} \right) \right)^2 - \mathrm{c}_{2\varepsilon}^2 v_{in}^2}{\left( i_{in} + y \mathrm{s}_{2\varepsilon} \left( \mathrm{s}_{2\varepsilon} q_{in} - \mathrm{c}_{2\varepsilon} u_{in} \right) \right)^2 - \mathrm{c}_{2\varepsilon}^2 v_{in}^2}}$$
(154)

The terms without the x-factor in Eq. (150) containing ω and in Eq. (153) containing ε are not
compensated with the Δ90-calibration in Eq. (151) and Eq. (154), even if a cleaned analyser is
used. This is a disadvantage of the QWP compared to the linear polariser (see Eq. (129)).
• From Eq.(153) without calibrator rotation ε we get the gain ratios Eqs. (155) and (156).



$$\omega = \varepsilon = 0 \Rightarrow$$

$$\frac{\eta^*}{\eta} = \frac{i_{in} - xyD_R v_{in}}{i_{in} - xyD_T v_{in}} \tag{155}$$
$$\frac{\eta^*_{\Delta 90}}{\eta} = \sqrt{\frac{i_{in}^2 - D_R^2 v_{in}^2}{i_{in}^2 - D_T^2 v_{in}^2}} \tag{156}$$
• With a cleaned analyser Eq. (156) becomes Eq. (157).
$$\omega = \varepsilon = 0, D_T = +1, D_R = -1 \Rightarrow$$
$$\eta^*_{\Delta 90} = \eta \tag{157}$$

• The advantage of the QWP calibrator is that we can retrieve from Eqs. (157) and (155) with
a cleaned analyser the degree of circular polarisation $v_{in}/i_{in}$ of the light before the polarising
beam-splitter according to Eq. (158). Bear in mind that $\eta^*$ and $\eta^*_{\Delta 90}$ in Eq. (158) are values
directly derived from measured signals. The errors due to uncertainties in $\varepsilon$ or $\omega$ can be
estimated by means of equations earlier in this section.
$$\omega = \varepsilon = 0, D_T = +1, D_R = -1 \Rightarrow$$
$$\frac{v_{in}}{i_{in}} = \frac{1}{xy} \frac{\eta^* - \eta^*_{\Delta 90}}{\eta^* + \eta^*_{\Delta 90}} \tag{158}$$

**9.2 Calibration with an ideal λ/4 plate before the receiving optics**
$$\boldsymbol{I}_S = \eta_S \mathbf{M}_S \mathbf{R}(y) \mathbf{M}_O(\gamma) \mathbf{M}_{QW}(x45° + \varepsilon) \mathbf{F}(a) \boldsymbol{I}_E(\beta, \alpha) =$$
$$= \eta_S \mathbf{A}_S(y, \gamma) \mathbf{M}_{QW}(x45° + \varepsilon) \boldsymbol{I}_{in}(a, \beta, \alpha) \tag{159}$$

With Eq. (D.7) for the analyser part $\mathbf{A}_S$, an ideal λ/4 plate $\mathbf{M}_{QW}$ Eq. (S.10.17.2), and with an
input Stokes vector $\boldsymbol{I}_{in}$ from App. E.3 we get the general calibration signals $I_S$ in Eq. (160).

$$
\frac{I_S}{\eta_S T_S T_O T_{QW} F_{11} T_E I_L} = \frac{\left\langle \mathbf{M}_S \mathbf{R}_y \mathbf{M}_O(\gamma) \middle| \mathbf{M}_{QW}(\text{x}45° + \varepsilon) \middle| \mathbf{F}(a) \mathbf{M}_E \mathbf{I}_L \right\rangle}{T_S T_O T_{QW} F_{11} T_E I_L} =
$$

$$
= \left\langle
\begin{matrix}
1 + yc_{2\gamma} D_S D_O \\
c_{2\gamma} D_O + yD_S\left(1 - s_{2\gamma}^2 W_O\right) \\
s_{2\gamma}\left(D_O + yc_{2\gamma} D_S W_O\right) \\
-ys_{2\gamma} D_S Z_O s_O
\end{matrix}
\middle|
\begin{pmatrix}
1 & 0 & 0 & 0 \\
0 & s_{2\varepsilon}^2 & -s_{2\varepsilon}c_{2\varepsilon} & -xc_{2\varepsilon} \\
0 & -s_{2\varepsilon}c_{2\varepsilon} & c_{2\varepsilon}^2 & -xs_{2\varepsilon} \\
& xc_{2\varepsilon} & xs_{2\varepsilon} & 0
\end{pmatrix}
\middle|
\begin{matrix}
i_E \\
aq_E \\
-au_E \\
(1-2a)v_E
\end{matrix}
\right\rangle =
$$
(160)

$$
= \left\langle
\begin{matrix}
+ yc_{2\gamma} D_S D_O \\
s_{2\varepsilon} D_O s_{2\varepsilon-2\gamma} + yD_S\left[s_{2\varepsilon}^2 - s_{2\varepsilon}s_{2\gamma}W_O c_{2\varepsilon-2\gamma} - xc_{2\varepsilon}s_{2\gamma}Z_O s_O\right] \\
-c_{2\varepsilon} D_O s_{2\varepsilon-2\gamma} - yD_S\left[s_{2\varepsilon}c_{2\varepsilon} - c_{2\varepsilon}s_{2\gamma}W_O c_{2\varepsilon-2\gamma} + xs_{2\varepsilon}s_{2\gamma}Z_O s_O\right] \\
-x\left\{D_O c_{2\varepsilon-2\gamma} + yD_S\left[c_{2\varepsilon} + s_{2\gamma}W_O s_{2\varepsilon-2\gamma}\right]\right\}
\end{matrix}
\middle|
\begin{matrix}
i_E \\
aq_E \\
-au_E \\
(1-2a)v_E
\end{matrix}
\right\rangle
$$

• Special cases: Without receiver optics rotation $\gamma$ we get from Eq. (E.19) and Eq.(160) the
calibration signals Eq. (161) and the gain ratios Eq. (162).

$$\gamma = 0 \Rightarrow$$

$$
\frac{I_S}{\eta_S T_S T_O T_{QW} F_{11} T_E I_L} = \frac{\left\langle \mathbf{A}_S(y,0) \middle| \mathbf{M}_{QW}(\text{x}45° + \varepsilon) \middle| \mathbf{F}(a)\mathbf{M}_E \mathbf{I}_L \right\rangle}{T_S T_O T_{QW}} = \frac{}{F_{11} T_E I_L} =
$$

$$
= \left\langle
\begin{matrix}
1 + yD_S D_O \\
s_{2\varepsilon}^2\left(D_O + yD_S\right) \\
-c_{2\varepsilon}s_{2\varepsilon}\left(D_O + yD_S\right) \\
-xc_{2\varepsilon}\left(D_O + yD_S\right)
\end{matrix}
\middle|
\begin{matrix}
i_E \\
aq_E \\
-au_E \\
(1-2a)v_E
\end{matrix}
\right\rangle =
$$
(161)

$$
= \left(1 + yD_S D_O\right)i_E + \left(D_O + yD_S\right)\left[s_{2\varepsilon}a\left(s_{2\varepsilon}q_E + c_{2\varepsilon}u_E\right) - xc_{2\varepsilon}\left(1-2a\right)v_E\right]
$$

$$\gamma = 0 \Rightarrow$$

$$
\frac{\eta^*}{\eta} = \frac{\left(1 + yD_R D_O\right)i_E + \left(D_O + yD_R\right)\left[s_{2\varepsilon}a\left(s_{2\varepsilon}q_E + c_{2\varepsilon}u_E\right) - xc_{2\varepsilon}\left(1-2a\right)v_E\right]}{\left(1 + yD_T D_O\right)i_E + \left(D_O + yD_T\right)\left[s_{2\varepsilon}a\left(s_{2\varepsilon}q_E + c_{2\varepsilon}u_E\right) - xc_{2\varepsilon}\left(1-2a\right)v_E\right]}
$$
(162)

6 • Eq.(162) with a cleaned PBS (S.10.10) becomes Eq. (163), and without calibrator rotation $\varepsilon$

7 Eq.(162) becomes Eq. (164).

$$\gamma = 0, D_T = +1, D_R = -1 \Rightarrow$$

$$
\frac{\eta^*}{\eta} = \frac{1 - yD_O}{1 + yD_O} \frac{i_E - y\left[s_{2\varepsilon}a\left(s_{2\varepsilon}q_E + c_{2\varepsilon}u_E\right) - xc_{2\varepsilon}\left(1-2a\right)v_E\right]}{i_E + y\left[s_{2\varepsilon}a\left(s_{2\varepsilon}q_E + c_{2\varepsilon}u_E\right) - xc_{2\varepsilon}\left(1-2a\right)v_E\right]}
$$
(163)

$$\gamma = \varepsilon = 0 \Rightarrow$$

$$
\frac{\eta^*}{\eta} = \frac{\left(1 + yD_R D_O\right)i_E - x\left(D_O + yD_R\right)\left(1-2a\right)v_E}{\left(1 + yD_T D_O\right)i_E - x\left(D_O + yD_T\right)\left(1-2a\right)v_E}
$$
(164)



Atmospheric
Measurement
Techniques


Discussions

• With a cleaned analyser and without calibrator rotation $\varepsilon$ the gain ratios in Eq.(162) become
Eq. (165) and for the $\Delta 90$-calibration Eq. (166).

$$\gamma = \varepsilon = 0, D_T = +1, D_R = -1 \Rightarrow$$

$$\frac{\eta^*}{\eta} = \frac{1 - yD_O}{1 + yD_O} \frac{i_E + xy(1 - 2a)v_E}{i_E - xy(1 - 2a)v_E} \qquad (165)$$

$$\frac{\eta^*_{\Delta 90}}{\eta} = \frac{1 - yD_O}{1 + yD_O} \qquad (166)$$

• Eq. (165) can be rearranged with Eq. (166) to Eq. (167), from which we get the degree of
circular polarisation $v_E / i_E$ of the beam behind the emitter optics in Eq. (168). The atmospheric
polarisation parameter $a$ must be estimated from a standard measurement, and if we use an
atmospheric range without aerosols it becomes $a \approx 1$. While $v_{in}$ in Eq. (158) includes the
mostly unknown retardation terms of the receiving optics, $v_E$ in Eq. (168) is free of them and
hence a better estimation for the elliptical polarisation of the laser.

$$\gamma = \varepsilon = 0, D_T = +1, D_R = -1 \Rightarrow$$

$$\frac{\eta^*}{\eta^*_{\Delta 90}} = \frac{i_E + xy(1 - 2a)v_E}{i_E - xy(1 - 2a)v_E} \Rightarrow \qquad (167)$$

$$\frac{v_E}{i_E} = \frac{1}{xy(1 - 2a)} \frac{\eta^* - \eta^*_{\Delta 90}}{\eta^* + \eta^*_{\Delta 90}} \qquad (168)$$

**9.3   Calibration with an ideal λ/4 plate behind the emitter optics**

$$I_S = \eta_S \mathbf{M}_S \mathbf{R}(\mathrm{y}) \mathbf{M}_O(\gamma) \mathbf{F}(a) \mathbf{M}_{QW}(\mathrm{x}45° + \varepsilon) \mathbf{M}_E(\beta) I_L(\alpha) =$$
$$= \eta_S \mathbf{A}_S(\mathrm{y}, \gamma, a) \mathbf{M}_{QW}(\mathrm{x}45° + \varepsilon) I_{in}(\beta, \alpha) \qquad (169)$$

With Eq. (D.13) for the analyser part $\mathbf{A}_S$, an ideal λ/4 plate $\mathbf{M}_{QW}$ Eq. (S.10.17.2), and with an
input Stokes vector $I_{in}$ from Eq.(E.8) we get the general calibration signals $I_S$ in Eq. (170).



$$\frac{I_S}{\eta_S T_S T_O F_{11} T_{QW} T_E I_L} = \frac{\langle \mathbf{M}_S \mathbf{R}_y \mathbf{M}_O(\gamma) \mathbf{F}(a) | \mathbf{M}_{QW}(\text{x}45° + \varepsilon) | \mathbf{M}_E \boldsymbol{I}_L \rangle}{T_S T_O F_{11} T_{QW} T_E I_L} =$$

$$= \left\langle \begin{matrix} 1 + \text{yc}_{2\gamma} D_S D_O \text{c} \\ a \left[ \text{c}_{2\gamma} D_O + \text{y} D_S \left( 1 - \text{s}_{2\gamma}^2 W_O \right) \right] \\ -a\text{s}_{2\gamma} \left( D_O + \text{yc}_{2\gamma} D_S W_O \right) \\ -(1-2a)\text{ys}_{2\gamma} D_S Z_O \text{s}_O \end{matrix} \middle| \begin{matrix} 1 & 0 & 0 & 0 \\ 0 & \text{s}_{2\varepsilon}^2 & -\text{s}_{2\varepsilon}\text{c}_{2\varepsilon} & -\text{xc}_{2\varepsilon} \\ 0 & -\text{s}_{2\varepsilon}\text{c}_{2\varepsilon} & \text{c}_{2\varepsilon}^2 & -\text{xs}_{2\varepsilon} \\ 0 & \text{xc}_{2\varepsilon} & \text{xs}_{2\varepsilon} & 0 \end{matrix} \middle| \begin{matrix} i_E \\ q_E \\ u_E \\ v_E \end{matrix} \right\rangle$$

(170)

• Special cases: Equivalent to Sect.9.2 we get from Eq. (170) without receiver optics rotation
$\gamma$ the calibration signals in Eq. (171).

with $\gamma = 0 \Rightarrow$

$$\frac{I_S}{\eta_S T_S T_O T_{QW} F_{11} T_E I_L} = \frac{\langle \mathbf{A}_S(\text{y},0,a) | \mathbf{M}_{QW}(\text{x}45° + \varepsilon) | \mathbf{M}_E \boldsymbol{I}_L \rangle}{T_S T_O T_{QW} F_{11} T_E I_L} = \left\langle \begin{matrix} 1 + \text{y} D_S D_O \\ \text{s}_{2\varepsilon}^2 a(D_O + \text{y} D_S) \\ -\text{c}_{2\varepsilon}\text{s}_{2\varepsilon} a(D_O + \text{y} D_S) \\ -\text{xc}_{2\varepsilon} a(D_O + \text{y} D_S) \end{matrix} \middle| \begin{matrix} i_E \\ q_E \\ u_E \\ v_E \end{matrix} \right\rangle$$

(171)

• From Eq. (171) without calibrator rotation $\varepsilon$ we get the gain ratios Eq. (172), • with
additionally a cleaned analyser we get Eq. (173), and with the corresponding $\Delta$90-calibration
Eq. (174).

with $\gamma = \varepsilon = 0 \Rightarrow$

$$\frac{\eta^*}{\eta} = \frac{(1 + \text{y} D_R D_O) i_E - \text{xa}(D_O + \text{y} D_R) v_E}{(1 + \text{y} D_T D_O) i_E - \text{xa}(D_O + \text{y} D_T) v_E}$$

(172)

with $\gamma = \varepsilon = 0, D_T = +1, D_R = -1 \Rightarrow$

$$\frac{\eta^*}{\eta} = \frac{1 - \text{y} D_O}{1 + \text{y} D_O} \frac{i_E + \text{xya} v_E}{i_E - \text{xya} v_E}$$

(173)

$$\frac{\eta^*_{\Delta 90}}{\eta} = \frac{1 - \text{y} D_O}{1 + \text{y} D_O}$$

(174)

• Eq. (173) can be rearranged with Eq. (174) to Eq. (175) from which we get the degree of
circular polarisation $v_E/i_E$ of the beam behind the emitter optics if the atmospheric polarisation
parameter $a$ is known, as e.g. when we use the lidar signals from an atmospheric range
without aerosols where $a \approx 1$.
$$\frac{\eta^*}{\eta^*_{\Delta 90}} = \frac{i_E + \text{xya} v_E}{i_E - \text{xya} v_E} \Rightarrow \frac{v_E}{i_E} = \frac{1}{\text{xya}} \frac{\eta^* - \eta^*_{\Delta 90}}{\eta^* + \eta^*_{\Delta 90}}$$

(175)



**10  Calibration with a circular polariser (CP)**
The use of a circular polariser seems to be ideal for the calibration, but the uncertainties of a
real circular polariser are usually not provided by manufacturers and might be difficult to
determine. A real CP is mostly a combination of a linear polariser followed by a QWP at z45°
($z = \pm 1$) (see S.10.18), and therefore it combines the uncertainties of both (see Sects. 8 and 9).
Before the results of a circularly polarising calibrator can be trusted, the diattenuation of the
linear polariser and the phase shift uncertainties should be determined and the error
assessment performed using the general Eq. (C.10) for the calibration signals. If we consider
all possible error terms, the Müller matrix for a real CP becomes too complex for this
investigation, wherefore we assume a circular polariser with phase shift error ω but with an
ideal linear polariser from Eq. (S.10.18.4) in the following in order to show the possibilities of
this calibrator.
**10.1  Calibration with a circular polariser before the polarising beam-splitter**

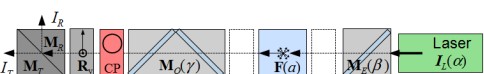

$$\mathbf{I}_S = \eta_S \mathbf{M}_S \mathbf{R}_y \mathbf{M}_{CP}(z, x45° + \varepsilon, \omega) \mathbf{M}_O(\gamma) \mathbf{F}(a) \mathbf{M}_E(\beta) \mathbf{I}_L(\alpha) =$$
$$= \eta_S \mathbf{A}_S(y) \mathbf{M}_{CP}(z, x45° + \varepsilon, \omega) \mathbf{I}_{in}(\gamma, a, \beta, \alpha)$$
(176)

With $\mathbf{A}_S$ from Eq. (D.5), the circularly polarising calibrator $\mathbf{M}_{CP}$ with retardation error $\omega$ from
Eq. (S.10.18.4), and the input Stokes vectors $\mathbf{I}_{in}$ from App. E.4 we get Eq. (177) for the
calibration signals $I_S$. As the last term of Eq. (177) is independent of $D_S$, it cancels out in the
gain ratios Eq. (178), which is therefore independent of the input Stokes vector, but still
includes $\varepsilon$ and $\omega$ terms.

$$\frac{I_S}{\eta_S T_S T_{CP} I_{in}} = \frac{\langle \mathbf{M}_S \mathbf{R}_y | \mathbf{M}_{CP}(z, x45° + \varepsilon, \omega) | \mathbf{I}_{in} \rangle}{T_S T_{CP} I_{in}} =$$

$$= \left\langle \begin{array}{c} 1 \\ yD_S \\ 0 \\ 0 \end{array} \middle\| \begin{array}{c} 1 \\ xs_{2\varepsilon}s_\omega \\ -xc_{2\varepsilon}s_\omega \\ zc_\omega \end{array} \right\rangle \left\langle \begin{array}{c} 1 \\ -xs_{2\varepsilon} \\ xc_{2\varepsilon} \\ 0 \end{array} \middle\| \begin{array}{c} i_{in} \\ q_{in} \\ u_{in} \\ v_{in} \end{array} \right\rangle = [1 + xyD_S s_{2\varepsilon} s_\omega] \left\langle \begin{array}{c} 1 \\ -xs_{2\varepsilon} \\ xc_{2\varepsilon} \\ 0 \end{array} \middle\| \begin{array}{c} i_{in} \\ q_{in} \\ u_{in} \\ v_{in} \end{array} \right\rangle =$$

$$= (1 + xyD_S s_{2\varepsilon} s_\omega) [i_{in} - x(s_{2\varepsilon} q_{in} - c_{2\varepsilon} u_{in})]$$
(177)



$$\frac{\eta^*}{\eta} = \frac{1 + xyD_R s_{2\varepsilon} s_\omega}{1 + xyD_T s_{2\varepsilon} s_\omega} \tag{178}$$
• If $\omega$ is zero, we have an ideal circular polariser with which we get the gain ratio
independently of $\varepsilon$, and if $\varepsilon$ is zero $\omega$ doesn't matter (Eq. (179)).

$$\omega = 0 \vee \varepsilon = 0 \Rightarrow$$

$$\frac{\eta^*}{\eta} = 1 \tag{179}$$
• With a cleaned analyser we get from Eq. (178) Eqs. (180) and (181), which show that the
deviations of the gain ratios are fully compensated by the Δ90-calibration. $\omega$ can be
determined by means of the successive approximation in Sect. 11, Eqs. (198) ff.

$$D_T = +1, D_R = -1 \Rightarrow$$

$$\frac{\eta^*}{\eta} = \frac{1 - xy s_{2\varepsilon} s_\omega}{1 + xy s_{2\varepsilon} s_\omega} \tag{180}$$
$$\frac{\eta^*_{\Delta 90}}{\eta} = 1 \tag{181}$$
**10.2  Calibration with a circular polariser before the receiving optics**
$$\begin{aligned} \boldsymbol{I}_S &= \eta_S \mathbf{M}_S \mathbf{R}_y \mathbf{M}_O(\gamma) \mathbf{M}_{CP}(z, x45° + \varepsilon, \omega) \mathbf{F}(a) \mathbf{M}_E(\beta) \boldsymbol{I}_L(\alpha) = \\ &= \eta_S \mathbf{A}_S(y, \gamma) \mathbf{M}_{CP}(z, x45° + \varepsilon, \omega) \boldsymbol{I}_{in}(a, \beta, \alpha) \end{aligned} \tag{182}$$
With $\mathbf{A}_S$ from App. D.2, $\mathbf{M}_{CP}$ with retardation error $\omega$ from Eq. (S.10.18.4), and $\boldsymbol{I}_{in}$ from App.
E.3 we get Eq. (183) for the calibration signals $I_S$.



$$\frac{I_S}{\eta_S T_S T_{CP} T_O F_{11} T_E I_L} = \frac{\langle \mathbf{M}_S \mathbf{R}_y \mathbf{M}_O(\gamma) | \mathbf{M}_{CP}(z, x45° + \varepsilon, \omega) | \mathbf{F}(a) \mathbf{M}_E I_L \rangle}{T_S T_O T_{CP} F_{11} T_E I_L} =$$

$$1 = \left\langle \begin{matrix} 1 + yc_{2\gamma}D_S D_O \\ c_{2\gamma}D_O + yD_S\left(1 - s_{2\gamma}^2 W_O\right) \\ s_{2\gamma}\left(D_O + yc_{2\gamma}D_S W_O\right) \\ -ys_{2\gamma}D_S Z_O s_O \end{matrix} \middle\| \begin{matrix} 1 \\ xs_{2\varepsilon}s_\omega \\ -xc_{2\varepsilon}s_\omega \\ zc_\omega \end{matrix} \middle\rangle \left\langle \begin{matrix} 1 \\ -xs_{2\varepsilon} \\ xc_{2\varepsilon} \\ 0 \end{matrix} \middle\| \begin{matrix} i_E \\ aq_E \\ -au_E \\ (1-2a)v_E \end{matrix} \right\rangle =$$

$$= \left\{ \begin{matrix} 1 + yD_S\left(c_{2\gamma}D_O - s_{2\gamma}Z_O s_O zc_\omega\right) + \\ + xs_\omega\left[D_O s_{2\varepsilon-2\gamma} + yD_S\left(s_{2\varepsilon} - s_{2\gamma}W_O c_{2\varepsilon-2\gamma}\right)\right] \end{matrix} \right\} \left[ i_E - xa\left(s_{2\varepsilon}q_E + c_{2\varepsilon}u_E\right) \right] \tag{183}$$

As the last term of Eq. (183) is independent of $D_S$, it cancels out in the gain ratio. However,
as long as the receiver optics rotation $\gamma$ doesn't vanish, the gain ratios include deviations
which don't cancel with the Δ90-calibration, even if we • use a cleaned analyser (Eq. (184))
and • additionally an ideal circular polariser (Eq. (185)) or • without calibrator error ε (Eq.

6 (186)).

$D_T = +1, D_R = -1 \Rightarrow$

$$\frac{\eta^*}{\eta} = \frac{1 - y\left(c_{2\gamma}D_O - s_{2\gamma}Z_O s_O zc_\omega\right) + xs_\omega\left[D_O s_{2\varepsilon-2\gamma} - y\left(s_{2\varepsilon} - s_{2\gamma}W_O c_{2\varepsilon-2\gamma}\right)\right]}{1 + y\left(c_{2\gamma}D_O - s_{2\gamma}Z_O s_O zc_\omega\right) + xs_\omega\left[D_O s_{2\varepsilon-2\gamma} + y\left(s_{2\varepsilon} - s_{2\gamma}W_O c_{2\varepsilon-2\gamma}\right)\right]} \tag{184}$$

$D_T = +1, D_R = -1, \omega = 0 \Rightarrow$

$$\frac{\eta^*}{\eta} = \frac{1 - y\left(c_{2\gamma}D_O - s_{2\gamma}Z_O s_O\right)}{1 + y\left(c_{2\gamma}D_O - s_{2\gamma}Z_O s_O\right)} \tag{185}$$

$D_T = +1, D_R = -1, \varepsilon = 0 \Rightarrow$

$$\frac{\eta^*}{\eta} = \frac{1 - y\left(c_{2\gamma}D_O - s_{2\gamma}Z_O s_O zc_\omega\right) - xs_\omega s_{2\gamma}\left[D_O - yW_O c_{2\gamma}\right]}{1 + y\left(c_{2\gamma}D_O - s_{2\gamma}Z_O s_O zc_\omega\right) - xs_\omega s_{2\gamma}\left[D_O + yW_O c_{2\gamma}\right]} \tag{186}$$

• From Eq. (183) without receiver optics rotation $\gamma$ we get Eq. (187), and
• with additionally a cleaned analyser the Eqs. (188) and (189) are the same as in the previous
sections but with the prefactor of Eq. (189).
$\gamma = 0 \Rightarrow$

$$\frac{\eta^*}{\eta} = \frac{1 + yD_R D_O + xs_\omega s_{2\varepsilon}\left(D_O + yD_R\right)}{1 + yD_T D_O + xs_\omega s_{2\varepsilon}\left(D_O + yD_T\right)} \tag{187}$$

$\gamma = 0, D_T = +1, D_R = -1 \Rightarrow$

$$\frac{\eta^*}{\eta} = \frac{1 - yD_O}{1 + yD_O}\frac{1 - xys_\omega s_{2\varepsilon}}{1 + xys_\omega s_{2\varepsilon}} \tag{188}$$



$$\left[\gamma = 0, D_T = +1, D_R = -1\right] \wedge \left[\omega = 0 \vee \varepsilon = 0\right] \Rightarrow \frac{\eta^*}{\eta} = \frac{1 - yD_O}{1 + yD_O}$$ (189)
## 10.3 Calibration with a circular polariser behind the emitter optics



$$I_S = \eta_S \mathbf{M}_S \mathbf{R}_y \mathbf{M}_O(\gamma) \mathbf{F}(a) \mathbf{M}_{CP}(z, x45° + \varepsilon) \mathbf{M}_E(\beta) \boldsymbol{I}_L(\alpha) =$$
$$= \eta_S \mathbf{A}_S(y, \gamma, a) \mathbf{M}_{CP}(z, x45° + \varepsilon) \boldsymbol{I}_E(\beta, \alpha)$$ (190)

With $\mathbf{A}_S$ from App. D.3, $\mathbf{M}_{CP}$ with retardation error $\omega$ from Eq. (S.10.18.4), and $\boldsymbol{I}_{in}$ from App.
E.2 we get Eq. (191) for the calibration signals $I_S$, which differs from Eq. (183) in the last
section just by the prefactors depending on the atmospheric polarisation parameter $a$. The
same holds for the gain ratios derived with a • cleaned analyser in Eqs. (192) compared to Eq.
(184) and all the subsequent derivations there.
$$\frac{I_S}{\eta_S T_S T_O F_{11} T_{CP} T_E I_L} = \frac{\langle \mathbf{M}_S \mathbf{R}_y \mathbf{M}_O(\gamma) \mathbf{F}(a) | \mathbf{M}_{CP}(z, x45° + \varepsilon, \omega) | \mathbf{M}_E \boldsymbol{I}_L \rangle}{T_S T_O F_{11} T_{CP} T_E I_L} =$$

$$= \left\langle \begin{matrix} 1 + yc_{2\gamma} D_S D_O c \\ a\left[c_{2\gamma} D_O + yD_S\left(1 - s_{2\gamma}^2 W_O\right)\right] \\ -as_{2\gamma}\left(D_O + yc_{2\gamma} D_S W_O\right) \\ -(1 - 2a) ys_{2\gamma} D_S Z_O s_O \end{matrix} \middle\| \begin{matrix} 1 \\ xs_{2\varepsilon} s_\omega \\ -xc_{2\varepsilon} s_\omega \\ zc_\omega \end{matrix} \right\rangle \left\langle \begin{matrix} 1 \\ -xs_{2\varepsilon} \\ xc_{2\varepsilon} \\ 0 \end{matrix} \middle\| \begin{matrix} i_E \\ q_E \\ u_E \\ v_E \end{matrix} \right\rangle =$$

$$= \left\{ \begin{matrix} 1 + yD_S\left(c_{2\gamma} D_O - (1 - 2a)s_{2\gamma} Z_O s_O zc_\omega\right) + \\ + xas_\omega\left[D_O s_{2\varepsilon - 2\gamma} + yD_S\left(s_{2\varepsilon} - s_{2\gamma} W_O c_{2\varepsilon - 2\gamma}\right)\right] \end{matrix} \right\} \left[i_E - x\left(s_{2\varepsilon} q_E + c_{2\varepsilon} u_E\right)\right]$$ (191)

$$D_T = +1, D_R = -1 \Rightarrow$$

$$\frac{\eta^*}{\eta} = \frac{1 - y\left(c_{2\gamma} D_O - s_{2\gamma}(1 - 2a) Z_O s_O zc_\omega\right) + xas_\omega\left[D_O s_{2\varepsilon - 2\gamma} - y\left(s_{2\varepsilon} - s_{2\gamma} W_O c_{2\varepsilon - 2\gamma}\right)\right]}{1 + y\left(c_{2\gamma} D_O - s_{2\gamma}(1 - 2a) Z_O s_O zc_\omega\right) + xas_\omega\left[D_O s_{2\varepsilon - 2\gamma} + y\left(s_{2\varepsilon} - s_{2\gamma} W_O c_{2\varepsilon - 2\gamma}\right)\right]}$$ (192)
## 11 Determination of the calibrator rotation ε
The calibration measurements can be used to determine and consequentially correct the
calibrator rotation $\varepsilon$, which is especially important for the rotation calibrator (Sect. 7), because
here the rotation error $\varepsilon$ is also present in the standard atmospheric measurements and has to
be corrected, either mechanically before the measurements or analytically after the





measurements. If the ±45° calibration measurements can be described or approximated by Eq.
(193) with $f(y,...)$ being a function of any parameter but not of x and $\varepsilon$, it is possible to
estimate the calibrator rotation $\varepsilon$ by means of the relative difference of the ±45° gain ratios as
in Eq. (194) and using the tangent half-angle substitution (S.12.1) to achieve $\varepsilon$ from Eq. (195).
Note: $\eta$ is assumed to be unknown.

$$\frac{\eta^*}{\eta} = f(y,...)\frac{1+xs_{2\varepsilon}}{1-xs_{2\varepsilon}} \tag{193}$$

$$Y(\varepsilon) \equiv \frac{\eta^*(y,+45°+\varepsilon)-\eta^*(y,-45°+\varepsilon)}{\eta^*(y,+45°+\varepsilon)+\eta^*(y,-45°+\varepsilon)} = \frac{\dfrac{1+s_{2\varepsilon}}{1-s_{2\varepsilon}}-\dfrac{1-s_{2\varepsilon}}{1+s_{2\varepsilon}}}{\dfrac{1+s_{2\varepsilon}}{1-s_{2\varepsilon}}+\dfrac{1-s_{2\varepsilon}}{1+s_{2\varepsilon}}} = \frac{2s_{2\varepsilon}}{1+s_{2\varepsilon}^2} \tag{194}$$

$$\varepsilon(Y) = 0.5 * \arcsin\left[\tan\left(0.5 * \arcsin\left[Y\right]\right)\right] \tag{195}$$

With the assumption $\sin(2\varepsilon) << 1$ we get a good approximation for $\varepsilon$ in the simple Eq. (196),
which deviates by about 5% at $\varepsilon \approx 6°$ and $Y(\varepsilon) \approx 0.4$.

$$s_{2\varepsilon} \ll 1 \Rightarrow \quad Y(\varepsilon) \approx 2s_{2\varepsilon} \Rightarrow \quad \varepsilon \approx 0.25 * Y \tag{196}$$

Eq. (193) is applicable in Eqs. (130) and (137) for the linear polariser calibrator, and it is a
good approximation for Eq (144) if the atmospheric polarisation parameter $a \approx 1$. For the
rotation calibration before the receiving optics (Sect. 7.2, Eq.(117)) we have to assume that $a$
$\approx 1$ and additionally that the laser beam behind the emitter optics is horizontal-linearly
polarised. Eq.(117) can then be approximated by Eq. (197).

with $\gamma = 0, D_T = +1, D_R = -1, i_E = q_E = 1, u_E = v_E = 0, a \approx 1 \Rightarrow$
$$\frac{\eta^*}{\eta} \approx \frac{1-yD_O}{1+yD_O}\frac{1+xs_{2\varepsilon}}{1-xs_{2\varepsilon}} \tag{197}$$

If instead of Eq.(193) we have a form as Eq. (198) (see Sect. S.12.1 ), we get Eqs. (199) and
(200). If $\varepsilon$ is known, Eq.(200) can be solved for $K$, which yields Eq. (201).

$$\frac{\eta^*}{\eta} = f(y,...)\frac{1+Kxs_{2\varepsilon}}{1-Kxs_{2\varepsilon}} \quad \text{with} \quad K \leq 1 \tag{198}$$

$$Y(\varepsilon, K) \equiv \frac{\eta^*(y,+45°+\varepsilon,K)-\eta^*(y,-45°+\varepsilon,K)}{\eta^*(y,+45°+\varepsilon,K)+\eta^*(y,-45°+\varepsilon,K)} = \frac{2Ks_{2\varepsilon}}{1+K^2s_{2\varepsilon}^2} \tag{199}$$



$$\varepsilon = \frac{1}{2}\arcsin\left[\frac{1}{K}\tan\left(\frac{\arcsin\left[Y(\varepsilon,K)\right]}{2}\right)\right] \qquad (200)$$
$$K = \left[\frac{1}{\sin 2\varepsilon}\tan\left(\frac{\arcsin\left[Y(\varepsilon,K)\right]}{2}\right)\right] \qquad (201)$$
If the true $\varepsilon$ and $K$ are unknown, we can retrieve them by successive approximation. With $K <$
1 we find as a first approximation $\varepsilon_1$ from Eq. (202) and make the next calibration
measurement after adjusting the calibrator rotation by $-\varepsilon_1$, which results in the actual position
$(\varepsilon - \varepsilon_1)$ and the corresponding Eq. (203).
$$\varepsilon_1 = \frac{1}{2}\arcsin\left[\tan\left(\frac{\arcsin\left[Y(\varepsilon,K)\right]}{2}\right)\right] < \varepsilon \qquad (202)$$
$$Y(\varepsilon - \varepsilon_1, K) = \frac{2Ks_{2(\varepsilon-\varepsilon_1)}}{1 + K^2 s^2_{2(\varepsilon-\varepsilon_1)}} \qquad (203)$$
Using the calibration measurements at the two positions $\varepsilon$ and $(\varepsilon - \varepsilon_1)$ with Eqs.(199) and
(203), we get an estimation of the true $\varepsilon$ with Eq. (205) derived from the ratio in Eq. (204).
$$\frac{Y(\varepsilon - \varepsilon_1, K)}{Y(\varepsilon, K)} = \frac{\left(1 + K^2 s^2_{2\varepsilon}\right)2Ks_{2(\varepsilon-\varepsilon_1)}}{\left(1 + K^2 s^2_{2(\varepsilon-\varepsilon_1)}\right)2Ks_{2\varepsilon}} \approx \frac{s_{2(\varepsilon-\varepsilon_1)}}{s_{2\varepsilon}} \approx \frac{(\varepsilon - \varepsilon_1)}{\varepsilon} = 1 - \frac{\varepsilon_1}{\varepsilon} \qquad (204)$$
$$\varepsilon \approx \frac{Y(\varepsilon, K)}{Y(\varepsilon, K) - Y(\varepsilon - \varepsilon_1, K)}\varepsilon_1 \qquad (205)$$
Finally, with known $\varepsilon$, we can use Eq. (201) to estimate $K$.
**12  Determination of the rotation $\alpha$ of the plane of polarisation of the emitted**
**laser beam.**
The orientation of the plane of polarisation of the laser beam is in general specified by
manufacturers just as *vertical* or *horizontal*, without specifying the reference and the
accuracy. Furthermore, the assembly of the laser with the telescope and the receiver optics in
a lidar system can often not be done with similar accuracy as the assembly of the optical
elements in the receiver optics, and the necessary alignment mechanisms for the tilt between
the laser and telescope axes additionally introduces variability and uncertainty. On top of that,





the adjustments may change after every laser maintenance. Therefore it is desirable to
determine the laser rotation once in a while.
Using the calibrator equations for the calibrator before the receiver optics from App. C with
an analyser without receiver optics rotation ($\gamma = 0$; Eq. (D.8)), i.e.

$$\gamma = 0° \Rightarrow \langle \mathbf{A}_S | (y,0°) = \langle \mathbf{M}_S \mathbf{R}_y | \mathbf{M}_O(0°) = T_O T_S \langle 1 + y D_S D_O \quad D_O + y D_S \quad 0 \quad 0 | =$$
$$= \langle \mathbf{M}_{SyO}(0°) | = T_{SyO} \langle 1 \quad D_{SyO} \quad 0 \quad 0 |$$
$$\Rightarrow A_S^3 = A_S^4 = 0$$
,

with elliptically polarised emitted laser light as Eq. (E.25),

$$\frac{\mathbf{I}_{in}(a,b,\alpha)}{I_{in}} = \frac{\mathbf{F}(a)\mathbf{I}_E}{F_{11} T_E I_L} = \left| 1 \quad ab\mathrm{c}_{2\alpha} \quad -ab\mathrm{s}_{2\alpha} \quad (1-2a)\sqrt{1-b^2} \right\rangle ,$$

and with ideal calibrators, we get the signals for the four ideal calibrator types in Eqs. (206)
to (209).

$$\frac{I_S}{\eta_S I_{in}} = \frac{\langle \mathbf{A}_S | \mathbf{M}_{rot}(\mathrm{x}45° + \varepsilon, \mathrm{h}) | \mathbf{I}_{in} \rangle}{I_{in}} =$$
$$= A_S^1 i_{in} + A_S^4 \mathrm{h} v_{in} - \mathrm{x} \left[ \left( \mathrm{s}_{2\varepsilon} A_S^2 - \mathrm{c}_{2\varepsilon} A_S^3 \right) q_{in} + \left( \mathrm{c}_{2\varepsilon} A_S^2 + \mathrm{s}_{2\varepsilon} A_S^3 \right) \mathrm{h} u_{in} \right] =$$
$$= T_{SyO} \left( 1 - \mathrm{x} ab D_{SyO} \mathrm{s}_{2\varepsilon - \mathrm{h}2\alpha} \right)$$
(206)

$$D_P = 1 \Rightarrow$$
$$\frac{I_S}{\eta_S T_P I_{in}} = \frac{\langle \mathbf{A}_S | \mathbf{M}_P(\mathrm{x}45° + \varepsilon) | \mathbf{I}_{in} \rangle}{T_P I_{in}} = \left[ A_S^1 + \mathrm{x} \left( \mathrm{c}_{2\varepsilon} A_S^3 - \mathrm{s}_{2\varepsilon} A_S^2 \right) \right] \left[ i_{in} + \mathrm{x} \left( \mathrm{c}_{2\varepsilon} u_{in} - \mathrm{s}_{2\varepsilon} q_{in} \right) \right] =$$
$$= T_{SyO} \left( 1 - \mathrm{x} \mathrm{s}_{2\varepsilon} D_{SyO} \right) \left( 1 - \mathrm{x} ab \mathrm{s}_{2\alpha + 2\varepsilon} \right) = T_{SyO} \left[ 1 + ab D_{SyO} \mathrm{s}_{2\varepsilon} \mathrm{s}_{2\alpha + 2\varepsilon} - \mathrm{x} \left( \mathrm{s}_{2\varepsilon} D_{SyO} + ab \mathrm{s}_{2\alpha + 2\varepsilon} \right) \right]$$
(207)

$$\omega = 0 \Rightarrow$$
$$\frac{I_S}{\eta_S T_{QW} I_{in}} = \frac{\langle \mathbf{A}_S | \mathbf{M}_{QW}(\mathrm{x}45° + \varepsilon, 0) | \mathbf{I}_{in} \rangle}{T_{QW} I_{in}} =$$
$$= A_S^1 i_{in} - \left( \mathrm{s}_{2\varepsilon} A_S^2 - \mathrm{c}_{2\varepsilon} A_S^3 \right) \left( \mathrm{s}_{2\varepsilon} q_{in} - \mathrm{c}_{2\varepsilon} u_{in} \right) - \mathrm{x} \left[ A_S^4 \left( \mathrm{c}_{2\varepsilon} q_{in} + \mathrm{s}_{2\varepsilon} u_{in} \right) + \left( \mathrm{c}_{2\varepsilon} A_S^2 + \mathrm{s}_{2\varepsilon} A_S^3 \right) v_{in} \right] =$$
$$= T_{SyO} \left[ 1 - ab D_{SyO} \mathrm{s}_{2\varepsilon} \mathrm{s}_{2\varepsilon + 2\alpha} + \mathrm{x} D_{SyO} \mathrm{c}_{2\varepsilon} (1-2a)\sqrt{1-b^2} \right]$$
(208)

$$\omega = 0, D_P = 1 \Rightarrow$$
$$\frac{I_S}{\eta_S T_{CP} I_{in}} = \frac{\langle \mathbf{A}_S | \mathbf{M}_{CP}(\mathrm{z}, \mathrm{x}45° + \varepsilon) | \mathbf{I}_{in} \rangle}{T_{CP} I_{in}} = \left( A_S^1 + \mathrm{z} A_S^4 \right) \left( i_{in} - \mathrm{x} \left( \mathrm{c}_{2\varepsilon} u_{in} - \mathrm{s}_{2\varepsilon} q_{in} \right) \right) =$$
$$= T_{SyO} \left( 1 + \mathrm{x} ab \mathrm{s}_{2\alpha + 2\varepsilon} \right)$$
(209)





Eqs. (206) and (209) are of the type of Eq. (193), wherefore the solutions described in Sect.
11 can be applied, but only to determine $\varepsilon \pm \alpha$. In order to determine $\alpha$ alone, $\varepsilon$ must be
known, or a series of measurements with variable $\varepsilon$ are fitted to the gain ratios $\eta^*$ formulated
with one of the Eqs. (206) to (209), as explained by Alvarez et al. (2006).
Furthermore, for the case of the linear polariser calibrator (Eq. (207)), an unpolarised light
source (i.e. $i_{in} = 1$ and $q_{in} = u_{in} = v_{in} = 0$ ) before the receiver optics / telescope gives Eq. (210)
from Eq. (207), which is of the type of Eq. (193), and with a cleaned analyser $D_{SyO} = \pm 1$.

$$D_P = 1, \;\; i_{in} = 1, \;\; q_{in} = u_{in} = v_{in} = 0 \Rightarrow$$

$$\frac{I_S}{\eta_S T_P I_{in}} = \frac{\langle \mathbf{A}_S | \mathbf{M}_P (x45° + \varepsilon) | \boldsymbol{I}_{in} \rangle}{T_P I_{in}} = T_{SyO} \left(1 - xs_{2\varepsilon} D_{SyO}\right) \qquad (210)$$

**13   Summary and conclusions**
The presented equations can be used to analyse the effects of polarising optics of a variety of
lidar systems and to assess the accuracy and error of several calibration techniques. Major
findings are, that a cleaned analyser and no rotation of the receiving optics with respect to the
laser polarisation avoid many error terms and allow to determine and correct other
misalignments and the optics diattenuation, and that the $\Delta90$-calibration can decrease the error
of a single $\pm45°$ calibration into insignificance.
We showed that a linear polariser as calibrator should have a very good extinction ratio in
order to avoid large calibration errors (Eq. (126)). The advantage of a sheet polariser (and $\lambda/4$
sheet filters) is its tenuity, wherefore it can be included in many existing lidar systems with
minimal space requirement, for example with a sheet holder as shown in Fig. 4. Such a sheet
holder guarantees an accurate $\Delta90°$ rotation of the sheet, wherefore the absolute accuracy of
the 45° orientation is not important. Together with an existing calibration technique or
inserted at different positions, the filter holder can be used to determine the diattenuation of
the optics between the two positions (see Eqs. (131) and (138) / (145)). Furthermore, the
determination of the calibration factor with an ideal linear polariser calibrator is always
independent of changes of the input light and hence independent of the atmospheric
depolarisation, in contrast to the other calibrators. Plastic sheet filters can easily be cut to be
used in a rotation holder as in Fig. 5, so that the filter can be automatically rotated to $\Delta90°$
positions and out of the optical path for standard measurements. Large acceptance angles of
linearly polarising sheet filters allows the mounting close to the telescope focus where we



have some free space and the filter diameter and mechanical mounting can be small due to the
small beam diameter. However, it should be considered that the direction of the polarising
structure of a sheet filter is not necessarily constant over the whole sheet, which is usually not
specified by the manufacturers and should be inquired before the purchase.
$\lambda/4$ plates and circular polariser made of sheet films have similar constraints. Furthermore, the
$\Delta 90$-calibration doesn't work with a $\lambda/4$ plate, because the $\pm 45°$ errors don't compensate (Eqs.
(154), (164)), but in exchange we can determine with it the amount of circular polarisation
(Eqs. (158) and (168), and S.14). In contrast to that, the ideal circular polariser calibration
does not depend on the rotation error $\varepsilon$ and the input light polarisation at all and doesn't need a
$\Delta 90$-calibration, but inherent errors of a real circular polariser, which usually are not
sufficiently specified by manufacturers, would be difficult to assess, and the resulting error
equations are complex.
While all optical calibrators exhibit wavelength dependency and have the disadvantage of
possible inhomogeneities over the surface and other optical errors as inaccurate phase shift or
cross-talk, the only possible error source of the mechanical rotation calibrator (Sect. 7) is the
accuracy of the rotation itself. Although more bulky, it is the most reliable calibrator if used
with a cleaned analyser and accurate $\Delta 90°$ rotation (Eq. (105)). It is independent of
wavelength, has no internal uncertainties, and is insensitive to temporal changes and
degradation.

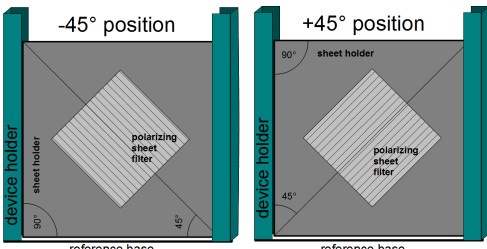

Figure 4: Simple holder for sheet filters (linear polariser or $\lambda/4$ plate) with accurate
positioning for the $\Delta 90$-calibration.



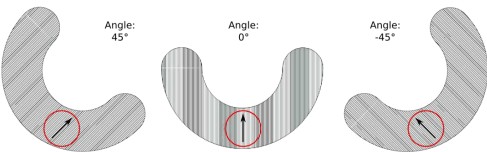

Figure 5: Linearly polarising sheet filter cutout for use in a rotation mount. The optical axis of
the filtered light beam is in the centre of the red circle. Reproduced with permission from
Kölbl (2010).

## 14  Acknowledgments

The financial support for EARLINET in the ACTRIS Research Infrastructure Project by the
European Union's Horizon 2020 research and innovation programme under grant agreement
n. 654169 and previously under grant agreement n. 262254 in the 7thFramework Programme
(FP7/2007-2013) is gratefully acknowledged.





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



**15 Appendix**
**App. A Acronyms and shortcuts**
*a*              polarisation parameter of the atmospheric volume; see Eq. 9
$a_L$            polarisation parameter of the light beam leaving the laser
$a'$             $a' = aa_L$, combined laser-atmosphere polarisation parameter
$\alpha$         Rotation of the plane of horizontal-linear polarisation of the laser around the z-
7                   axis (laser rotation)
$\beta$          Rotation of the emitter optics around the z-axis
$\gamma$         Rotation of the receiver optics around the z-axis
$c_{2\varepsilon}$   $\cos(2\varepsilon)$
$\delta$         *(*volume) linear depolarisation ratio of the atmospheric scattering volume; see
12                  Eq. 12
$\delta^*$       calibrated signal ratio including cross talk and alignment errors
*D*              diattenuation parameter. See
$\varepsilon$    error angle of the Δ90-calibration setup
$\eta_{T,R}$     electronic amplification of individual transmitted/reflected channels
$\eta$           $\eta = \eta_R T_R / \eta_T T_T$ calibration factor including only the electronic amplification
18                  and the optical diattenuation of the polarising beam-splitter
$\eta^*$         gain ratio i.e. the measured, apparent calibration factor $\eta^*$ of the polarisation
20                  channels, i.e. the calibration factor $\eta$ including the cross talk from optics before
21                  the polarising beam-splitter and from system alignment errors
$\eta^*_{\Delta 90}$   Δ90-gain   ratio   $\eta^*_{\Delta 90} \equiv \sqrt{\eta^*(+45°+\varepsilon)\eta^*(-45°+\varepsilon)}$;   measured,   apparent
23                  calibration factor retrieved with the Δ90-calibration method
*I*              Power/flux of the light beam [watt/lumen] (colloquially: intensity)
**I**            Stokes vector of the light beam
LDR              linear depolarisation ratio = $\delta$
**F**            Müller matrix of the atmospheric scattering volume in backscattering direction
$F_{ij}$         Element *ij* of **F**
$\mathbf{M}_S, \mathbf{M}_{T,R}$   Müller matrix of the polarising beam-splitter $_S$ , e.g a polarising beam-splitter
30                  cube, in the transmission $_T$ and reflection $_R$ path.
PBS              polarising beam-splitter
$s_{2\varepsilon}$   $\sin(2\varepsilon)$
$T_S$            Transmission of matrix $\mathbf{M}_S$ for unpolarised light (alias average transmission)
$T^p, T^s, R^p, R^s$   Intensity transmission and reflection coefficients of the polarising beam-
35                  splitter for parallel *p* and perpendicular *s* linearly polarised light with respect to
36                  the plane of incidence.
$Z_O$            $Z_O = \sqrt{1 - D_O^2}$
$W_O$            $W_O = 1 - Z_O c_O = 1 - c_O \sqrt{1 - D_O^2}$
$c_O$            $\cos(\Delta_O)$
$\Delta$         differential phase shift of the *p* and *s* polarised light  $\varphi^p - \varphi^s$
$\varphi^p \varphi^s$   phase of the *p* and *s* polarised light
$\psi$           Rotation of the calibrator around the z-axis
$\phi$           Rotation around z-axis
$\langle |$      First row vector of a matrix; bra-vector.



$| \, \rangle$            Stokes vector; always a column vector; ket-vector.
Setup parameters:
h                binary operator to select either manual rotation (h = +1) or rotation by means
4                     of a λ/2 plate (h = −1).
x, z             binary operators to select angles of +45° (x, z = +1) or –45° (x, z = −1)
y                binary operator to select angles of +0° (y = +1) or +90° (y = −1)
**App. B The <bra|ket> notation**
Superscript $^T$ means the transposition of a row vector to a column vector and vice versa, while
the |ket> and <bra| vector symbols always stand for a column vector and row vector,
respectively. That means:

$$\begin{pmatrix} a \\ b \\ c \\ d \end{pmatrix} = \begin{pmatrix} a & b & c & d \end{pmatrix}^T = \left| a \quad b \quad c \quad d \right\rangle = \left| \begin{matrix} a \\ b \\ c \\ d \end{matrix} \right\rangle$$
(B.1)

are forms of column vectors, and

$$\begin{pmatrix} a & b & c & d \end{pmatrix} = \begin{pmatrix} a \\ b \\ c \\ d \end{pmatrix}^T = \left\langle a \quad b \quad c \quad d \right| = \left\langle \begin{matrix} a \\ b \\ c \\ d \end{matrix} \right|$$
(B.2)

are forms of row vectors.
**App. C The calibration equation**
The general equation for the calibration signals Eq. (81) can be written similar to Kaul et al.
(2004) using general expressions for the analyser row vector $\langle \mathbf{A}_S|$ (see App. D) and for the
input Stokes vector $|\mathbf{I}_{in}\rangle$ (see App. E)) as in Eq. (C.1), irrespective of the actual position of the
calibrator.

$$\frac{I_S}{\eta_S} = \left\langle \mathbf{A}_S \left| \mathbf{C}(x45° + \varepsilon) \right| \mathbf{I}_{in} \right\rangle = I_{in} \left\langle \begin{matrix} A_S^1 \\ A_S^2 \\ A_S^3 \\ A_S^4 \end{matrix} \right| \mathbf{C}(x45° + \varepsilon) \left| \begin{matrix} i_{in} \\ q_{in} \\ u_{in} \\ v_{in} \end{matrix} \right\rangle$$
(C.1)

For certain setups the fully expanded equations are very complex. But sometimes slighty
expanded versions are sufficient to achieve significant insights. Demerging the (±45° + ε)





rotation from the calibrator, as in Eq. (C.2), or just the $\varepsilon$ – rotations, as in Eq. (C.3), and
applying the appropriate parts to the analyser and to the input Stokes vector can help to show
general relations. For this purpose we define the rotated analyser vector $<\mathbf{A}_{S,\varepsilon}|$ and the rotated
input Stokes vector $|\mathbf{I}_{in,\varepsilon}>$ as shown in Eq. (C.3).

$$I_S = \eta_S \left\langle \mathbf{A}_S \left| \mathbf{C}(\text{x}45°+\varepsilon) \right| \mathbf{I}_{in} \right\rangle =$$
$$= \eta_S \left\langle \mathbf{A}_S \left| \mathbf{R}(\text{x}45°+\varepsilon)\mathbf{C}(0)\mathbf{R}(-\text{x}45°-\varepsilon) \right| \mathbf{I}_{in} \right\rangle \Rightarrow$$

$$\frac{I_S}{\eta_S I_{in}} = \left\langle \begin{matrix} A_S^1 \\ A_S^2 \\ A_S^3 \\ A_S^4 \end{matrix} \middle| \begin{pmatrix} 1 & 0 & 0 & 0 \\ 0 & -\text{xs}_{2\varepsilon} & -\text{xc}_{2\varepsilon} & 0 \\ 0 & \text{xc}_{2\varepsilon} & -\text{xs}_{2\varepsilon} & 0 \\ 0 & 0 & 0 & 1 \end{pmatrix} \mathbf{C}(0) \begin{pmatrix} 1 & 0 & 0 & 0 \\ 0 & -\text{xs}_{2\varepsilon} & \text{xc}_{2\varepsilon} & 0 \\ 0 & -\text{xc}_{2\varepsilon} & -\text{xs}_{2\varepsilon} & 0 \\ 0 & 0 & 0 & 1 \end{pmatrix} \middle| \begin{matrix} i_{in} \\ q_{in} \\ u_{in} \\ v_{in} \end{matrix} \right\rangle =$$

$$= \left\langle \begin{matrix} A_S^1 \\ \text{x}\left(c_{2\varepsilon}A_S^3 - s_{2\varepsilon}A_S^2\right) \\ -\text{x}\left(c_{2\varepsilon}A_S^2 + s_{2\varepsilon}A_S^3\right) \\ A_S^4 \end{matrix} \middle| \mathbf{C}(0) \middle| \begin{matrix} i_{in} \\ \text{x}\left(c_{2\varepsilon}u_{in} - s_{2\varepsilon}q_{in}\right) \\ -\text{x}\left(c_{2\varepsilon}q_{in} + s_{2\varepsilon}u_{in}\right) \\ v_{in} \end{matrix} \right\rangle = \left\langle \begin{matrix} A_S^1 \\ \text{x}A_{S,\varepsilon}^3 \\ -\text{x}A_{S,\varepsilon}^2 \\ A_S^4 \end{matrix} \middle| \mathbf{C}(0) \middle| \begin{matrix} i_{in} \\ \text{x}u_{in,\varepsilon} \\ -\text{x}q_{in,\varepsilon} \\ v_{in} \end{matrix} \right\rangle \qquad \text{(C.2)}$$

$$I_S = \eta_S \left\langle \mathbf{A}_S \left| \mathbf{C}(\text{x}45°+\varepsilon) \right| \mathbf{I}_{in} \right\rangle =$$
$$= \eta_S \left\langle \mathbf{A}_S \left| \mathbf{R}(+\varepsilon)\mathbf{C}(\text{x}45°)\mathbf{R}(-\varepsilon) \right| \mathbf{I}_{in} \right\rangle = \eta_S \left\langle \mathbf{A}_{S,\varepsilon} \left| \mathbf{C}(\text{x}45°) \right| \mathbf{I}_{in,\varepsilon} \right\rangle \Rightarrow$$

$$\frac{I_S}{\eta_S I_{in}} = \left\langle \begin{matrix} A_S^1 \\ A_S^2 \\ A_S^3 \\ A_S^4 \end{matrix} \middle| \begin{pmatrix} 1 & 0 & 0 & 0 \\ 0 & c_{2\varepsilon} & -s_{2\varepsilon} & 0 \\ 0 & s_{2\varepsilon} & c_{2\varepsilon} & 0 \\ 0 & 0 & 0 & 1 \end{pmatrix} \mathbf{C}(\text{x}45°) \begin{pmatrix} 1 & 0 & 0 & 0 \\ 0 & c_{2\varepsilon} & s_{2\varepsilon} & 0 \\ 0 & -s_{2\varepsilon} & c_{2\varepsilon} & 0 \\ 0 & 0 & 0 & 1 \end{pmatrix} \middle| \begin{matrix} i_{in} \\ q_{in} \\ u_{in} \\ v_{in} \end{matrix} \right\rangle =$$

$$= \left\langle \begin{matrix} A_S^1 \\ c_{2\varepsilon}A_S^2 + s_{2\varepsilon}A_S^3 \\ c_{2\varepsilon}A_S^3 - s_{2\varepsilon}A_S^2 \\ A_S^4 \end{matrix} \middle| \mathbf{C}(\text{x}45°) \middle| \begin{matrix} i_{in} \\ c_{2\varepsilon}q_{in} + s_{2\varepsilon}u_{in} \\ c_{2\varepsilon}u_{in} - s_{2\varepsilon}q_{in} \\ v_{in} \end{matrix} \right\rangle \equiv \left\langle \begin{matrix} A_S^1 \\ A_{S,\varepsilon}^2 \\ A_{S,\varepsilon}^3 \\ A_S^4 \end{matrix} \middle| \mathbf{C}(\text{x}45°) \middle| \begin{matrix} i_{in} \\ q_{in,\varepsilon} \\ u_{in,\varepsilon} \\ v_{in} \end{matrix} \right\rangle \qquad \text{(C.3)}$$

Note the exchange of places of $A^2{}_{S,\varepsilon}$ and $A^3{}_{S,\varepsilon}$ and of $q_{in,\varepsilon}$ and $u_{in,\varepsilon}$ between Eqs. (C.2) and (C.3).
**App. C.1    Calibration with a rotator**
From Eqs. (C.1), (C.3), and (S.10.15.2) we get the general calibration signals Eq. (C.4) with
analyser vectors $<\mathbf{A}|$ from App. D and input Stokes vectors $|\mathbf{I}_{in}>$ from App. E.

$$\frac{I_S}{\eta_S I_{in}} = \frac{\left\langle \mathbf{A}_S \left| \mathbf{M}_{rot}\left(\mathrm{x}45° + \varepsilon, \mathrm{h}\right)\right| \boldsymbol{I}_{in} \right\rangle}{I_{in}} =$$

$$= \left\langle \begin{matrix} A_S^1 \\ A_S^2 \\ A_S^3 \\ A_S^4 \end{matrix} \left| \begin{pmatrix} 1 & 0 & 0 & 0 \\ 0 & -\mathrm{xs}_{2\varepsilon} & -\mathrm{xhc}_{2\varepsilon} & 0 \\ 0 & \mathrm{xc}_{2\varepsilon} & -\mathrm{xhs}_{2\varepsilon} & 0 \\ 0 & 0 & 0 & \mathrm{h} \end{pmatrix} \right| \begin{matrix} i_{in} \\ q_{in} \\ u_{in} \\ v_{in} \end{matrix} \right\rangle =$$

$$= A_S^1 i_{in} + A_S^4 \mathrm{h} v_{in} - \mathrm{x}\left[\left(\mathrm{s}_{2\varepsilon}A_S^2 - \mathrm{c}_{2\varepsilon}A_S^3\right)q_{in} + \left(\mathrm{c}_{2\varepsilon}A_S^2 + \mathrm{s}_{2\varepsilon}A_S^3\right)\mathrm{h}u_{in}\right] =$$

$$= A_S^1 i_{in} + A_S^4 \mathrm{h} v_{in} + \mathrm{x}\left[\quad A_{S,\varepsilon}^3 q_{in} \quad - \quad A_{S,\varepsilon}^2 \mathrm{h}u_{in}\right]$$

(C.4)

**App. C.2    Calibration with a linear polariser**
From Eq. (C.3) and (S.10.7.1) we get the general calibration signals Eq. (C.5) with analyser
vectors <**A**| from App. D and input Stokes $|\boldsymbol{I}_{in}>$ vectors from App. E. With an ideal linear
polariser Eq. (C.5) reduces to Eq. (C.6).

$$I_S = \eta_S \left\langle \mathbf{A}_S \left| \mathbf{M}_P\left(\mathrm{x}45° + \varepsilon\right)\right| \boldsymbol{I}_{in}\right\rangle = \eta_S \left\langle \mathbf{A}_S \left| \mathbf{R}(+\varepsilon)\mathbf{M}_P\left(\mathrm{x}45°\right)\mathbf{R}(-\varepsilon)\right| \boldsymbol{I}_{in}\right\rangle \Rightarrow$$

$$\frac{I_S}{\eta_S T_P I_{in}} = \left\langle \begin{matrix} A_S^1 \\ A_{S,\varepsilon}^2 \\ A_{S,\varepsilon}^3 \\ A_S^4 \end{matrix} \left| \begin{pmatrix} 1 & 0 & \mathrm{x}D_P & 0 \\ 0 & Z_P \mathrm{c}_P & 0 & -\mathrm{x}Z_P \mathrm{s}_P \\ \mathrm{x}D_P & 0 & 1 & 0 \\ 0 & \mathrm{x}Z_P \mathrm{s}_P & 0 & Z_P \mathrm{c}_P \end{pmatrix} \right| \begin{matrix} i_{in} \\ q_{in,\varepsilon} \\ u_{in,\varepsilon} \\ v_{in} \end{matrix} \right\rangle =$$

$$= \left\{ \begin{matrix} A_S^1 i_{in} + A_{S,\varepsilon}^3 u_{in,\varepsilon} + Z_P \mathrm{c}_P\left(A_{S,\varepsilon}^2 q_{in,\varepsilon} + A_S^4 v_{in}\right) + \\ +\mathrm{x}\left[D_P\left(A_S^1 u_{in,\varepsilon} + A_{S,\varepsilon}^3 i_{in}\right) - Z_P \mathrm{s}_P\left(A_{S,\varepsilon}^2 v_{in} - A_S^4 q_{in,\varepsilon}\right)\right] \end{matrix} \right\}$$

(C.5)

$$D_P = 1, Z_P = 0 \Rightarrow$$

$$\frac{I_S}{\eta_S T_P I_{in}} = A_S^1 i_{in} + A_{S,\varepsilon}^3 u_{in,\varepsilon} + \mathrm{x}\left(A_S^1 u_{in,\varepsilon} + A_{S,\varepsilon}^3 i_{in}\right) = \left(A_S^1 + \mathrm{x}A_{S,\varepsilon}^3\right)\left(i_{in} + \mathrm{x}u_{in,\varepsilon}\right)$$

(C.6)

**App. C.3    Calibration with a λ/4 plate (QWP)**
From Eq. (C.2) and Eq.(S.10.11.1) for the λ/4-plate with retardation error ω as in Eq. (C.7) we
get the general calibration signals Eq. (C.8) with an analyser vectors <**A**| from App. D and
input Stokes vectors $|\boldsymbol{I}_{in}>$ from App. E.

$$\varDelta_{QW} = 90° + \omega \Rightarrow \mathrm{c}_{QW} = -\mathrm{s}_\omega, \ \mathrm{s}_{QW} = \mathrm{c}_\omega$$

(C.7)



$$\frac{I_S}{\eta_S T_{QW} I_{in}} = \frac{\langle \mathbf{A}_S | \mathbf{M}_{QW}(\mathrm{x}45°+\varepsilon,\omega) | \mathbf{I}_{in} \rangle}{T_{QW} I_{in}} =$$

$$= \frac{\langle \mathbf{A}_S | \mathbf{R}(\mathrm{x}45°+\varepsilon) \mathbf{M}_{QW}(0,\omega) \mathbf{R}(-\mathrm{x}45°-\varepsilon) | \mathbf{I}_{in} \rangle}{T_{QW} I_{in}} =$$

$$= \left\langle \begin{array}{c} A_S^1 \\ \mathrm{x} A_{S,\varepsilon}^3 \\ -\mathrm{x} A_{S,\varepsilon}^2 \\ A_S^4 \end{array} \middle| \begin{pmatrix} 1 & 0 & 0 & 0 \\ 0 & 1 & 0 & 0 \\ 0 & 0 & -\mathrm{s}_\omega & \mathrm{c}_\omega \\ 0 & 0 & \mathrm{c}_\omega & -\mathrm{s}_\omega \end{pmatrix} \middle| \begin{array}{c} i_{in} \\ \mathrm{x} u_{in,\varepsilon} \\ -\mathrm{x} q_{in,\varepsilon} \\ v_{in} \end{array} \right\rangle =$$

$$= A_S^1 i_{in} + A_{S,\varepsilon}^3 u_{in,\varepsilon} - \mathrm{s}_\omega \left( A_{S,\varepsilon}^2 q_{in,\varepsilon} + A_S^4 v_{in} \right) - \mathrm{x}\mathrm{c}_\omega \left( A_S^4 q_{in,\varepsilon} + A_{S,\varepsilon}^2 v_{in} \right)$$

(C.8)

$$\omega = 0 \Rightarrow$$

$$\frac{I_S}{\eta_S T_{QW} I_{in}} = \frac{\langle \mathbf{A}_S | \mathbf{M}_{QW}(\mathrm{x}45°+\varepsilon,0) | \mathbf{I}_{in} \rangle}{T_{QW} I_{in}} = A_S^1 i_{in} + A_{S,\varepsilon}^3 u_{in,\varepsilon} - \mathrm{x}\left( A_S^4 q_{in,\varepsilon} + A_{S,\varepsilon}^2 v_{in} \right) =$$

$$= A_S^1 i_{in} - \left( \mathrm{s}_{2\varepsilon} A_S^2 - \mathrm{c}_{2\varepsilon} A_S^3 \right) \left( \mathrm{s}_{2\varepsilon} q_{in} - \mathrm{c}_{2\varepsilon} u_{in} \right) - \mathrm{x}\left[ A_S^4 \left( \mathrm{c}_{2\varepsilon} q_{in} + \mathrm{s}_{2\varepsilon} u_{in} \right) + \left( \mathrm{c}_{2\varepsilon} A_S^2 + \mathrm{s}_{2\varepsilon} A_S^3 \right) v_{in} \right]$$

(C.9)

**App. C.4    Calibration with a circular polariser (CP)**
From Eq. (C.2) for a circular polariser composed of a linear polariser and a λ/4-plate with
retardation error ω as in Eq. (C.7) we get the general calibration signals Eq. (C.10) with
analyser vectors <**A**| from App. D and input Stokes vectors |$\mathbf{I}_{in}$> from App. E. Note that z =
±1 discerns between a right and left circular polariser, and x = ±1 between the ±45°
orientations of the whole circular polariser. With an ideal linear polariser this quite complex
equation reduces to Eq. (C.11), with an ideal QWP without retardation error to Eq. (C.12), and
to Eq. (C.13) with both constraints, i.e. for an ideal circular polariser. Since only the terms
with an x in Eqs. (C.11) to (C.13) are compensated by means of the Δ90-calibration, neither
of the two constraints alone is sufficient to reduce the uncertainty.

$$\frac{I_S}{\eta_S T_{CP} I_{in}} = \frac{\langle \mathbf{A}_S | \mathbf{M}_{CP}(z, x45° + \varepsilon) | \mathbf{I}_{in} \rangle}{T_{CP} I_{in}} =$$

$$= \frac{\langle \mathbf{A}_S \mathbf{R}(x45° + \varepsilon) | \mathbf{M}_{QW}(z45°, \omega) \mathbf{M}_P | \mathbf{R}(-x45° - \varepsilon) \mathbf{I}_{in} \rangle}{T_{QW} T_P I_{in}} =$$

$$= \left\langle \begin{matrix} A_S^1 \\ xA_{S,\varepsilon}^3 \\ -xA_{S,\varepsilon}^2 \\ A_S^4 \end{matrix} \left| \begin{pmatrix} 1 & 0 & 0 & 0 \\ 0 & -s_\omega & 0 & -zc_\omega \\ 0 & 0 & 1 & 0 \\ 0 & zc_\omega & 0 & -s_\omega \end{pmatrix} \begin{pmatrix} 1 & D_P & 0 & 0 \\ D_P & 1 & 0 & 0 \\ 0 & 0 & Z_P c_P & Z_P s_P \\ 0 & 0 & -Z_P s_P & Z_P c_P \end{pmatrix} \right| \begin{matrix} i_{in} \\ xu_{in,\varepsilon} \\ -xq_{in,\varepsilon} \\ v_{in} \end{matrix} \right\rangle =$$  (C.10)

$$= \left\{ \begin{matrix} \left(A_S^1 + A_S^4 zc_\omega D_P\right) i_{in} + \left(A_{S,\varepsilon}^2 Z_P c_P - zA_{S,\varepsilon}^3 c_\omega Z_P s_P\right) q_{in,\varepsilon} - A_{S,\varepsilon}^3 s_\omega u_{in,\varepsilon} - A_S^4 s_\omega Z_P c_P v_{in} + \\ +x \begin{bmatrix} -A_{S,\varepsilon}^3 s_\omega D_P i_{in} - A_S^4 s_\omega Z_P s_P q_{in,\varepsilon} + \\ +\left(A_S^1 D_P + A_S^4 zc_\omega\right) u_{in,\varepsilon} - \left(A_{S,\varepsilon}^2 s_P + A_{S,\varepsilon}^3 zc_\omega c_P\right) Z_P v_{in} \end{bmatrix} \end{matrix} \right\}$$

From Eq.(C.10) we get with different conditions:

$$D_P = 1, Z_P = 0 \Rightarrow$$

$$\frac{I_S}{\eta_S T_{CP} I_{in}} = \left(A_S^1 + zc_\omega A_S^4 - s_\omega xA_{S,\varepsilon}^3\right)\left(i_{in} + xu_{in,\varepsilon}\right)$$  (C.11)

$$\omega = 0 \Rightarrow$$

$$\frac{I_S}{\eta_S T_{CP} I_{in}} = \left\{ \begin{matrix} \left(A_S^1 + A_S^4 zD_P\right) i_{in} + \left(A_{S,\varepsilon}^2 Z_P c_P - zA_{S,\varepsilon}^3 Z_P s_P\right) q_{in,\varepsilon} \\ +x\left[\left(A_S^1 D_P + zA_S^4\right) u_{in,\varepsilon} - \left(A_{S,\varepsilon}^2 s_P + zA_{S,\varepsilon}^3 c_P\right) Z_P v_{in}\right] \end{matrix} \right\}$$  (C.12)

$$\omega = 0, D_P = 1, Z_P = 0 \Rightarrow$$

$$\frac{I_S}{\eta_S T_{CP} I_{in}} = \left(A_S^1 + zA_S^4\right)\left(i_{in} - xu_{in,\varepsilon}\right)$$  (C.13)

**App. D The analyser row vector <$A_S$|**
The general formulation for the Stokes vector of a standard lidar signal $\mathbf{I}_S$ at the detector in the
reflected channel, $\mathbf{I}_R$, and transmitted channel, $\mathbf{I}_T$, is
$$\mathbf{I}_S = \eta_S \mathbf{M}_S \mathbf{R}_y \mathbf{M}_O(\gamma) \mathbf{F}(a) \mathbf{M}_E(\beta) \mathbf{I}_L$$  (D.1)
Only the first Stokes parameter is directly measured, and therefore we can reduce the
complexity of the full matrix equations to an inner product between the analyser row vector
<$\mathbf{A}_S$| and the input Stokes column vector $\mathbf{I}_{in}$ similar to Kaul et al. (1992); Volkov et al. (2015)
$$I_S = \langle \mathbf{A}_S | \mathbf{I}_{in} \rangle$$  (D.2)



In case of a calibration measurement, we place a calibrator with matrix $\mathbf{C}$ between the input
Stokes vector and the analyser vector
$I_S = \langle \mathbf{A}_S | \mathbf{C} | I_{in} \rangle$ (D.3)
As calibrators we use a mechanical rotator, a rotation of the plane of polarisation by means of
a $\lambda/2$ plate, a linear polariser, a $\lambda/4$ plate, and a circular polariser. We can place the calibrator
anywhere in the optical setup, with different results. In the following we develop the general
expressions of the analyser vector in App. D and of the input Stokes vector in App. E for the
different setups.
**App. D.1    $<A_S|$ with C before the polarising beam-splitter**



$I_S = \eta_S \mathbf{M}_S \mathbf{R}_y \mathbf{C} \mathbf{M}_O \mathbf{F} I_E \Rightarrow \mathbf{A}_S = \mathbf{M}_S \mathbf{R}_y$ (D.4)
The analyser part consists of a polarising beam-splitter $\mathbf{M}_S$ and an optional 90° rotation of the
detector setup $\mathbf{R}_y$ (see Eq.( 47))

$$
\frac{\langle \mathbf{A}_S |}{T_S} = \frac{\langle \mathbf{M}_S \mathbf{R}_y |}{T_S} =
$$

$$
= \left\langle \begin{pmatrix} 1 & D_S & 0 & 0 \\ D_S & 1 & 0 & 0 \\ 0 & 0 & Z_S c_S & Z_S s_S \\ 0 & 0 & -Z_S s_S & Z_S c_S \end{pmatrix} \begin{pmatrix} 1 & 0 & 0 & 0 \\ 0 & y & 0 & 0 \\ 0 & 0 & y & 0 \\ 0 & 0 & 0 & 1 \end{pmatrix} \right| = \left\langle \begin{pmatrix} 1 & yD_S & 0 & 0 \\ D_S & y & 0 & 0 \\ 0 & 0 & yZ_S c_S & Z_S s_S \\ 0 & 0 & -yZ_S s_S & Z_S c_S \end{pmatrix} \right| = \left\langle \begin{pmatrix} 1 \\ yD_S \\ 0 \\ 0 \end{pmatrix} \right|
$$

(D.5)
**App. D.2    $<A_S|$ with C before the receiving optics**



$I_S = \eta_S \mathbf{M}_S \mathbf{R}_y \mathbf{M}_O \mathbf{C} \mathbf{F} I_E \Rightarrow \mathbf{A}_S = \mathbf{M}_S \mathbf{R}_y \mathbf{M}_O$ (D.6)
Using Eq. D.5 we get





$$\frac{\langle \mathbf{A}_S(\mathrm{y},\gamma)|}{T_O T_S} = \frac{\langle \mathbf{M}_S \mathbf{R}_\mathrm{y} | \mathbf{M}_O(\gamma)}{T_O T_S} =$$

$$= \left\langle \begin{array}{c} 1 \\ \mathrm{y}D_S \\ 0 \\ 0 \end{array} \right| \left( \begin{array}{cccc} 1 & \mathrm{c}_{2\gamma}D_O & \mathrm{s}_{2\gamma}D_O & 0 \\ \mathrm{c}_{2\gamma}D_O & 1-\mathrm{s}_{2\gamma}^2 W_O & \mathrm{s}_{2\gamma}\mathrm{c}_{2\gamma}W_O & -\mathrm{s}_{2\gamma}Z_O\mathrm{s}_O \\ \mathrm{s}_{2\gamma}D_O & \mathrm{s}_{2\gamma}\mathrm{c}_{2\gamma}W_O & 1-\mathrm{c}_{2\gamma}^2 W_O & \mathrm{c}_{2\gamma}Z_O\mathrm{s}_O \\ 0 & \mathrm{s}_{2\gamma}Z_O\mathrm{s}_O & -\mathrm{c}_{2\gamma}Z_O\mathrm{s}_O & Z_O\mathrm{c}_O \end{array} \right) = \left\langle \begin{array}{c} 1+\mathrm{yc}_{2\gamma}D_O D_S \\ \mathrm{c}_{2\gamma}D_O + \mathrm{y}D_S\left(1-\mathrm{s}_{2\gamma}^2 W_O\right) \\ \mathrm{s}_{2\gamma}\left(D_O + \mathrm{yc}_{2\gamma}D_S W_O\right) \\ -\mathrm{ys}_{2\gamma}D_S Z_O\mathrm{s}_O \end{array} \right| \qquad \text{(D.7)}$$

Simplifications: A rotation $\gamma$ of a retarding diattenuator $\mathbf{M}_O$ between the calibrator and the
polarising beam-splitter $\mathbf{M}_S$ complicates the equations considerably. In case $\mathbf{M}_O$ is not rotated
($\gamma = 0$), the matrices $\mathbf{M}_S$, the optional 90° rotation $\mathbf{R}_\mathrm{y}$, and $\mathbf{M}_O$ and can be combined to a new
polarising beam-splitter module $\mathbf{M}_{SyO}$ according to S.10.10, and all equations developed for
the Sect. 7.1 case can be applied in Sect. 7.2. For $\gamma = \mathbf{0°}$ Eq. (D.7) becomes

$$\gamma = 0° \Rightarrow$$
$$\langle \mathbf{A}_S|(\mathrm{y},0°) = \langle \mathbf{M}_S \mathbf{R}_\mathrm{y} | \mathbf{M}_O(0°) = T_O T_S \langle 1 + \mathrm{y}D_S D_O \quad D_O + \mathrm{y}D_S \quad 0 \quad 0| =$$
$$= \langle \mathbf{M}_{SyO}(0°)| = T_{SyO} \langle 1 \quad D_{SyO} \quad 0 \quad 0| \qquad \text{(D.8)}$$

with $T_{SyO} = T_O T_S\left(1 + \mathrm{y}D_S D_O\right)$ and $D_{SyO} = \dfrac{D_O + \mathrm{y}D_S}{1 + \mathrm{y}D_S D_O}$        (D.9)
With a cleaned analyser we get from Eq. (D.9)

$$D_R = -1, D_T = +1 \Rightarrow$$
$$D_{SyO} = \mathrm{y}D_S, \quad D_{RyO} = -\mathrm{y}, \quad D_{TyO} = +\mathrm{y}$$
$$T_{RyO} = T_O T_R\left(1-\mathrm{y}D_O\right), \quad T_{TyO} = T_O T_T\left(1+\mathrm{y}D_O\right) \qquad \text{(D.10)}$$

and explicitly with Eqs. (S.10.10.11) and (S.10.10.14)

$$D_R = -1, D_T = +1, \mathrm{y} = +1 \Rightarrow$$
$$T_{R+O} = T_O T_R\left(1-D_O\right) = 0.5 T_R^s k_1 T_O^s, \quad D_{R+O} = -1$$
$$T_{T+O} = T_O T_T\left(1+D_O\right) = 0.5 T_T^p k_1 T_O^p, \quad D_{T+O} = +1$$
$$D_R = -1, D_T = +1, \mathrm{y} = -1 \Rightarrow$$
$$T_{R-O} = T_O T_R\left(1+D_O\right) = 0.5 T_R^s k_1 T_O^p, \quad D_{R-O} = +1$$
$$T_{T-O} = T_O T_T\left(1-D_O\right) = 0.5 T_T^p k_1 T_O^s, \quad D_{T-O} = -1 \qquad \text{(D.11)}$$

See also S.10.10 and S.6 .
Only few special cases with rotated $\mathbf{M}_O$ ($\gamma \neq \mathbf{0}$) (see Eq. (S.5.1.4)) are discussed additionally.


**App. D.3     <$A_S$| with C behind the emitter optics**

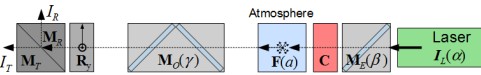

$I_S = \eta_S \mathbf{M}_S \mathbf{R}_y \mathbf{M}_O \mathbf{FC} I_E \Rightarrow \mathbf{A}_S = \mathbf{M}_S \mathbf{R}_y \mathbf{M}_O \mathbf{FC}$  and  $I_{in} = I_E$          (D.12)
The additional effect of the atmospheric depolarisation, $\mathbf{F}(a)$, on the analyser Eq. (D.7) is

$$\frac{\langle \mathbf{A}_S |}{T_O T_S F_{11}} = \frac{\langle \mathbf{M}_S \mathbf{R}_y \mathbf{M}_O (\gamma) | \mathbf{F}(a)}{T_O T_S F_{11}} =$$

$$= \left\langle \begin{matrix} 1 + yc_{2\gamma} D_S D_O \\ c_{2\gamma} D_O + y D_S (1 - s_{2\gamma}^2 W_O) \\ s_{2\gamma} (D_O + yc_{2\gamma} D_S W_O) \\ -y s_{2\gamma} D_S Z_O s_O \end{matrix} \right| \begin{pmatrix} 1 & 0 & 0 & 0 \\ 0 & a & 0 & 0 \\ 0 & 0 & -a & 0 \\ 0 & 0 & 0 & 1-2a \end{pmatrix} = \left\langle \begin{matrix} 1 + yc_{2\gamma} D_S D_O \\ a[c_{2\gamma} D_O + y D_S (1 - s_{2\gamma}^2 W_O)] \\ -a s_{2\gamma} (D_O + yc_{2\gamma} D_S W_O) \\ -(1-2a) y s_{2\gamma} D_S Z_O s_O \end{matrix} \right|$$          (D.13)

Without receiver optics rotation $\mathbf{M}_O$ ($\gamma = 0°$) we get with Eq. (D.8) ff.
$\langle \mathbf{A}_S | = \langle \mathbf{M}_{SyO} (0°) | \mathbf{F}(a) = T_{SyO} \langle 1 \quad a D_{SyO} \quad 0 \quad 0 |$          (D.14)
**App. E  The input Stokes vector $I_{in}$**
The formulation for the most general input Stokes vector $\boldsymbol{I}_{in}$ into the analyser part $\mathbf{A}_S$ is
$\boldsymbol{I}_{in} (\gamma, a, \beta) = \mathbf{M}_O (\gamma) \mathbf{F}(a) \mathbf{M}_E (\beta) \boldsymbol{I}_L$          (E.1)
and assuming a rotated, partly linear polarised laser with polarisation parameter $a_L$
$\boldsymbol{I}_{in} (\gamma, a, \beta, \alpha, a_L) = \mathbf{M}_O (\gamma) \mathbf{F}(a) \mathbf{M}_E (\beta) \boldsymbol{I}_L (\alpha, a_L)$          (E.2)
In the ideal case the laser has no depolarisation ($a_L = 1$) and is horizontal linearly polarised
(see Eq. (E.6)), and the optical elements are not rotated, which results in Eq.(E.3):

$$a_L = 1, i_L = q_L = 1, u_L = v_L = 0, \alpha = \beta = \gamma = 0 \Rightarrow$$
$$\boldsymbol{I}_{in} (0,0,0,0,1) = T_O F_{11} T_E I_L (1 + D_E) | 1 + a D_O \quad D_O + a \quad 0 \quad 0 \rangle$$          (E.3)

**App. E.1     Laser $I_L$**
We start with the Stokes vector for the laser beam with arbitrary state of polarisation and
additionally roated by angle $\alpha$ around the optical axis (see Eq. S.5.1.1)



$$\boldsymbol{I}_L(\alpha) = I_L \begin{vmatrix} i_{L,\alpha} \\ q_{L,\alpha} \\ u_{L,\alpha} \\ v_{L,\alpha} \end{vmatrix} = \begin{pmatrix} 1 & 0 & 0 & 0 \\ 0 & c_{2\alpha} & -s_{2\alpha} & 0 \\ 0 & s_{2\alpha} & c_{2\alpha} & 0 \\ 0 & 0 & 0 & 1 \end{pmatrix} I_L \begin{vmatrix} i_L \\ q_L \\ u_L \\ v_L \end{vmatrix} = I_L \begin{vmatrix} i_L \\ c_{2\alpha}q_L - s_{2\alpha}u_L \\ s_{2\alpha}q_L + c_{2\alpha}u_L \\ v_L \end{vmatrix}$$  (E.4)
The total, linear, and circular degree of polarisation (DOP, DLP, and DCP, respectively) don't
change with such a rotation.
• We get for a rotated, horizontal-linear polarised laser
$$\boldsymbol{I}_L(\alpha) = I_L \begin{vmatrix} 1 & c_{2\alpha} & s_{2\alpha} & 0 \end{vmatrix}$$  (E.5)
• for a horizontal-linear polarised laser
$$\boldsymbol{I}_L(0) = I_L \begin{vmatrix} 1 & 1 & 0 & 0 \end{vmatrix}$$  (E.6)
• and for a rotated, linearly polarised laser with polarisation parameter $a_L$ with $\delta_L = (1-a_L)/$
$(1+a_L)$
$$\boldsymbol{I}_L(\alpha, a_L) = \begin{pmatrix} 1 & 0 & 0 & 0 \\ 0 & c_{2\alpha} & -s_{2\alpha} & 0 \\ 0 & s_{2\alpha} & c_{2\alpha} & 0 \\ 0 & 0 & 0 & 1 \end{pmatrix} I_L \begin{vmatrix} 1 \\ a_L \\ 0 \\ 0 \end{vmatrix} = I_L \begin{vmatrix} 1 \\ c_{2\alpha}a_L \\ s_{2\alpha}a_L \\ 0 \end{vmatrix}$$  (E.7)
**App. E.2    $I_{in}$ with C behind the emitter optics**



$$\frac{\boldsymbol{I}_{in}(\beta, \alpha)}{T_E I_L} = \frac{\boldsymbol{I}_E(\beta, \alpha)}{T_E I_L} = \frac{\mathbf{M}_E(\beta) \boldsymbol{I}_L(\alpha)}{T_E I_L} =$$

$$= \begin{vmatrix} i_{in} & q_{in} & u_{in} & v_{in} \end{vmatrix} = \begin{vmatrix} i_E & q_E & u_E & v_E \end{vmatrix} = \frac{\mathbf{M}_E(\beta)}{T_E} \begin{vmatrix} i_L & q_L & u_L & v_L \end{vmatrix}$$  (E.8)

Eq. (E.8) with input $\boldsymbol{I}_L$ from a rotated, linearly polarised laser Eq. (E.4) and with rotated
emitter optics Eq. (S.10.4.1) results in Eq. (E.9).

$$
\frac{\boldsymbol{I}_{in}(\beta,\alpha)}{T_E I_L} = \frac{\boldsymbol{I}_E(\beta,\alpha)}{T_E I_L} = \frac{\mathbf{M}_E(\beta)\big|\boldsymbol{I}_L(\alpha)\big\rangle}{T_E I_L} = \big| i_E \quad q_E \quad u_E \quad v_E \big\rangle =
$$

$$
= \begin{pmatrix} 1 & c_{2\beta}D_E & s_{2\beta}D_E & 0 \\ c_{2\beta}D_E & 1-s_{2\beta}^2 W_E & s_{2\beta}c_{2\beta}W_E & -s_{2\beta}Z_E s_E \\ s_{2\beta}D_E & s_{2\beta}c_{2\beta}W_E & 1-c_{2\beta}^2 W_E & c_{2\beta}Z_E s_E \\ 0 & s_{2\beta}Z_E s_E & -c_{2\beta}Z_E s_E & Z_E c_E \end{pmatrix} \begin{vmatrix} i_L \\ c_{2\alpha}q_L - s_{2\alpha}u_L \\ s_{2\alpha}q_L + c_{2\alpha}u_L \\ v_L \end{vmatrix} =
$$

(E.9)

$$
= \begin{vmatrix} i_L + D_E\big(c_{2\alpha-2\beta}q_L - s_{2\alpha-2\beta}u_L\big) \\ c_{2\beta}D_E i_L + \big(c_{2\alpha}q_L - s_{2\alpha}u_L\big) + s_{2\beta}\Big[W_E\big(s_{2\alpha-2\beta}q_L + c_{2\alpha-2\beta}u_L\big) - Z_E s_E v_L\Big] \\ s_{2\beta}D_E i_L + \big(s_{2\alpha}q_L + c_{2\alpha}u_L\big) - c_{2\beta}\Big[W_E\big(s_{2\alpha-2\beta}q_L + c_{2\alpha-2\beta}u_L\big) - Z_E s_E v_L\Big] \\ -Z_E s_E\big(s_{2\alpha-2\beta}q_L + c_{2\alpha-2\beta}u_L\big) + Z_E c_E v_L \end{vmatrix}
$$

• Special cases: Eq. (E.9) without rotation of the emitter optics with respect to the plane of
polarisation of the laser

$$\alpha = \beta \Rightarrow$$

$$
\frac{\boldsymbol{I}_{in}(\alpha,\alpha)}{T_E I_L} = \frac{\boldsymbol{I}_E(\alpha,\alpha)}{T_E I_L} = \begin{vmatrix} i_E \\ q_E \\ u_E \\ v_E \end{vmatrix} = \begin{vmatrix} i_L + D_E q_L \\ c_{2\alpha}D_E i_L + \big(c_{2\alpha}q_L - s_{2\alpha}u_L\big) + s_{2\alpha}\big[W_E u_L - Z_E s_E v_L\big] \\ s_{2\alpha}D_E i_L + \big(s_{2\alpha}q_L + c_{2\alpha}u_L\big) - c_{2\alpha}\big[W_E u_L - Z_E s_E v_L\big] \\ -Z_E s_E u_L + Z_E c_E v_L \end{vmatrix} =
$$

(E.10)

$$
= \begin{vmatrix} i_L + D_E q_L \\ c_{2\alpha}\big(D_E i_L + q_L\big) - s_{2\alpha}Z_E\big(c_E u_L + s_E v_L\big) \\ s_{2\alpha}\big(D_E i_L + q_L\big) + c_{2\alpha}Z_E\big(c_E u_L + s_E v_L\big) \\ -Z_E\big(s_E u_L - c_E v_L\big) \end{vmatrix}
$$

• Eq. (E.9) without  laser and emitter optics rotation

$$\alpha = \beta = 0 \Rightarrow$$

$$
\frac{\boldsymbol{I}_{in}(0,0)}{T_E I_L} = \frac{\boldsymbol{I}_E(0,0)}{T_E I_L} = \frac{\mathbf{M}_E(0)\big|\boldsymbol{I}_L(0)\big\rangle}{T_E I_L} = \begin{vmatrix} i_E \\ q_E \\ u_E \\ v_E \end{vmatrix} = \begin{vmatrix} i_L + D_E q_L \\ D_E i_L + q_L \\ Z_E\big(c_E u_L + s_E v_L\big) \\ Z_E\big(-s_E u_L + c_E v_L\big) \end{vmatrix}
$$

(E.11)

• Eq. (E.9) with rotated, horizontal-linearly polarised laser with rotated emitter optics





$$\boldsymbol{I}_L = I_L \big| 1 \quad 1 \quad 0 \quad 0 \big\rangle \Rightarrow$$

$$\frac{\boldsymbol{I}_{in}(\beta,\alpha)}{T_E I_L} = \frac{\boldsymbol{I}_E(\beta,\alpha)}{T_E I_L} = \frac{\mathbf{M}_E(\beta)\big|\boldsymbol{I}_L(\alpha)\big\rangle}{T_E I_L} = \begin{vmatrix} i_E \\ q_E \\ u_E \\ v_E \end{vmatrix} = \begin{vmatrix} 1 + D_E c_{2\alpha-2\beta} \\ c_{2\alpha} + c_{2\beta}D_E + s_{2\beta}W_E s_{2\alpha-2\beta} \\ s_{2\alpha} + s_{2\beta}D_E - c_{2\beta}W_E s_{2\alpha-2\beta} \\ -Z_E s_E s_{2\alpha-2\beta} \end{vmatrix}$$
(E.12)

• Eq. (E.9) with rotated, linearly polarised laser without emitter optics rotation

$$\alpha = \beta \wedge \boldsymbol{I}_L = I_L \big| 1 \quad 1 \quad 0 \quad 0 \big\rangle \Rightarrow$$

$$\frac{\boldsymbol{I}_{in}(\alpha,\alpha)}{T_E I_L} = \big| i_E \quad q_E \quad u_E \quad v_E \big\rangle = (1 + D_E)\big| 1 \quad c_{2\alpha} \quad s_{2\alpha} \quad 0 \big\rangle$$
(E.13)

• Rotated, elliptically polarised light behind the emitter optics with.

$$\boldsymbol{I}_{in} = \boldsymbol{I}_E = T_E I_L \big| i_E \quad q_E \quad u_E \quad v_E \big\rangle = T_E I_L \big| 1 \quad bc_{2\alpha} \quad bs_{2\alpha} \quad v_E \big\rangle$$
(E.14)

with the degree of polarisation $DOP_E = 1$ and the degree of linear polarisation $DOLP_E = b$

$$DOP_E = \sqrt{q_E^2 + u_E^2 + v_E^2} = \sqrt{b^2 + v_E^2} = 1 \Rightarrow v_E = \sqrt{1 - b^2}$$
(E.15)

$$\boldsymbol{I}_{in} = \boldsymbol{I}_E = T_E I_L \big| i_E \quad q_E \quad u_E \quad v_E \big\rangle = T_E I_L \big| 1 \quad bc_{2\alpha} \quad bs_{2\alpha} \quad \sqrt{1-b^2} \big\rangle$$
(E.16)

• Rotated, linearly polarised laser with linear polarisation parameter $a_L$ with rotated emitter
optics: Laser Stokes vector Eq.(E.7) and rotated diattenuator Eq.(S.10.4.1)

$$\boldsymbol{I}_L = I_L \big| 1 \quad c_{2\alpha}a_L \quad s_{2\alpha}a_L \quad 0 \big\rangle \Rightarrow$$

$$\frac{\boldsymbol{I}_{in}}{T_E I_L} = \frac{\boldsymbol{I}_E}{T_E I_L} = \frac{\mathbf{M}_E(\beta)\big|\boldsymbol{I}_L(\alpha,a_L)\big\rangle}{T_E I_L} = \big| i_E \quad q_E \quad u_E \quad v_E \big\rangle =$$

$$= \begin{pmatrix} 1 & c_{2\beta}D_E & s_{2\beta}D_E & 0 \\ c_{2\beta}D_E & 1 - s_{2\beta}^2 W_E & s_{2\beta}c_{2\beta}W_E & -s_{2\beta}Z_E s_E \\ s_{2\beta}D_E & s_{2\beta}c_{2\beta}W_E & 1 - c_{2\beta}^2 W_E & c_{2\beta}Z_E s_E \\ 0 & s_{2\beta}Z_E s_E & -c_{2\beta}Z_E s_E & Z_E c_E \end{pmatrix} \begin{vmatrix} 1 \\ c_{2\alpha}a_L \\ s_{2\alpha}a_L \\ 0 \end{vmatrix} = \begin{vmatrix} 1 + a_L D_E c_{2\alpha-2\beta} \\ c_{2\beta}D_E + a_L\big(c_{2\alpha} + s_{2\beta}W_E s_{2\alpha-2\beta}\big) \\ s_{2\beta}D_E + a_L\big(s_{2\alpha} - c_{2\beta}W_E s_{2\alpha-2\beta}\big) \\ -a_L Z_E s_E s_{2\alpha-2\beta} \end{vmatrix}$$

(E.17)





**App. E.3**  $I_{in}$ **with C before the receiver optics**

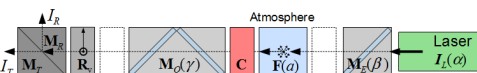

General input Stokes $I_{in}$ vector with atmospheric backscatter.
$$I_S = \eta_S \mathbf{M}_S \mathbf{R}_y \mathbf{M}_O \mathbf{C} \mathbf{F} I_E \Rightarrow I_{in} = \mathbf{F} I_E \qquad (E.18)$$
With atmospheric depolarisation from Eq. (S.3.1) and an emitter beam $I_E$ from App. E.2:
$$I_{in}(a) = |\mathbf{F}(a) I_E\rangle = F_{11} T_E I_L |i_E \quad aq_E \quad -au_E \quad (1-2a)v_E\rangle \qquad (E.19)$$
• Special cases: Eq. (E.19) becomes Eq. (E.20) with a rotated linearly polarised laser with
linear polarisation parameter $a_L$, with rotated emitter optics, and atmospheric backscatter, i.e.
Eq. (E.17). Note, that without laser depolarisation $a_L = 1$.

$$\frac{I_{in}(a,\beta,\alpha,a_L)}{F_{11}T_E I_L} = \frac{\mathbf{F}(a)|\mathbf{M}_E(\beta) I_L(\alpha,a_L)\rangle}{F_{11}T_E I_L} =$$
$$= \begin{vmatrix} i_{in} \\ q_{in} \\ u_{in} \\ v_{in} \end{vmatrix} = \begin{vmatrix} i_E \\ aq_E \\ -au_E \\ (1-2a)v_E \end{vmatrix} = \begin{vmatrix} 1 + a_L D_E c_{2\alpha-2\beta} \\ a\left[c_{2\beta}D_E + a_L\left(c_{2\alpha} + s_{2\beta}W_E s_{2\alpha-2\beta}\right)\right] \\ -a\left[s_{2\beta}D_E + a_L\left(s_{2\alpha} - c_{2\beta}W_E s_{2\alpha-2\beta}\right)\right] \\ -(1-2a)a_L Z_E s_E s_{2\alpha-2\beta} \end{vmatrix}$$
(E.20)

• Eq. (E.20) without rotation errors becomes Eq. (E.21), and additionally without laser
depolarisation, i.e. $a_L = 1$, Eq. (E.22).

$$\alpha = \beta = 0 \Rightarrow$$

$$\frac{I_{in}(a,0,0,a_L)}{F_{11}T_E I_L} = \frac{\mathbf{F}(a)|\mathbf{M}_E(0) I_L(0,a_L)\rangle}{F_{11}T_E I_L} = |1 + a_L D_E \quad aD_E + aa_L \quad 0 \quad 0\rangle \qquad (E.21)$$
$$I_{in}(a,0,0,0) = F_{11}T_E I_L (1+D_E)|1 \quad a \quad 0 \quad 0\rangle \qquad (E.22)$$
• Eq. (E.20) without emitter optics becomes Eq. (E.23).

$$[D_E = 0 \Rightarrow Z_E = 1, s_E = 0 \Rightarrow c_E = 1 \Rightarrow W_E = 0] \Rightarrow$$

$$\frac{I_{in}(a,,\alpha,a_L)}{F_{11} I_L} = \frac{\mathbf{F}(a)|I_L(\alpha,a_L)\rangle}{F_{11} I_L} = |1 \quad aa_L c_{2\alpha} \quad -aa_L s_{2\alpha} \quad 0\rangle \qquad (E.23)$$





Note it is impossible to combine $a' = aa_L$ if emitter optics $\mathbf{M}_E$ with diattenuation parameter $D_E$
$\neq 0$ or retardation (i.e. $Z_E \neq 0$ and $s_E \neq 0$) are between the laser and the atmosphere $\mathbf{F}$, even if
there are no angular misalignments $\alpha$ and $\beta$ in the emitter, which means that the atmospheric
depolarisation cannot be retrieved without detailed   knowledge of the emitter optics
parameters and alignment errors.
• Eq. (E.20) without emitter optics $\mathbf{M}_E$ and without laser depolarisation becomes Eq. (E.24).

$$a_L = 1, \left[ D_E = 0 \Rightarrow Z_E = 1, s_E = 0 \Rightarrow c_E = 1 \Rightarrow W_E = 0 \right] \Rightarrow$$

$$\frac{\boldsymbol{I}_{in}(a,,\alpha)}{I_{in}} = \frac{\mathbf{F}(a)\big|\boldsymbol{I}_L(\alpha)\big\rangle}{F_{11}I_L} = \big|1 \quad ac_{2\alpha} \quad -as_{2\alpha} \quad 0\big\rangle \tag{E.24}$$

• Eq. (E.19) with $\boldsymbol{I}_E$ from Eq. (E.14), i.e. with rotated, elliptically polarised light behind the
emitter optics

$$\frac{\boldsymbol{I}_{in}(a,b,\alpha)}{I_{in}} = \frac{\mathbf{F}(a)\boldsymbol{I}_E}{F_{11}T_E I_L} = \big|i_E \quad aq_E \quad -au_E \quad (1-2a)v_E\big\rangle =$$

$$= \big|1 \quad abc_{2\alpha} \quad -abs_{2\alpha} \quad (1-2a)\sqrt{1-b^2}\big\rangle \tag{E.25}$$

• Including the calibrator rotation R($\varepsilon$) in $I_{in}$ in Eq. (E.19) with Eq. (S.10.15.1) gives Eq.
(E.26), and with elliptically polarise laser of Eq. (E.16) we get Eq. (E.27), which results
without emitter optics and horizontal-linear polarised laser light ($b = 1$) in Eq. (E.28).

$$\frac{\boldsymbol{I}_{in,\varepsilon}(\varepsilon,\mathrm{h},a)}{I_{in}} = \frac{\big|\mathbf{R}(\varepsilon)\mathbf{M}_\mathrm{h}\mathbf{F}(a)\boldsymbol{I}_E\big\rangle}{T_{rot}F_{11}T_E I_L} =$$

$$= \begin{vmatrix} i_{in,\varepsilon} \\ q_{in,\varepsilon} \\ u_{in,\varepsilon} \\ v_{in,\varepsilon} \end{vmatrix} = \begin{pmatrix} 1 & 0 & 0 & 0 \\ 0 & c_{2\varepsilon} & -hs_{2\varepsilon} & 0 \\ 0 & s_{2\varepsilon} & hc_{2\varepsilon} & 0 \\ 0 & 0 & 0 & h \end{pmatrix} \begin{vmatrix} i_E \\ aq_E \\ -au_E \\ (1-2a)v_E \end{vmatrix} = \begin{vmatrix} i_E \\ a(q_E c_{2\varepsilon} + hu_E s_{2\varepsilon}) \\ a(q_E s_{2\varepsilon} - hu_E c_{2\varepsilon}) \\ (1-2a)hv_E \end{vmatrix} \tag{E.26}$$

$$\boldsymbol{I}_E = T_E I_L \big|i_E \quad q_E \quad u_E \quad v_E\big\rangle = T_E I_L \big|1 \quad bc_{2\alpha} \quad bs_{2\alpha} \quad \sqrt{1-b^2}\big\rangle \Rightarrow$$

$$\frac{\boldsymbol{I}_{in,\varepsilon}(\varepsilon,\mathrm{h},a,\alpha,b)}{I_{in}} = \frac{\big|\mathbf{R}(\varepsilon)\mathbf{M}_\mathrm{h}\mathbf{F}(a)\boldsymbol{I}_E(\alpha,b)\big\rangle}{T_{rot}F_{11}T_E I_L} =$$

$$= \big|i_{in,\varepsilon} \quad q_{in,\varepsilon} \quad u_{in,\varepsilon} \quad v_{in,\varepsilon}\big\rangle = \big|1 \quad abc_{2\varepsilon-\mathrm{h}2\alpha} \quad abs_{2\varepsilon+\mathrm{h}2\alpha} \quad (1-2a)\mathrm{h}\sqrt{1-b^2}\big\rangle \tag{E.27}$$



$$\mathbf{M}_E = idendity, b = 1 \Rightarrow$$

$$\frac{\mathbf{I}_{in,\varepsilon}(\varepsilon, \mathrm{h}, a, \alpha, b)}{I_{in}} = \frac{\left| \mathbf{R}(\varepsilon) \mathbf{M}_\mathrm{h} \mathbf{F}(a) \mathbf{I}_L(\alpha, b) \right\rangle}{T_{rot} F_{11} I_L} = \tag{E.28}$$

$$= \left| i_{in,\varepsilon} \quad q_{in,\varepsilon} \quad u_{in,\varepsilon} \quad v_{in,\varepsilon} \right\rangle = \left| 1 \quad a\mathrm{c}_{2\varepsilon-\mathrm{h}2\alpha} \quad a\mathrm{s}_{2\varepsilon+\mathrm{h}2\alpha} \quad 0 \right\rangle$$

## App. E.4  $I_{in}$ with C before the polarising beam-splitter

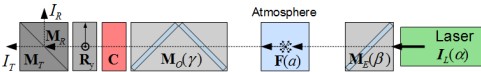

General input vector $I_{in}$ with atmospheric backscatter and emitter and receiver optics.

$$\mathbf{I}_S = \eta_S \mathbf{M}_S \mathbf{R}_\mathrm{y} \mathbf{C} \mathbf{M}_O \mathbf{F} \mathbf{M}_E \mathbf{I}_L \Rightarrow \mathbf{I}_{in} = \mathbf{M}_O \mathbf{F} \mathbf{I}_E \tag{E.29}$$

The most complex case for the input Stokes vector $I_{in}$ is, if the calibrator is placed before the polarising beam-splitter, because here we have to multiply several matrices. All other cases can be derived from this case by neglecting the appropriate parameters (see App. D). The emitted beam Stokes vector $I_E$ from App. E.2 has to be multiplied with the atmospheric backscatter matrix $\mathbf{F}$ (Eq. (S.3.1)) and the receiver optics matrix $\mathbf{M}_O$, the latter expressed as a rotated diattenuator (see Eq. (E.32)). In general the emitter optics and the laser polarisation $I_L$ are rotated as in Eq. (E.30), which is not mentioned explicitly when needless.

$$\mathbf{I}_E(\beta, \alpha) = \mathbf{M}_E(\beta) \left| \mathbf{I}_L(\alpha) \right\rangle = T_E I_L \left| i_E(\beta, \alpha) \quad q_E(\beta, \alpha) \quad u_E(\beta, \alpha) \quad v_E(\beta, \alpha) \right\rangle \tag{E.30}$$

$$\frac{\mathbf{I}_{in}(\gamma, a,,)}{T_{in} I_L} = \frac{\mathbf{M}_O(\gamma) \left| \mathbf{F}(a) \mathbf{M}_E \mathbf{I}_L \right\rangle}{T_O F_{11} T_E I_L} = \frac{\mathbf{M}_O(\gamma) \left| \mathbf{F}(a) \mathbf{I}_E \right\rangle}{T_O F_{11} T_E I_L} = \left| i_{in} \quad q_{in} \quad u_{in} \quad v_{in} \right\rangle =$$

$$= \begin{pmatrix} 1 & \mathrm{c}_{2\gamma} D_O & \mathrm{s}_{2\gamma} D_O & 0 \\ \mathrm{c}_{2\gamma} D_O & 1 - \mathrm{s}_{2\gamma}^2 W_O & \mathrm{s}_{2\gamma} \mathrm{c}_{2\gamma} W_O & -\mathrm{s}_{2\gamma} Z_O \mathrm{s}_O \\ \mathrm{s}_{2\gamma} D_O & \mathrm{s}_{2\gamma} \mathrm{c}_{2\gamma} W_O & 1 - \mathrm{c}_{2\gamma}^2 W_O & \mathrm{c}_{2\gamma} Z_O \mathrm{s}_O \\ 0 & \mathrm{s}_{2\gamma} Z_O \mathrm{s}_O & -\mathrm{c}_{2\gamma} Z_O \mathrm{s}_O & Z_O \mathrm{c}_O \end{pmatrix} \left| \begin{matrix} i_E \\ a q_E \\ -a u_E \\ (1 - 2a) v_E \end{matrix} \right\rangle = \tag{E.31}$$

$$= \left| \begin{matrix} i_E + D_O a \left( \mathrm{c}_{2\gamma} q_E - \mathrm{s}_{2\gamma} u_E \right) \\ \mathrm{c}_{2\gamma} D_O i_E + a q_E - \mathrm{s}_{2\gamma} \left[ W_O a \left( \mathrm{s}_{2\gamma} q_E + \mathrm{c}_{2\gamma} u_E \right) + Z_O \mathrm{s}_O (1 - 2a) v_E \right] \\ \mathrm{s}_{2\gamma} D_O i_E - a u_E + \mathrm{c}_{2\gamma} \left[ W_O a \left( \mathrm{s}_{2\gamma} q_E + \mathrm{c}_{2\gamma} u_E \right) + Z_O \mathrm{s}_O (1 - 2a) v_E \right] \\ Z_O \mathrm{s}_O a \left( \mathrm{s}_{2\gamma} q_E + \mathrm{c}_{2\gamma} u_E \right) + Z_O \mathrm{c}_O (1 - 2a) v_E \end{matrix} \right\rangle$$

• Special cases: From Eq. (E.31) without receiver optics rotation $\gamma$ we get Eq. (E.32).



$\gamma = 0 \Rightarrow$

$\dfrac{\boldsymbol{I}_{in}(0,a,,)}{T_{in}I_L} = \dfrac{\mathbf{M}_O(0)\big|\mathbf{F}(a)\mathbf{M}_E\boldsymbol{I}_L\big\rangle}{T_O F_{11} T_E I_L} = \big| i_{in} \quad q_{in} \quad u_{in} \quad v_{in} \big\rangle = \left|\begin{array}{c} i_E + aD_O q_E \\ D_O i_E + aq_E \\ Z_O\big[-\text{c}_O a u_E + \text{s}_O(1-2a)v_E\big] \\ Z_O\big[\text{s}_O a u_E + \text{c}_O(1-2a)v_E\big] \end{array}\right\rangle$   (E.32)
• With linearly polarised laser $\boldsymbol{I}_L$ with polarisation parameter $a_L$, with emitter optics $\mathbf{M}_E$,
atmosphere $\mathbf{F}$, and receiver optics $\mathbf{M}_O$, and with Eqs. (E.32) and (E.20) we get Eq. (E.33).

$i_L = q_L = 1, u_L = v_L = 0 \Rightarrow$

$\dfrac{\boldsymbol{I}_{in}(\gamma,a,\beta,\alpha,a_L)}{T_{in}I_L} = \dfrac{\mathbf{M}_O(\gamma)\big|\mathbf{F}(a)\mathbf{M}_E(\beta)\boldsymbol{I}_L(\alpha,a_L)\big\rangle}{T_O F_{11} T_E I_L} =$

$= \begin{pmatrix} 1 & \text{c}_{2\gamma}D_O & \text{s}_{2\gamma}D_O & 0 \\ \text{c}_{2\gamma}D_O & 1-\text{s}_{2\gamma}^2 W_O & \text{s}_{2\gamma}\text{c}_{2\gamma}W_O & -\text{s}_{2\gamma}Z_O\text{s}_O \\ \text{s}_{2\gamma}D_O & \text{s}_{2\gamma}\text{c}_{2\gamma}W_O & 1-\text{c}_{2\gamma}^2 W_O & \text{c}_{2\gamma}Z_O\text{s}_O \\ 0 & \text{s}_{2\gamma}Z_O\text{s}_O & -\text{c}_{2\gamma}Z_O\text{s}_O & Z_O\text{c}_O \end{pmatrix} \left|\begin{array}{c} 1 + a_L D_E \text{c}_{2\alpha-2\beta} \\ a\big[\text{c}_{2\beta}D_E + a_L(\text{c}_{2\alpha}+\text{s}_{2\beta}W_E\text{s}_{2\alpha-2\beta})\big] \\ -a\big[\text{s}_{2\beta}D_E + a_L(\text{s}_{2\alpha}-\text{c}_{2\beta}W_E\text{s}_{2\alpha-2\beta})\big] \\ -(1-2a)a_L Z_E \text{s}_E \text{s}_{2\alpha-2\beta} \end{array}\right\rangle$   (E.33)
• Eq. (E.33) with rotated, linearly polarised laser without laser depolarisation ($a_L = 1$) and
rotated emitter optics (Eq. (E.20)) the input Stokes vector becomes explicitly

$a_L = 1, i_L = q_L = 1, u_L = v_L = 0, \gamma = 0 \Rightarrow$

$\dfrac{\boldsymbol{I}_{in}}{T_{in}I_L} = \dfrac{\mathbf{M}_O(0)\big|\mathbf{F}(a)\mathbf{M}_E(\beta)\boldsymbol{I}_L(\alpha)\big\rangle}{T_O F_{11} T_E I_L} =$

$= \left|\begin{array}{c} (1+D_E\text{c}_{2\alpha-2\beta}) + aD_O(\text{c}_{2\alpha}+\text{c}_{2\beta}D_E+\text{s}_{2\beta}W_E\text{s}_{2\alpha-2\beta}) \\ D_O(1+D_E\text{c}_{2\alpha-2\beta}) + a(\text{c}_{2\alpha}+\text{c}_{2\beta}D_E+\text{s}_{2\beta}W_E\text{s}_{2\alpha-2\beta}) \\ -Z_O\big\{\text{s}_O Z_E\text{s}_E\text{s}_{2\alpha-2\beta} + a\big[\text{c}_O(\text{s}_{2\alpha}+\text{s}_{2\beta}D_E-\text{c}_{2\beta}W_E\text{s}_{2\alpha-2\beta}) - 2\text{s}_O Z_E\text{s}_E\text{s}_{2\alpha-2\beta}\big]\big\} \\ -Z_O\big\{\text{c}_O Z_E\text{s}_E\text{s}_{2\alpha-2\beta} - a\big[\text{s}_O(\text{s}_{2\alpha}+\text{s}_{2\beta}D_E-\text{c}_{2\beta}W_E\text{s}_{2\alpha-2\beta}) + 2\text{c}_O Z_E\text{s}_E\text{s}_{2\alpha-2\beta}\big]\big\} \end{array}\right\rangle$   (E.34)
• Eq. (E.34) with laser polarisation and emitter optics aligned

$a_L = 1, i_L = q_L = 1, u_L = v_L = 0, \gamma = 0, \beta = \alpha \Rightarrow$

$\dfrac{\boldsymbol{I}_{in}}{T_{in}I_L} = \dfrac{\mathbf{M}_O(0)\big|\mathbf{F}(a)\mathbf{M}_E(\alpha)\boldsymbol{I}_L(\alpha)\big\rangle}{T_O F_{11} T_E I_L} = (1+D_E)\left|\begin{array}{c} 1 + aD_O\text{c}_{2\alpha} \\ D_O + a\text{c}_{2\alpha} \\ -Z_O a\text{c}_O\text{s}_{2\alpha} \\ +Z_O a\text{s}_O\text{s}_{2\alpha} \end{array}\right\rangle$   (E.35)
• and without any optics and laser rotation



$$a_L = 1, i_L = q_L = 1, u_L = v_L = 0, \alpha = \beta = \gamma = 0 \Rightarrow$$
$$\frac{\boldsymbol{I}_{in}(0,0,0,0,1)}{T_{in}I_L} = \frac{\mathbf{M}_O(0)\big|\mathbf{F}(a)\mathbf{M}_E(0)\boldsymbol{I}_L(0)\big\rangle}{T_O F_{11} T_E I_L} = (1+D_E)\big|1+aD_O \quad D_O+a \quad 0 \quad 0\big\rangle \tag{E.36}$$

• Eq. (E.33) without emitter optics $\mathbf{M}_E$
$$D_E = 0, \ s_E = 0, \ W_E = 0, \ a' = aa_L \Rightarrow$$
$$\frac{\boldsymbol{I}_{in}(\gamma,a,0,\alpha,a_L)}{T_{in}I_L} = \frac{\mathbf{M}_O(\gamma)\mathbf{F}(a)\boldsymbol{I}_L(\alpha,a_L)}{T_O F_{11} I_L} = \left|\begin{matrix} 1+c_{2\gamma+2\alpha}a'D_O \\ c_{2\gamma}D_O + a'\big[c_{2\alpha}-s_{2\gamma}s_{2\gamma+2\alpha}W_O\big] \\ s_{2\gamma}D_O - a'\big[s_{2\alpha}-c_{2\gamma}s_{2\gamma+2\alpha}W_O\big] \\ s_{2\gamma+2\alpha}a'Z_O s_O \end{matrix}\right\rangle \tag{E.37}$$

• No emitter optics $\mathbf{M}_E$ and no receiver optics rotation
with $\gamma = 0$, $T_E = 1$, $D_E = 0$, $s_E = 0$, $W_E = 0$, $a' = aa_L \Rightarrow$
$$\frac{\boldsymbol{I}_{in}(0,a,0,\alpha,a_L)}{T_{in}I_L} = \frac{\mathbf{M}_O(0)\mathbf{F}(a)\boldsymbol{I}_L(\alpha,a_L)}{T_O F_{11} I_L} =$$
$$= \big|1+c_{2\alpha}a'D_O \quad D_O+c_{2\alpha}a' \quad -s_{2\alpha}a'Z_O c_O \quad s_{2\alpha}a'Z_O s_O\big\rangle \tag{E.38}$$

• The latter and no laser rotation
with $\alpha = 0$, $\gamma = 0$, $T_E = 1$, $D_E = 0$, $s_E = 0$, $W_E = 0$, $a' = aa_L \Rightarrow$
$$\frac{\boldsymbol{I}_{in}(0,a,0,0,a_L)}{T_{in}I_L} = \frac{\mathbf{M}_O(0)\mathbf{F}(a)\boldsymbol{I}_L(0,a_L)}{T_O F_{11} I_L} = \big|1+a'D_O \quad D_O+a' \quad 0 \quad 0\big\rangle \tag{E.39}$$

## App. E.5  $\boldsymbol{I}_{in}$ with C amidst the receiving optics

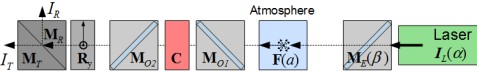

In case there is polarising or/and retarding optics before ($\mathbf{M}_{O1}$) and after ($\mathbf{M}_{O2}$) the calibrator
as in Eq. (E.40), the basic equations can be constructed by using the analyser matrix $\mathbf{A}_S$ from
App. D.2 and the input Stokes vectors $\boldsymbol{I}_{in}$ from App. E.4.
$$\boldsymbol{I}_S = \eta_S \mathbf{M}_S \mathbf{R}_y \mathbf{M}_{O2} \mathbf{C} \mathbf{M}_{O1} \mathbf{F} \boldsymbol{I}_E \Rightarrow \mathbf{A}_S = \mathbf{M}_S \mathbf{R}_y \mathbf{M}_{O2} \quad \text{and} \quad \boldsymbol{I}_{in} = \mathbf{M}_{O1} \mathbf{F} \boldsymbol{I}_E \tag{E.40}$$