# Peer review of "About the effects of polarising optics on lidar signals and the Δ90-calibration"

_Atmospheric Measurement Techniques, 2015_

## Referee Comment (RC1) · Anonymous Referee #2 · 10 Mar 2016

The work presented here evaluates a number of polarization measurement schemes for sensitivity to errors introduced by non-ideal optical elements.  This analysis includes consideration of different polarization calibration schemes that, as far as I can tell, the author has previously published.

The work presented here is thorough and detailed and I saw no obvious errors.  That said, I have trouble finding any novel aspects to the work.  It seems like this would be better suited as a white paper.

I have two major and related criticisms that I think significantly weaken this work and make it very difficult to review this work for any novelty.

***It belabors a number of topics that have been thoroughly addressed in other published works while completely ignoring other aspects, creating the false impression of a comprehensive analysis***
The following are a few examples.  This list is not comprehensive.

Section 2.1 Depolarization atmospheric aerosol
There have been a large number of works that cover the backscatter matrix of randomly oriented particles, many of which are already referenced in this section.  There are 6 equations that could be summarized in 2.
Eq. (9) and (10) are completely obvious from Eq. (8).
Eq. (11) is belaboring an obvious and trivial point that can be made in one sentence.
Eq. (12) might be worth keeping, but it does not need every step of algebra.  Just include the result.  It is obvious how you get this result.
Eq. (13) is covering the obvious.  Hopefully the reader already knows how to do matrix multiplication.

For all the maticulous steps in this section, there is absolutely no mention of oriented particles.  In my view, the reader may be given the false impression that this scattering matrix is somehow comprehensive, which it clearly is not, since it only covers single scattering by randomly oriented particles.

Section 2.2 Optical Parts:
In Eq. (14) (and others) pick a matrix form and stick to it.  You don't need multiple forms of a diattenuating retarder, the terms of which, you subsequently define in the very next equations.

Eq. (45) conveys nothing to the reader.  The Mueller matrix and Stokes vector are completely arbitrary.  Again, hopefully the reader already knows how to pull a common term out of a matrix.

***It spends too much space evaluating matrix equations to produce lengthy scalar equations with too many variables.***

The presentation in this work uses far more equations than necessary. Frankly, most of the results and derivations are actually just evaluations of matrix equations into a scalar form. I don't really see the value in that. Matrix equations are a nice concise way to present Mueller calculus. They are designed for the purpose of concisely and elegantly handling systems of equations. Computers with all the most popular numerical analysis software handle matrix operations without any difficulty. There really isn't a reason to evaluate to a scalar equation unless it produces a result that is simple, concise and reveals some previously not obvious fact. That is, evaluating the matrices should produce a reduction in complexity, not the other way around.

A better approach is to leave most of the equations in their matrix form. There are not a lot of different matrix forms in use, so provide those definitions once and allow the reader to substitute them. My guess is that a few matrix equations and a few plots would go a long way in reducing the size of this work.

This is a pretty consistent criticism of nearly every section. One example of this is the beginning of section 4, where you could easily take a reductionist approach. Present the overall Mueller equation describing $I_s$ (Eq. (64)?), then give the evaluation in Eq. (68), but get rid of all those coefficients in front, which can just as easily fold into G and H and aren't important since absolute intensity measurements are almost never used in atmospheric lidar. One can obtain this result without ever assuming a cascade of particular polarization element matrices. All the equations between (64) and (68) look like noise to me. I can't keep track of all those variables, and frankly, they probably are not representative of my lidar. I'd rather do the Mueller calculus myself, and I can't follow your inbetween steps anyway.

The explicit definitions of G and H seem unnecessary. Again, can I really expect this to be representative of my lidar? And again, I can't keep track of what all the variables mean. Do these equations ever get used again in the paper? If not, all the more reason to get rid of them.

**Final comments to improve readability:**

My view is that this work is most likely to be used by those unfamiliar with polarization theory and Mueller calculus. Assuming that is the audience, I have the following suggestions:

Keep in mind that it is extremely unlikely that a reader will be reading this work the entire way through, so try to make the it easy to skip through. I know this is a vague comment, but keeping this fact in your mind will make the paper readable. It is good there is a table of variable definitions, but maybe it would be better to break them up a bit based on where they are used in the paper. Also, make a table of assumptions used in these derivations or put those assumptions at the beginning of the relevant section.

Drop the  notation. That is going to confuse readers more than anything. All the operations you are trying to convey already exist in standard matrix notations. Stick to dot products or transpose, which can be used equally well to express the same operations. The  notation will more than likely just cause the reader to "check-out" when reading.

Consider presenting the result (what the user will actually use in calibration) before any of the derivations. This makes the outcome more accessible.

Be very clear about the assumptions applied in this work (preferably itemize them or keep them in a table). Users unfamiliar with polarization are likely to take your work and run with it, without ever double checking the assumptions you give. Take measures to make sure your work isn't misused. For example, these corrections only apply to scattering matrices in the form presented by Gimmestad AO 2008. So oriented ice crystals, and of greater concern rain (see Hayman Opt. Express 2014), are not likely to have accurate polarization corrections. Also, you assume optical elements are some combination of retarder and diattenuator. That is not always true, mostly depending on what level of accuracy you are hoping to obtain. The point is, your assumptions seem reasonable to me, but they may not be a reasonable for *all possible* cases, so make sure you are clear about them and make sure they are easily accessible.

**A Personal Opinion from the Reviewer:**
I don't really understand all this interest in obtaining "correction equations" for scalar polarization variables. The approach used in Kaul Appl. Opt. 2004, Hayman, Opt. Express 2012, Volkov Appl. Opt. 2015 and countless other polarimetry papers avoids any need to belabor "corrections" and just retrieves the relevant scattering matrix terms based on the lidar's operational parameters--"errors" or whatever you want to call them. It is trivial to adjust the approach presented there for a randomly oriented matrix and a stationary polarization (which I use as standard practice in my own analysis). Framing the problem like this makes one realize there are a *maximum* of 9 parameters that describe the standard stationary depolarization lidar. Obtaining the those parameters by some means is necessary, so the calibration techniques outlined here are important, but obtaining an accurate depolarization measurement is needlessly complex when it is presented as a scalar correction formula to a depolarization ratio measurement. Let the computer do the hard part.

---

## Referee Comment (RC2) · Anonymous Referee #1 · 11 Apr 2016

Date: April. 11, 2016 From: "Sergei Nikolaevich Volkov" srgy_volkov@yahoo.com To: "AMT" editorial@copernicus.org Subject: Expert review

Title: About the effects of polarising optics on lidar signals and the Δ90-calibration Author(s): V. Freudenthaler MS No.: amt-2015-338 MS Type: Research article Iteration: Manuscript under review for journal Atmos. Meas. Tech. Special Issue: EARLINET, the European Aerosol Research Lidar Network

Summary of Recommendations

My recommendation for this paper in its existing form is:

The manuscript is suitable for publication.

Detailed Comments:

The author of this paper consider a general model of the polarisation sensitivity of typical lidar systems and propose to use a $\Delta$90-calibration for decrease the error of a single $\pm 45°$ calibration into insignificance.

Pro: The paper is certainly of interest. The material presented in the paper is original. The references used in the paper are well balanced with the subject discussed. The formulae presented in the paper are quite correct.

Contra: I had to make considerable efforts in order to concentrate on reading this article. You might think that expressing thoughts in simple language becomes an art. In General this is what is needed as the basis for writing this article. The last effort of will. As a result, 95 % of it is just good material should go in the trash, and the remaining 5% should be pleasing to the reader.

With best Regards, Good luck!

Please also note the supplement to this comment:
http://www.atmos-meas-tech-discuss.net/amt-2015-338/amt-2015-338-RC2-supplement.pdf
* * *

---

## Referee Comment (RC3) · Anonymous Referee #3 · 28 Apr 2016

This manuscript represents a very large and comprehensive body of work. Yes, it may be pedantic, but it is very well organized and very thorough. Best of all, the extensive mathematical formalism is boiled down by the author to just a few very valuable conclusions, so the main points are never lost. The conclusions suggest "best practices" for building polarization-sensitive lidars, so they are valuable to the worldwide lidar community.

The paper is well suited for the special issue on EARLINET, for this reason: EARLINET is not a geographically-distributed set of identical lidars (as MPLNET is), it is a network of mostly home-grown lidars that are different in their design details. For that reason, one paper discussing polarization techniques for the whole network must necessarily cover a wide range of designs and techniques. In writing such a paper, the author has also benefitted a much wider community of lidar practitioners. This is a happy outcome.

[Figure]

For these reasons, I regard the paper as an excellent contribution that is well worth publishing.
* * *

---

## Author Comment (AC1) · 28 Jun 2016

Response to the reviews of manuscript amt-2015-338.

I am deeply grateful to the reviewers for reading this very long manuscript.

Its lengthiness and the explicit derivation of many scalar equations where matrix equations would be much more concise is the main objection of reviewers 1 and 2.

There are several reasons why I think this elaborateness is necessary.

Most of the lidar papers lack a thorough error analysis, especially regarding systematic errors. Why is that, and how can it be changed?

As can be seen from this manuscript, which deals only with polarisation related uncertainties, there are many different error sources, and it is laborious to consider them all together. I have not seen a complete error analysis of lidar measurements and their products published yet. If at all, uncertainties due to signal and background noise are processed, and in only few cases one or the other systematic uncertainty is included. The description how the error bars are achieved is often insufficient and important error sources are often neglected. But we urgently need verifiable error bars which reflect the true uncertainty without systematic biases, if we want our data to be assimilated in weather models and used in micro-physical retrieval algorithms, whose outcomes strongly depend on the uncertainty of the input data.

The persons which are able to prepare the error analysis of a lidar system are the ones which built it, because only these persons can be aware of all the error sources which have to be included. But these persons have enough troubles to deal with the instrumental issues and are often under pressure of time by the schedule of the project which funds the lidar or by the limited time span for their PhD study, which must present some new, exiting atmospheric measurements at the end. Usually, there is no time left for a thorough error analysis, which would cost maybe another year, and for which there would be only little appreciation value.

On the other hand, many error sources could already be eliminated in the design of the lidar systems, but concise matrix formulas, as proposed by the reviewers, don't reveal the influence of the individual elements and of the parameters of the matrices. If we want to estimate the importance of a certain parameter either for system design or error estimation, or to compare different system or calibration setups, it is necessary to expand the concise matrix equations as it is done in this manuscript, hopefully saving many colleagues a lot of time and mistakes.

There are several papers dealing with one or the other error source, and much could be learned from publications in the field of ellipsometry. But these publications are not only in another "language", but in many different "languages", with different assumptions, different definitions and conventions, different neglects also, which takes a lot of time to sort out what can be used. And then, finally, it takes again some time to write an error analysis code and to verify that it really works correctly.

To my opinion, concise matrix equations don't help those which actually have to run the lidar systems and do the error analysis.

I admit that such a lengthy manuscript as present here is a burden as well, but it tries to cover the different variants of the most common type of lidar systems with different calibration techniques, and it is also meant as a starting point for further developments in the direction of a complete error analysis and a related open source software. Therefore I want to present all definitions, derivations, and variants in one paper in such a way, that misunderstandings and mistakes are less probable. The manuscript is also meant as a reference for future papers, so that the derivations don't have to be repeated again and again with different definitions. The companion paper "*Assessment of lidar depolarization uncertainty by means of a polarimetric lidar simulator* " by J. A. Bravo-Aranda et al. (doi:10.5194/amt-2015-339 ) shows already an application.

Another critique is, that we should "let the computer do the hard part" and use software which can deal with matrix calculations. The original Monte-Carlo code in the above mentioned companion paper was written with such a software, and it took orders of magnitude longer than the scalar code we wrote for verification. And this is only the beginning, because we have to include more parameters for a complete error analysis and make it somehow operational. The error debugging of these two codes showed, that a detailed reference for the used equations is necessary.

Most of the above is already mentioned in the introduction of the manuscript.

**Detailed answers to Anonymous Referee #2:**

This analysis includes consideration of different polarization calibration schemes that, as far as I can tell, the author has previously published.

There is only one publication (Freudenthaler et al., 2009), which presents the basic idea of the Δ90-calibration in a very reduced form compared to this manuscript, and only for the rotation calibrator. Section 2 of this manuscript starts from the concept of that paper, and transfers the pure scalar concept there into the more general context of the Müller-Stokes formalism with adapted terminology, which serves as an introduction to the following sections.

It belabors a number of topics that have been thoroughly addressed in other published works while sometimes completely ignoring other aspects, creating the false impression of a comprehensive analysis.

The published works about these topics I am aware of are mentioned in the introduction. But none of those includes all the calibration techniques and all the error sources included in this manuscript. This manuscript describes several techniques in the same mathematical framework and compares the different advantages and disadvantages.

Already in the introduction I present the general setup of the sort of lidar systems which is covered by the presented model. It covers at least 80% of the lidar systems in EARLINET, and I guess this holds for other networks. I don't see the necessity to explicitly exclude everything which is not covered.

For all the meticulous steps in this section, there is absolutely no mention of oriented particles or multiple scattering. In my view, the reader may be given the false impression that this scattering matrix is somehow comprehensive, which it clearly is not, since it only covers single scattering by randomly oriented particles.

I am thankful for this hint. I include Section 13 (see below), which lists the assumptions and constraints of this model:

**Assumptions and constraints of the model**

1.     The correction of the standard signals (Sect. 4) and of the calibration factor (Sects. 5 ff) are only applicable in scattering ranges without aerosol or with randomly oriented, non-spherical particles with rotation and reflection symmetry as described in Sect. 2.1, and not for clouds with oriented particles as in cirrus and rain clouds (Kaul et al. (2004); Hayman et al. (2014); Volkov et al. (2015)). However, the scattering volume for the calibration measurements can be chosen to avoid oriented particles in the calibration range, and then the calibration corrections in Sects. 5 to 10 can be applied for the retrieval of the calibration factor $\eta = (\eta_R \, T_R) \, / \, (\eta_T \, T_T)$, which itself is general for the considered types of lidar set-ups in Fig. 1.

2.     We assume that the extinction in the range between the lidar and the scattering volume is polarisation independent and that signal contributions due to multiple scattering can be neglected.

3.      We assume that the atmospheric depolarisation in the calibration range does not change between the two measurements of the Δ90-calibrations. This can be verified by comparison of standard measurements before and after and maybe even between the two calibration measurements.

4.      Not considered are range dependent effects as the overlap function and the range dependent transmission and polarisation of interference filters and dichroic beam-splitters, which is caused by the range dependent incident angles on the optics.

5.      We assume that the optical elements of the lidar do not depolarize. Such depolarization can be caused by variable retardation or diattenuation over the aperture of optical elements, for example due to crystalline (e.g. $CaF_2$ and $MgF_2$ lenses) or stress birefringence. The latter can be present in all optical elements if they are inappropriately restrained in their holders. Larger optics, as e.g. telescope windows, can exhibit inherent stress birefringence due to annealing and/or their own weight. Such optical elements can easily be visually inspected by means of crossed polarising sheet filters before and after the sample. Furthermore, non-parallel (converging or diverging) incident beams on optics with polarisation effects depending on the incidence angle will cause depolarisation. The manufacturers specification of dedicated polarisation optics should be sufficient to determine the maximum allowable divergence of the incident beam, but, for example, the coatings of 90° reflecting mirrors in Newtonian telescopes are usually not sufficiently specified to determine their polarisation effects. The depolarising effects of optics can additionally depend on the state of polarisation of the incident beam.

It is good there is a table of variable definitions, but maybe it would be better to break them up a bit based on where they are used in the paper.

Most of the listed definitions are used in many places in the manuscript. The idea of one comprehensive list is to make the definitions easy to find in one place.

Be very clear about the assumptions applied in this work (preferably itemize them or keep them in a table).

See above, new Sect. 13.

In Eq. (14) (and others) pick a matrix form and stick to it.

The various matrix forms are quoted on purpose in order to avoid mistakes, which would certainly arise because the definitions are different in different textbooks. It is especially important to emphasize that the diattenuation parameter is used in this manuscript and not the diattenuation as in most other literature and textbooks.

There really isn't a reason to evaluate to a scalar equation unless it produces a result that is simple, concise and reveals some previously not obvious fact. That is, evaluating the matrices should produce a reduction in complexity, not the other way around.

The fully evaluated scalar equations are necessary for the complete error calculation, as described above. Furthermore, I present for each calibration setup some equations (special cases) which are reduced in complexity and show the advantages of avoiding certain error sources already in the lidar design. This analysis of "previously not obvious facts" is continued in Sects. 11 and 12.

Present the overall Mueller equation describing $I_s$ (Eq. (64)?), then give the evaluation in Eq. (68), but get rid of all those coefficients in front, which can just as easily fold into G and H and aren't important since absolute intensity measurements are almost never used in atmospheric lidar.

The "coefficients in front" and the absolute signal intensities are important for numerical error analysis when we include signal and background noise.

I don't want to fold the coefficients into G and H, because G and H are the parameters which describe the polarization and misalignment dependencies, while the coefficients in front are the unpolarised transmittances. G and H are simply 1 and ±1, respectively, for the ideal cases. Furthermore, also for the comparison of different lidar systems in intercomparison campaigns it makes sense to keep them separate.

The explicit definitions of G and H seem unnecessary.

With G and H the general Eqs. (62) and (65) can be formulated. This is the reduction in complexity as required by the reviewer. These general equations are a superset of several equations presented in other papers considering only a part of the error sources in this manuscript. They are very helpful for a general lidar data analysis software as described, e.g., in *Mattis et al., 2016, EARLINET Single Calculus Chain – technical Part 2: Calculation of optical products* (doi:10.5194/amt-2016-43 ).

Drop the  notation.

I include after Eq. (43):

This split-up of the equations in an analyser bra-vector and an input Stokes ket-vector is similar to the split-up in instrumental vectors of the transmitter and receiver in Kaul et al. (2004) and Volkov et al. (2015).

The  notation makes it easy to discern between the instrumental vectors of the transmitter and receiver. It allows for an elegant restructuring of the matrix equations as for example in Eqs. (133) and (177).

I don't really understand all this interest in obtaining "correction equations" for scalar polarization variables. The approach used in Kaul Appl. Opt. 2004, Hayman, Opt. Express 2012, Volkov Appl. Opt. 2015 and countless other polarimetry papers avoids any need to belabor "corrections" and just retrieves the relevant scattering matrix terms based on the lidar's operational parameters"errors" or whatever you want to call them.

Kaul (2004), Hayman (2012), and Volkov (2015) determine the gain ratio with a calibration in clean air ranges, where they assume molecular depolarization. This assumption can cause large errors. Furthermore, the mentioned papers don't describe any error calculation for systematic errors.

In this manuscript I describe various versions of the more robust Δ90-calibration, which must and can be corrected for systematic errors, if they are not already avoided in the lidar design. The necessary correction equations are described in Sects. 5 to 10. A numerical error calculation code can directly be written from the given equations.

In order to make the connections between the standard measurements, the model corrections and the additional calibration measurements more clear, I included in Section 5 the following figure and corresponding text:

Figure (4) shows the steps in which the measurements are corrected for systematic errors by means of the model. If all system parameters (Eqs. (56) and (57)) are known, the cross-talk parameters $G_S$ and $H_S$ can be calculated (see Eqs. (68) to (79)) and we only need to determine the calibration factor $\eta$ by means of calibration measurements in step 2 and its correction for systematic errors (step 3) as explained in Sects. 6 to 10. Under certain conditions some instrumental parameters can be determined by means of additional calibration measurements (step 4) described under "special cases" in Sects. 6 to 10 and in Sects. 11 and 12.

[Figure]

| measurements | model |
|---|---|
| **1. standard measurements** | **corrections (Sect. 4)** |
| $I_S = \eta_S \langle \mathbf{A}_S \| \boldsymbol{I}_{in} \rangle \Rightarrow$ | $I_S = \eta_S T_S T_O T_{rot} F_{11} T_E I_L (G_S + a H_S)$

 $F_{11} \propto \eta H_R I_T - H_T I_R;$ $\qquad \eta = \dfrac{\eta_R T_R}{\eta_T T_T}$

 $\delta = \dfrac{\delta^* (G_T + H_T) - (G_R + H_R)}{(G_R - H_R) - \delta^* (G_T - H_T)};$ $\quad \delta^* = \dfrac{1}{\eta} \dfrac{I_R}{I_T}$ |
| **2. calibration => gain ratio (Sect. 5)** | |
| $I_S(\mathrm{x},\varepsilon) = \eta_S \langle \mathbf{A}_S | \mathbf{C}(\mathrm{x},\varepsilon) | \boldsymbol{I}_{in} \rangle \Rightarrow \; \eta^*_{\varDelta 90}$ | |
| **3.** | **gain ratio correction => calibration factor (Sects. 6 to 10)** |
| $\eta^*_{\varDelta 90} \qquad \Rightarrow$ | $\eta = \dfrac{\eta^*_{\varDelta 90}}{K}$ |
| **4. combined calibration measurements** | **determination of instrumental parameters** |
| $I_S(\mathrm{x},\varepsilon) = \eta_S \langle \mathbf{A}_S | \mathbf{C}(\mathrm{x},\varepsilon) | \boldsymbol{I}_{in}(\alpha) \rangle$

 $\Bigg\} \Rightarrow$ | $D_O$ (Sects. 7.2, 8.2)
 $v_{in}, v_E$ (Sects. 9.1, 9.2, 9.3)
 $\varepsilon$ (Sect. 11)
 $\alpha$ (Sect. 12) |

Figure 4: Four steps for calibrating and correcting the standard measurements for systematic errors by means of the model equations and additional calibration measurements.